# Mechanism of AAA+ ATPase-mediated RuvAB–Holliday junction branch migration

Jiri Wald[1,2,3,4,5✉], Dirk Fahrenkamp[1,2,3✉], Nikolaus Goessweiner-Mohr[1,2,3,4,5,6], Wolfgang Lugmayr[1,2,3,4,5], Luciano Ciccarelli[1,2,4,5,7], Oliver Vesper[1,2,3,4,5] & Thomas C. Marlovits[1,2,3,4,5✉]

The Holliday junction is a key intermediate formed during DNA recombination across all kingdoms of life[1]. In bacteria, the Holliday junction is processed by two homo-hexameric AAA+ ATPase RuvB motors, which assemble together with the RuvA–Holliday junction complex to energize the strand-exchange reaction[2]. Despite its importance for chromosome maintenance, the structure and mechanism by which this complex facilitates branch migration are unknown. Here, using time-resolved cryo-electron microscopy, we obtained structures of the ATP-hydrolysing RuvAB complex in seven distinct conformational states, captured during assembly and processing of a Holliday junction. Five structures together resolve the complete nucleotide cycle and reveal the spatiotemporal relationship between ATP hydrolysis, nucleotide exchange and context-specific conformational changes in RuvB. Coordinated motions in a converter formed by DNA-disengaged RuvB subunits stimulate hydrolysis and nucleotide exchange. Immobilization of the converter enables RuvB to convert the ATP-contained energy into a lever motion, which generates the pulling force driving the branch migration. We show that RuvB motors rotate together with the DNA substrate, which, together with a progressing nucleotide cycle, forms the mechanistic basis for DNA recombination by continuous branch migration. Together, our data decipher the molecular principles of homologous recombination by the RuvAB complex, elucidate discrete and sequential transition-state intermediates for chemo-mechanical coupling of hexameric AAA+ motors and provide a blueprint for the design of state-specific compounds targeting AAA+ motors.

Homologous recombination is a fundamental cellular process involved in the maintenance of genetic integrity and the generation of genetic diversity across all domains of life. The central and universal element in genetic recombination as well as in double strand break repair and in the process of replication fork rescue is a four-way DNA heteroduplex called the Holliday junction[1,3,4]. In prokaryotes, the two proteins RuvA and RuvB play critical roles in the processing of the Holliday junction by promoting the ATP-dependent unidirectional strand-exchange reaction known as active branch migration[2,5–11]. Previous biochemical and structural evidence suggests that branch migration is facilitated by a tripartite complex: RuvA tetramers assemble around the Holliday junction crossover to provide structural guidance for DNA separation and rewinding and are flanked by two hexameric RuvB AAA+ ATPases that together fuel the translocation of the newly emerged recombined DNA[12–19]. Furthermore, these studies demonstrated that domain III of RuvA (RuvA[D3]) binds to the presensor-1 β-hairpin of RuvB, a distinguishing feature of the PS1 insert superclade[20,21], regulates branch migration and increases ATPase activity of the RuvB motor[22,23]. Moreover, the ability for cross-species hetero-complementation established the existence of a robust and conserved mechanism of the RuvA- and RuvB AAA+-coordinated action at the Holliday junction[24,25]. Despite the large body of knowledge, the structure of the RuvAB–Holliday junction complex (hereafter referred to as RuvAB–HJ) and the molecular mechanisms by which the RuvB AAA+ motors drive the translocation of DNA to facilitate one of the most basic biological processes in living organisms—namely the maintenance and exchange of genetic information[26,27]—remain unknown. To unravel the architecture and decipher the operating principles of the RuvAB machinery, we applied time-resolved cryo-electron microscopy (cryo-EM) and single-particle analyses of in vitro reconstituted RuvAB complexes processing a Holliday junction. Our structural analyses reveal a highly coordinated conformational landscape of an active RuvAB branch migration complex and uncover the dynamic interplay between a completely resolved nucleotide cycle in a rotating RuvB AAA+ motor as well as DNA translocation. Furthermore, we show that RuvB motors translocate the DNA as molecular levers in an ATP-dependent power stroke to convert chemical energy to mechanical force.

[1]Institute of Structural and Systems Biology, University Medical Center Hamburg-Eppendorf, Hamburg, Germany. [2]Centre for Structural Systems Biology, Hamburg, Germany. [3]Deutsches Elektronen Synchrotron (DESY), Hamburg, Germany. [4]Institute of Molecular Biotechnology GmbH (IMBA), Austrian Academy of Sciences, Vienna, Austria. [5]Research Institute of Molecular Pathology (IMP), Vienna, Austria. [6]Present address: Institute of Biophysics, Johannes Kepler University (JKU), Linz, Austria. [7]Present address: GlaxoSmithKline Vaccines, Siena, Italy. ✉e-mail: jiri.wald@cssb-hamburg.de; dirk.fahrenkamp@cssb-hamburg.de; marlovits@marlovitslab.org

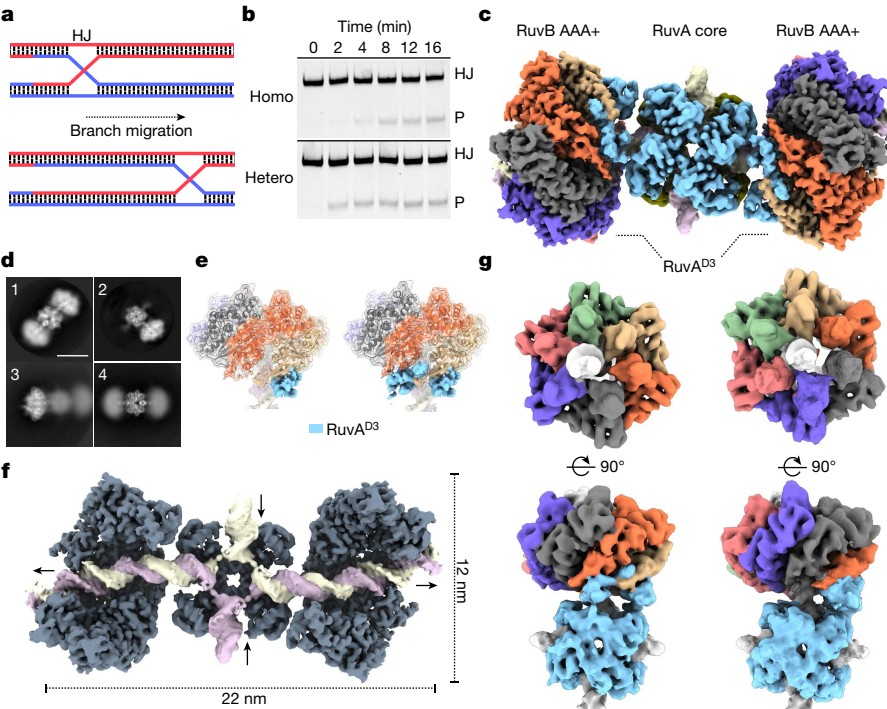

**Fig. 1 | Structure of the RuvAB–HJ complex. a**, Schematic diagram of the Holliday junction branch migration. HJ, Holliday junction. **b**, RuvAB homo- and hetero-complexes are active for branch migration. Comparison of the activity using fluorescently labelled Holliday junction (8 nM) recombinant *S. typhimurium* RuvA (60 nM) and recombinant RuvB originating either from *S. typhimurium* (160 nM) (homo) or *S. thermophilus* (160 nM) (hetero). The experiments were repeated three times. P, product. **c**, Cryo-EM composite map of the RuvAB complex (molecular mass approximately 650 kDa) bound to the Holliday junction. The absolute RuvA:RuvB stoichiometry is 8:12. Two RuvA tetramers (light blue (front) and olive (back)) sandwich the Holliday junction. The C-terminal RuvA$^{D3}$ domains extend from the central core and bind to the RuvB motor. **d**, RuvAB–HJ particles are highly flexible. Representative 2D classes from tripartite (1) and bipartite (2) particles. Focused classifications on one of the RuvB motors (3) or the central RuvA–HJ core (4) highlight the overall flexibility of tripartite particles. Scale bar, 10 nm. **e**, RuvB motors bind to one or two RuvA$^{D3}$ domains (blue). The two RuvA$^{D3}$ domains bind to adjacent RuvB subunits in the RuvB motors. **f**, RuvAB–HJ complex in which substrate-disengaged RuvB subunits and one RuvA tetramer are removed to visualize the Holliday junction and the interaction of each RuvB motor with its cognate DNA substrate. Arrows show the direction of movement of DNA at the entry to the RuvA core and the exit of the new DNA duplex from the RuvB motors. Dimensions of the complex are indicated. **g**, Configuration of RuvB hexamers that undergo a rotation by 60° relative to the RuvA core. Focused 3D classes (end-on (upper panel) and side views (lower panel) using a mask enclosing one RuvB motor and the central RuvA core. Interacting RuvA$^{D3}$ domains as well as conformation-specific RuvB subunits are rotated by 60° with respect to the RuvA–HJ core.

## Structure of the RuvAB–HJ complex

Branch migration of Holliday junctions driven by the RuvAB machinery is a fast and highly dynamic process that is essential during DNA recombination[28,29] (Fig. 1a). To visualize this process, we reconstituted RuvAB–HJ complexes in vitro from individually purified components originating from *Salmonella typhimurium* and *Streptococcus thermophilus*, respectively, and tested their function in a branch migration assay (Fig. 1b). Both homo- (RuvA and RuvB from *S. typhimurium*) and hetero- (RuvA from *S. typhimurium* and RuvB from *S. thermophilus*) complexes processed the Holliday junction similarly upon addition of ATP, suggesting a highly conserved underlying mechanism, owing to interchangeability of individual components (Fig. 1b and Extended Data Fig. 1a–h). To capture the catalytic steps of this rapid process, we first slowed down the reaction by replacing ATP with an equimolar mixture of the slowly hydrolysable ATPγS[30] and ADP and incubated the reaction on ice either for 30 min (dataset t1) or for 5 h (dataset t2) to mimic an initiation and an equilibration phase of the RuvAB–HJ complex (Extended Data Fig. 1h). Subsequent vitrification of samples led to aggregates and low numbers of individual particles for homo-complexes, whereas the distribution of hetero-complexes over the grid was largely monodisperse and suitable for single-particle analysis (Extended Data Fig. 1f–j). The cryo-EM structure of the RuvAB–HJ complex resolved to a resolution of 8 Å revealed highly flexible and linearly arranged tripartite

assemblies, with eight RuvA molecules symmetrically arranged in two tetramers (3.3 Å resolution) and the four-way Holliday junction flanked by, and flexibly connected to, one or two RuvB hexamers (2.9–4.1 Å resolution) as well as bipartite particles (3.9 Å resolution) (Fig. 1c–e and Extended Data Figs. 2a–c, 3a–d and 4a–b and Extended Data Table 1). This architecture is consistent with previously proposed models of the RuvAB machinery[14,15,17,22,23,31]. In both particle types, DNA enters and exits the RuvA core as a double helix, with one or two hexameric RuvB motors engaging the minor groove of the rejoined DNA (Fig. 1f). The RuvA core is physically connected to both RuvB motors through RuvA$^{D3}$ (Fig. 1c). On either side, two RuvA$^{D3}$ domains are bound to adjacently positioned RuvB subunits, indicating that these domains could cooperate to control the two RuvB AAA+ motors (Fig. 1c,e). Notably, all four RuvB-coordinating RuvA$^{D3}$ domains localize to the same side of the Holliday junction crossover (Extended Data Fig. 4f), implying that a single RuvA tetramer might be sufficient to operate both RuvB motors simultaneously. These findings are also in agreement with the proposed architecture of the RuvABC resolvasome, in which the Holliday junction is believed to be sandwiched by one RuvA tetramer and a dimer of the resolvase RuvC[32,33] (Extended Data Fig. 4g).

To investigate the flexibility of RuvAB–HJ complexes, we subjected our particles to further three-dimensional classifications. This analysis revealed that, besides the overall flexibility, in about 7% of the bipartite particles and about 6% of the tripartite particles, the position of

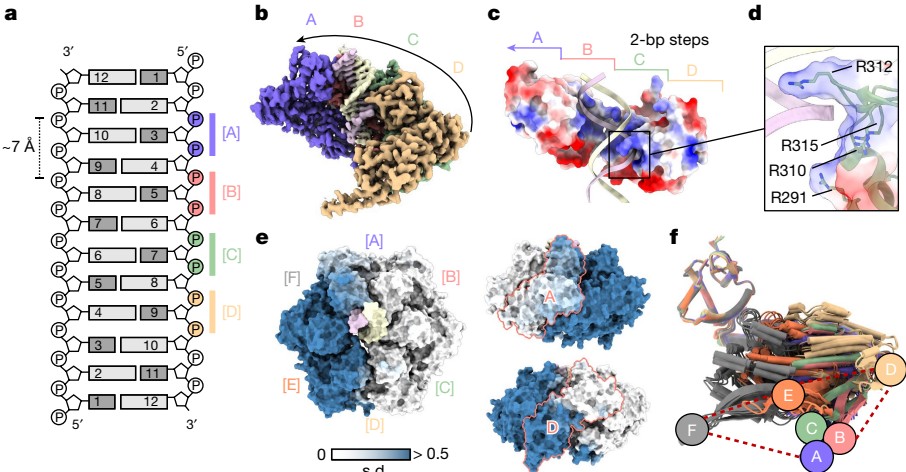

**Fig. 2 | Architecture and conformational variability of the RuvB AAA+ motor. a**, Schematic of the interface between the DNA substrate and the four staircase RuvB subunits (A, B, C and D). The subunits engage the DNA substrate along the phosphate backbone of only one DNA strand spaced by approximately 7 Å along the DNA axis (every two nucleotides). **b**, Cryo-EM map highlighting the formation of a spiral staircase by the DNA-interacting subunits (A, B, C and D) (in this view, subunits E and F, which are not binding the DNA, are removed). **c**, Surface charge representation of the head domains of the DNA-binding interface formed by the RuvB staircase (A, B, C and D). **d**, The spiral staircase forms a positively charged cleft composed of arginine residues (Arg291,

Arg310, Arg312 and Arg315) from A to D to bind one strand of the double-stranded DNA. **e**, Surface representation and variability analysis of RuvB. The analysis divides the RuvB hexamer into a rigid (white) and a flexible (steel blue) area, connected by the border subunits A and D. Colouring according to the standard deviation of the distance of Cα atoms (atomic models were aligned to RuvB subunit C). **f**, Superposition of 30 RuvB subunits extracted from the five hexameric RuvB motor states (s1 to s5). RuvB subunits were aligned to the head domain of RuvB. Coloured labels represent similar conformations (conformational clusters [A] to [F]).

DNA-engaged RuvB with respect to the RuvA–HJ core is rotated by around 60°. This suggests that the RuvB motors are able to rotate and that after completion of a 60° rotation, each RuvB subunit takes the position occupied by its neighbour before the rotation (Fig. 1g, Extended Data Fig. 5a,b and Supplementary Video 1). The rotation is further evidenced by multibody refinement analysis in which it accounts for around 45% of the total flexibility in the particles (Extended Data Fig. 5c–e and Supplementary Video 2). Thus, we reasoned that the reconstituted RuvAB complex is enzymatically active and has therefore been imaged in distinct conformational states. Moreover, our data reveal that the previously described continuous rotation of the DNA substrates[34] is accompanied by a concomitant rotation of the RuvB AAA+ motors themselves.

## Conformational landscape of RuvB motors

To understand how rotation of the RuvB motor is linked to branch migration, we applied iterative focused refinement together with rigorous three-dimensional classification to the RuvB hexamers from our t2 dataset. This analysis revealed 9 structurally distinct RuvB motor maps at resolutions ranging between 2.9 and 4.1 Å (Extended Data Figs. 2c,e and 3e–n and Extended Data Table 1). Two of these maps (at 3.9 and 4.1 Å resolution) could not be improved to a resolution that would allow unambiguous assignment of nucleotides and were therefore not considered further. The remaining seven RuvB motors can be grouped according to the number of bound RuvA[D3], with one map lacking RuvA[D3] (s0[−A]), two maps containing one RuvA[D3] (s0 and s1) and four maps showing two bound RuvA[D3] domains (s2, s3, s4 and s5), together suggesting a dynamic interplay between RuvA[D3] and the RuvB motors.

All RuvB motors assemble into closed, asymmetric hexamers, featuring an approximately 2 nm-wide central pore that accommodates the DNA (Extended Data Fig. 3e–m). Consistent with previous structural and interaction studies, RuvB oligomerization is driven by the large (RuvB[L]) and small (RuvB[S]) ATPase domains of adjacent subunits[18,19,35] (Extended Data Fig. 4c–e). Similar to other AAA+ translocases[36–39], four RuvB subunits (A, B, C and D) together assemble into a 'spiral staircase'. This generates a continuous interface that primarily binds one of the two DNA strands

(Fig. 2a,b), highlighting that only one strand from each double-stranded DNA entering the RuvA core is held by one RuvB motor (Extended Data Fig. 4h). The two remaining RuvB subunits (E and F) close the spiral staircase, but do not bind the DNA. The DNA-engaged subunits (A, B, C and D) bind the DNA through their C-terminal head domains (RuvB[H]). Each RuvB[H] domain harbours four conserved arginine residues Arg291, Arg310, Arg312 and Arg315, which generate a positively charged binding interface complementary to the negatively charged DNA backbone (Fig. 2c,d). (To aid comparison with the *Escherichia coli* RuvAB system, the corresponding residues are listed in Supplementary information Tables 1 and 2). The repeated binding pattern of the arginine residues originating from each of the subunits engages with the DNA separated by the distance of two nucleotides (approximately 7 Å). Moreover, as the RuvB subunits are positioned around 60° apart from each other within the RuvB hexamer, these data further imply that the rotation of the RuvB motors is linked to the events occurring within one translocation step.

To investigate the overall conformational plasticity of the hexamer, we analysed the variability for each Cα atom over all seven distinct motor structures expressed as the standard deviation of the distances to their corresponding centroids (Fig. 2e). This revealed that the RuvB hexamer can be divided into rigid (white), flexible (blue) and intermediate regions. Whereas the rigid area contains the DNA-bound subunits B and C, the DNA-disengaged subunits E and F reside in the flexible part. Notably, the DNA-bound subunits A and D, which connect the two unequal halves at the top and at the bottom of the staircase, respectively, are in intermediate regions, suggesting that the differential flexibility within the hexamer is involved in RuvAB-mediated branch migration. Of note, the extent of the variability is not necessarily confined to an entire RuvB subunit as exemplified for subunits A and D, which show both flexible and rigid areas (Fig. 2e). To further assess the plasticity of individual RuvB subunits, we determined the average root mean squared deviation (r.m.s.d.) between the 42 RuvB subunits, and also between their individual domains: RuvB[L] (residues 21–181), RuvB[S] (residues 182–254) and RuvB[H] (residues 255–330). This analysis revealed a low average r.m.s.d. (r.m.s.d.$_{\varnothing}$ 1.2 Å, 0.48 Å and 0.453 Å, respectively) for each domain (Extended Data Fig. 6a) showing that the overall structures

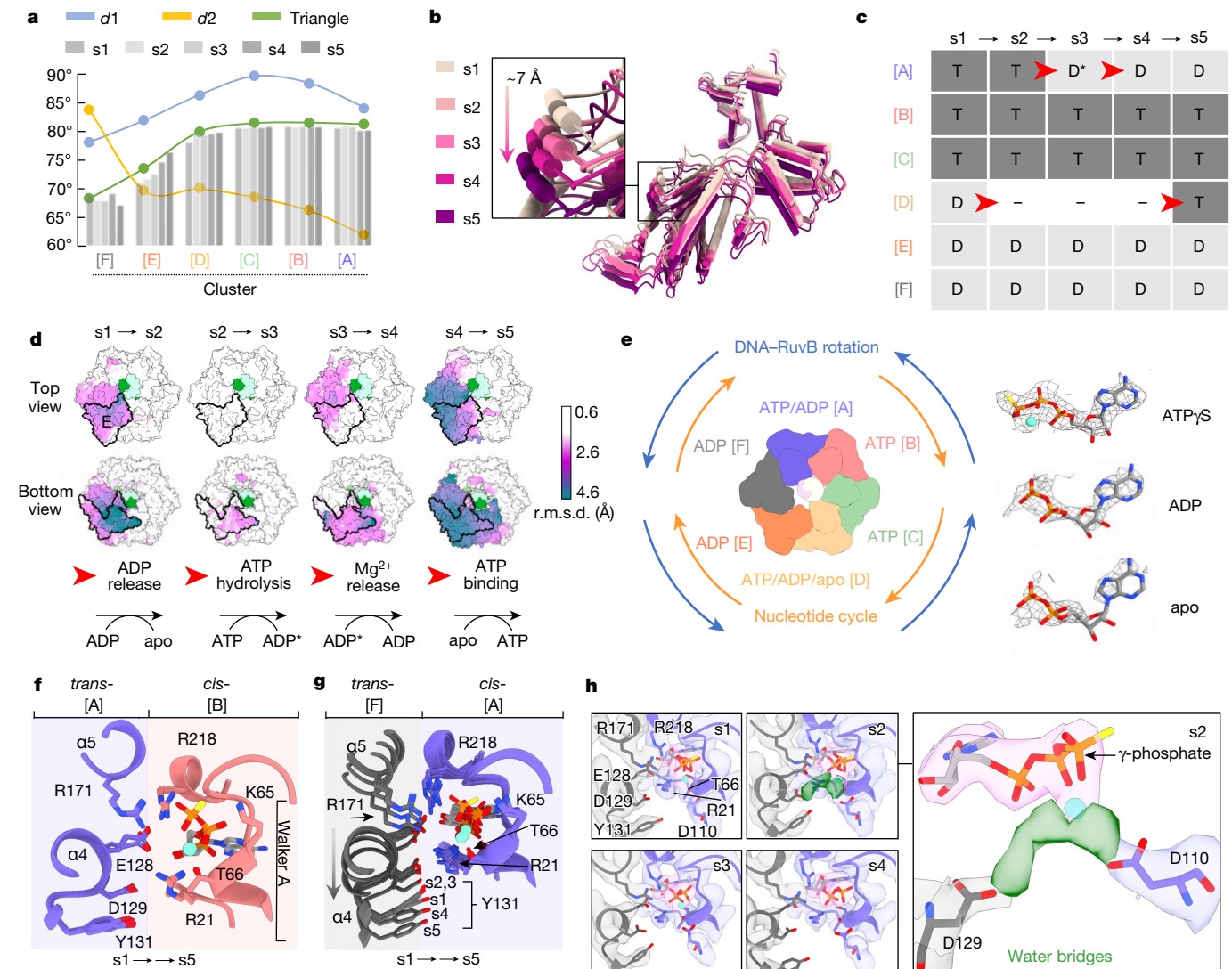

**Fig. 3 | Dynamics and nucleotide pocket analysis of the RuvB motor.**
**a**, Analysis of conformational clusters of RuvB subunits from state s1 to s5 using dihedral (*d*1 and *d*2) and triangle angles (Extended Data Fig. 6). Columns represent the triangle angle of individual RuvB subunits across the states. **b**, Unidirectional trajectory (arrow) of the subunits in cluster [E], covering a distance of around 7 Å. RuvB hexamers were superimposed on subunit C in the rigid area of the hexamer. **c**, Nucleotide occupancy of all RuvB subunits within hexamers in states s1 to s5. ATP hydrolysis and nucleotide exchange occur exclusively in cluster [A] and [D], respectively, and follow a chronology of events (red arrows) starting with ADP release (cluster [D], s1→s2), ATP hydrolysis (cluster [A], s2→s3→s4) and ATP uptake (cluster [D], s4→s5). Note: correlating this order of events leads to the conformational trajectory shown in **b**. **d**, Areas of conformational plasticity of the RuvB hexamer transition through states s1 to s5 measured as the r.m.s.d. of the Cα atoms between two consecutive RuvB motor states. All states were aligned to the DNA (green,

light green). For reference, subunit E is outlined in top and bottom views. **e**, Schematic showing relative directionalities of progression around the ring of the nucleotide cycle (orange) and DNA rotation together with RuvB (blue). Right, representative cryo-EM densities within the nucleotide-binding pockets and modelled nucleotides for ATPγS, ADP and apo. **f**, ATP-non-hydrolysing nucleotide-binding pocket. Superposition of the interfaces that form the nucleotide-binding pocket of subunit A (*trans*) and B (*cis*) from s1 to s5. The ligand pocket stays largely invariant. **g**, The ATP-hydrolysing nucleotide-binding pocket: superposition of the interfaces that form the nucleotide-binding pocket between subunit A (*cis*) and F (*trans*) from s1 to s5. α-Helices α4 and α5 in subunit F are gradually displaced leading to ATP hydrolysis. **h**, Magnification highlights unmodelled cryo-EM density (green density) in state s2, which probably corresponds to ordered water molecules initiating the nucleophilic attack on the ATP γ-phosphate.

of RuvB[L], RuvB[S] and RuvB[H] remain largely constant, yet their position relative to each other varies (Fig. 2f and Extended Data Fig. 6a,b). The presence of the DNA substrate within the hexamer further enabled us to determine that RuvB subunits display position-specific, distinct conformations, which hereafter are referred to as clusters (with cluster [A] corresponding to the position of subunit A in RuvB, cluster [B] corresponding to the position of subunit B, and so on) (Fig. 2f).

We then quantified the structural plasticity within RuvB clusters from state s1 to s5 by measuring two dihedral angles (*d*1 between RuvB[L] and RuvB[S], and *d*2 between RuvB[S] and RuvB[H]) and one triangle angle,

which provides a more holistic view on the changes occurring in RuvB (Extended Data Fig. 6c). We found that each of the RuvB clusters ([A] to [F]) is characterized by a unique combination of the three angles, and thus harbours a set of RuvB subunits with more similar conformations (Fig. 3a). RuvB is also subject to deformation within clusters and is most variable in cluster [E], in which the triangle angle covers a dynamic range of 5.6° (Fig. 3a and Extended Data Fig. 6c). To better characterize the motions in this flexible area of the RuvB hexamer, we aligned the five structures s1 to s5 to the almost invariant subunit C and analysed the movements of all the other subunits (Extended Data

Fig. 6d–f). This approach revealed that sequential conformational changes within cluster [E] can be described along a trajectory with an average length of around 7 Å (range: 6–10 Å), which is directed towards the RuvA–HJ core (Fig. 3b). Notably, the length of the trajectory within cluster [E] corresponds well to the step size of the RuvB staircase of two nucleotides (the distance between nucleotides in DNA is approximately 3.5 Å), suggesting that the five RuvB structures (s1 to s5) could represent consecutive atomic snapshots of an active RuvB motor as it progresses through one translocation step.

## Nucleotide cycle and conformational states

To investigate the interdependence between the observed conformational changes in RuvB hexamers and ATP hydrolysis, we first analysed the nucleotide identity and occupancy for all thirty nucleotide-binding pockets (Extended Data Fig. 7a,b). We found that cluster [A], which is positioned at the top of the staircase, contains either ATPγS (s1 and s2), ADP + $Mg^{2+}$ (s3) or ADP (s4 and s5), a configuration that is consistent with a progressing ATP hydrolysis reaction at this pocket. At the opposing lower side of the hexamer, cluster [D] contains either ADP (s1), fragmented and interrupted densities (s2 to s4) or ATPγS (s5). The fragmented and interrupted densities are indicative of low nucleotide occupancy, suggesting that these sites have an apo-like configuration. The DNA-bound clusters [B] and [C] are occupied exclusively by ATPγS, contrasting with the DNA-disengaged clusters [E] and [F], which have only ADP bound (Fig. 3c).

Irrespective of the previous ordering on the basis of conformational changes along a trajectory, the nucleotide cycle of the five states revealed the same sequence of structural states (s1→→s5), and thus independently validates their ordering: the cycle starts with ADP release in subunit D (s1→s2), followed by the catalytic reaction through three states in subunit A (ATP→ADP + $Mg^{2+}$→ADP (s2→s3→s4)) and is completed by ATP uptake in subunit D (s5) (Fig. 3d). These data highlight that ATP hydrolysis and nucleotide exchange occur in opposite clusters located at the top [A] and bottom [D] of the staircase, respectively, and individual steps are spatiotemporally separated (Fig. 3c–e). The need for structural cohesion to cycle between oppositely located subunits and the concomitant conformational changes described above suggests that there is an interlocked signalling chain between the subunits that connects the nucleotide cycle and, ultimately, DNA translocation.

Remarkably, the DNA remains bound to all four staircase subunits (A to D) across all five states and thus the interaction of the DNA substrate with these subunits is independent of the type of nucleotide bound, including at the ATP hydrolysis (subunit A) and at the exchange position (subunit D) (Extended Data Fig. 7c). Consequently, our data reveal that in order to relocate the DNA substrate inside the central RuvB motor pore, RuvB subunits must be subject to additional conformational changes that follow the nucleotide cycle. We therefore reason that the nucleotide cycle in fact functions first to prime the RuvB subunits over five states to then acquire the conformations of their respective neighbouring clusters (Extended Data Fig. 6f). This is also supported by the fact that the nucleotide arrangement in state s5 corresponds to the same configuration as in state s1, but the respective conformations of the six subunits have shifted forward by one to occupy the new successor state (s5→s1′: A(s5) →F(s1′), B(s5) →A(s1′), and so on). When this event occurs, all six RuvB subunits simultaneously transition to the next conformational cluster without any additional changes to the nucleotide arrangement (subunits in s5 and s1′ have the same nucleotide occupancy), resetting the conformation of the entire hexamer to state s1. We therefore refer to this process as a 'cluster switch' (s5→s1′) (Extended Data Fig. 8). It follows that all subsequent processes now take place in the respective adjacent subunit, implying that nucleotide hydrolysis and all other processes operate around the hexameric ring in repeated sequences.

## Reorganization of the catalytic centre

To gain structural insights into the events occurring at the nucleotide-binding pockets, we first analysed their common features. Nucleotides bind at the interface of two consecutive subunits (cis and trans), with the nucleoside exclusively clamped between the RuvB$^S$ and RuvB$^L$ domain of one subunit[35,40] (RuvB in cis) (Fig. 3f and Extended Data Fig. 9a,b). In all ATP-containing pockets, a conserved Walker-A motif binds the ATPγS–$Mg^{2+}$ complex in which previously identified Lys65 interacts with the ATP γ-phosphate[41], and Thr66 coordinates the $Mg^{2+}$ ion (Fig. 3f). Additional contacts are provided by two conserved cis-acting arginine residues: Arg21 and the sensor 2 arginine Arg218[42]. Arg21 is located at the N terminus and binds the ATP α-phosphate, whereas sensor 2 arginine Arg218 is in the small ATPase domain and mediates nucleotide sensing (Fig. 3f). In agreement with previous studies, ATPγS-$Mg^{2+}$ trans-sensing is achieved by two elements: a conserved signature motif (Glu127–Asp130), located on α-helix α4, and trans-acting Arg171 on α-helix α5[40,43] (Fig. 3f). Thus, Arg171 represents the canonical arginine finger that is conserved in most AAA+ ATPases and directly coordinates the γ-phosphate[44]. Furthermore, two additional acidic trans-residues, Glu128 and Asp129, sense cis-residues Arg21 and Arg218, respectively, and thus indirectly stabilize nucleotide binding (Fig. 3f).

To understand the molecular mechanism and chemistry of coupling ATP hydrolysis and signal transduction, we followed the fate of ATPγS before (s1), during (s2) and after (s3–s5) hydrolysis in subunit A, whose nucleotide-binding pocket interfaces in trans with DNA-disengaged subunit F. During the transition through the catalytic states (s1→→s5), helices α4 and α5 from subunit F undergo a concerted motion, which enables distinct local rearrangements of trans-residues critical for ATP hydrolysis in subunit A (Fig. 3g). In particular, the intermolecular interaction between trans-Glu128 and cis-Arg21, which is maintained in state s1, is lost in the following states, enabling trans-Glu128 to instead engage with the canonical arginine finger trans-Arg171. Further, in state s2, residue trans-Tyr131 joins cis-Arg21 in coordinating trans-Asp129, an event that coincides with the appearance of continuous density between trans-Asp129 and the ATPγS-$Mg^{2+}$ complex (Fig. 3h and Extended Data Fig. 9c). The connecting density is best described as ordered water molecules, which are required to facilitate the nucleophilic attack on the ATP γ-phosphate. The importance of this signature motif has been highlighted by mutational studies, in which the substitution of trans-Asp129 markedly compromised branch migration activity, and mutation of trans-Glu128 resulted in a bacterial growth defect[45]. As an additional validation of the ATP hydrolysis reaction taking place in subunit A of state s2, connecting density also emerges between the ATP γ-phosphate and the Walker-B motif residue cis-Asp110, which, similar to trans-Asp129, has been shown to be important for ATP hydrolysis[45,46] (Fig. 3h and Extended Data Fig. 9a).

In the next states, progression of the ATP hydrolysis reaction can be observed, which first results in the release of the γ-phosphate (s2→s3) (Fig. 3h and Extended Data Fig. 9a). As a result, the binding of sensor 2 cis-Arg218 to the nucleotide is released, whereas the coordination of the $Mg^{2+}$ ion through cis-Thr66 remains intact (Fig. 3h and Extended Data Fig. 9a). In the next transition (s3→s4), loss of the $Mg^{2+}$ ion liberates cis-Thr66, which now coordinates the ADP β-phosphate. Subsequently, (s4→s5) cis-Arg218 of sensor 2 moves away from its own binding pocket and demarcates subunit A to be primed to undergo a cluster switch.

## Information relay through the converter

The fact that we observed specific binding of RuvA$^{D3}$ to the RuvB hexamer opposite the catalytic centre in subunit A through all states (s1 to s5) at the bottom of the staircase does not explain an increase in ATPase activity in the presence of RuvA[9]. Instead, it suggests that RuvA$^{D3}$ in such an arrangement elicits a regulatory function onto the nucleotide cycle

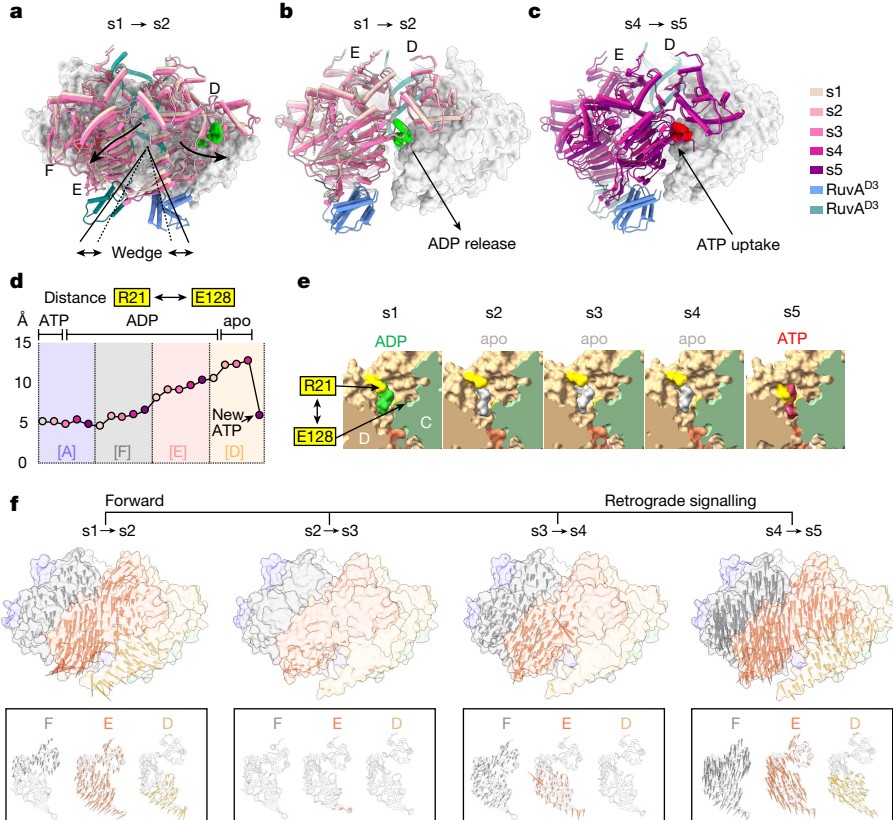

**Fig. 4 | RuvA operates the ATP hydrolysis cycle in the RuvB motor.**
**a**, The RuvA[D3]-induced wedge-like effect on the converter in the RuvB motor. Colours correspond to nucleotide cycle states (s1 and s2). Arrows indicate the displacement of the domain cores of RuvB[L] (subunit D) and RuvB[L] (subunit E) through the binding of the second RuvA[D3] (green). **b,c**, Long-range inter-subunit signalling triggered by RuvA[D3] binding causes conformational changes on subunit D resulting in ADP release (s1→s2) (ADP green) (**b**) and ATP uptake (s4→s5) (ATP red) (**c**). During s1 to s2, gate opening in subunit D allows ADP release, whereas gate closure during transition s4 to s5 is associated with the uptake of ATP molecule. Subunits D, E and F are shown in cartoon view; subunits A, B and C shown in surface view. **d**, Distance between *cis*-Arg21 and *trans*-Glu128 as a measure for gate-opening and gate-closing motions of the RuvB N terminus during the nucleotide cycle. Gate opening starts in cluster [F], continues in [E] and reaches its maximum in cluster [D], which results in the

release of ADP. **e**, Close-up view of gate opening in cluster [D]. *Cis*-Arg21 and *trans*-Glu128 are shown in yellow. Bound nucleotides are shown for ADP (green) and ATP (red). The apo-like state is represented by the volume enclosing an ADP molecule (grey). **f**, Spatiotemporal deconvolution of sequential signalling of the converter through the individual transitions s1→→s5. Conformational changes are shown as arrows within the context of the hexamer or the individual subunits (F, E and D; bottom row). Arrows indicate the directionality and magnitude of movements for the indicated transitions, where the base and the tip of the arrow represents the coordinate of the Cα atom at the start and the end of the transition. Arrows are coloured according to their subunit identity, only every other distance larger than 1 Å is shown, and arrow lengths are multiplied by a factor of 2.5. Structures shown in surface view represent the hexamer of the respective state at the start of the transition.

and directly coordinates branch migration. In particular, we found that a single RuvA[D3] is bound to subunit D during all five states, revealing that the RuvA–HJ complex is tethered to both opposing RuvB motors in tripartite particles throughout the entire nucleotide cycle. By contrast, a second RuvA[D3] binds exclusively to subunit E in states s2 to s5 (Extended Data Fig. 10a). Both RuvA[D3] bind to a previously described hydrophobic composite interface in their respective RuvB subunits, which is composed of RuvB[L] α-helix α3 and the presensor-1 β-hairpin[15] (Extended Data Fig. 10b), which in other hexameric AAA+ motors of the PS1 insert superclade coordinates with their substrates either directly or indirectly[20,36,47]. Analysing the effect of the RuvA[D3] recruitment (s1→s2) to subunit E revealed that the binding event exerts a wedge-like effect on the RuvB hexamer, which drives apart the large domains of subunits E and D. The motion of subunit E suggests that RuvA[D3] binding is achieved by an induced-fit mechanism (Fig. 4a, Extended Data Fig. 10c and Supplementary Video 3). The repositioning of subunit E causes a concomitant displacement of the large ATPase domain of subunit D, which then promotes the opening of its nucleotide-binding pocket and thereby enables the escape of the ADP molecule (Fig. 4b, Extended Data Fig. 10d and Supplementary Videos 4 and 5). Thus, our data reveal

that RuvA[D3] (binding to subunit E) functions as a nucleotide exchange factor by acting on subunit D. Notably, at the same time, repositioning of E causes a motion of the adjacent, DNA-disengaged subunit F, whose *trans*-acting residues Glu128, Asp129 and Arg171 facilitate the ATP hydrolysis reaction in A as described above (Fig. 3g,h, Extended Data Fig. 10e and Supplementary Videos 4 and 5). On the basis of this observation, we postulate that RuvA[D3] also acts through its binding to the presensor-1 β-hairpin on subunit E as an ATPase-activating domain that stimulates ATP hydrolysis in A through forward coordinated, inter-subunit signalling.

Of note, the *trans*-acting residues in subunit F disconnect from the nucleotide only upon loss of the Mg²⁺ ion, which in turn permits a large-scale motion of subunit F (s4→s5) (Extended Data Figs. 6e,f and 9). Releasing subunit F from its association with ADP in RuvB subunit A sets in motion a chain reaction, which also affects the position of DNA-disengaged subunit E. Thus, our data uncover that the dissociation of the Mg²⁺ ion triggers retrograde inter-subunit signalling confined within the flexible RuvB subunits (D, E and F) (Fig. 2e and Extended Data Fig. 6e,f). As one of the consequences, the gate-keeping *cis*-Arg21 of subunit E can no longer coordinate the ADP α-phosphate in

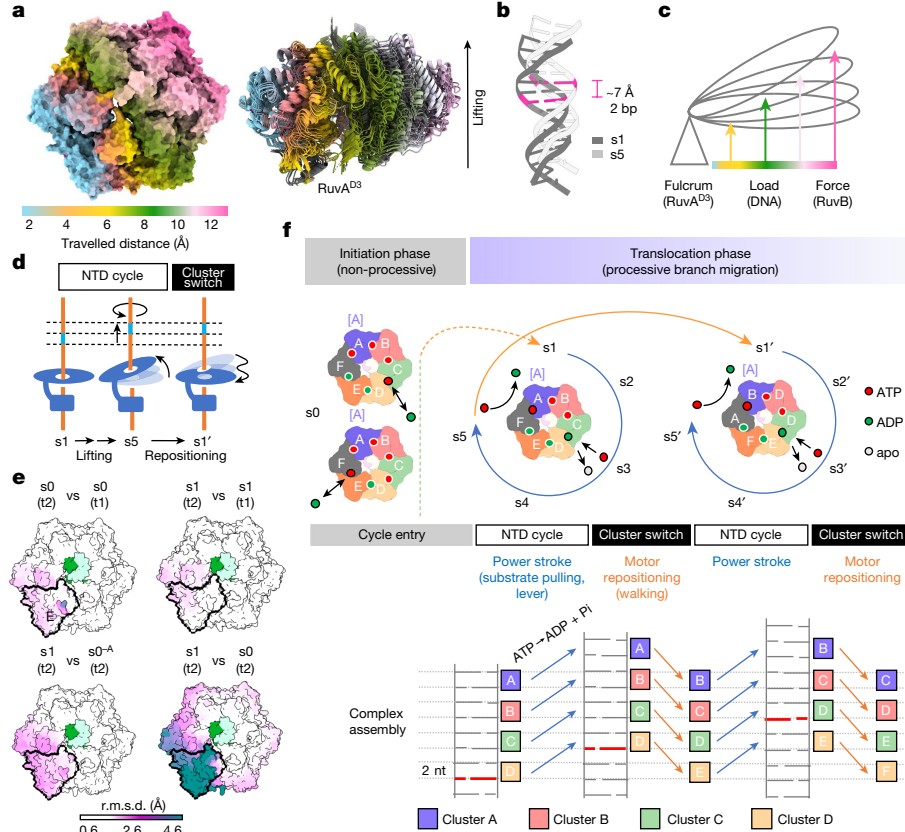

**Fig. 5 | RuvB motors facilitate substrate translocation by a lever mechanism. a**, Left, surface view of the RuvB hexamer coloured according to the change of height as a function to the distance to cluster [E]. Right, superposition of side views (cartoon) of individual RuvB motor states on RuvB subunit E. **b**, The DNA together with its bound RuvB subunits located at the centre of the RuvB hexamer is lifted by approximately 7 Å, equivalent to the distance of two nucleotides along the DNA (DNA in state s1 (grey) and s5 (white). **c**, Schematic of the RuvB motor lever mechanism. Binding of RuvA$^{D3}$ to substrate-disengaged RuvB subunit E generates a fulcrum, which enables RuvB to convert the energy contained in ATP into a lever action. **d**, Illustration of the difference between motor lifting (pulling) the DNA substrate accompanied by rotation and motor repositing (walking). **e**, Structural similarity of the converter between initiation (s0) and processivity states (s1) obtained by time-resolved cryo-EM (t1 and t2) and expressed as per-residue r.m.s.d. between corresponding Cα atoms. The s0 states in both datasets closely resemble each other but differ from the s1 states. Similarly, both s1 states are

very similar to each other, but differ from the s0 states. States are aligned to the DNA. **f**, Model for Holliday junction branch migration through continuous DNA translocation mediated by RuvAB. Initiation states provide a potential entry pathway (s0→s1) into the nucleotide cycle, which starts with state s1. The nucleotide cycle is represented by states s1 to s5, and nucleotide occupancy at subunit interfaces is schematized as coloured circles (red (ATP), green (ADP) and grey (apo)). During the nucleotide cycle, the energy contained in ATP is converted into a lever action or power stroke, causing DNA translocation of two base pairs per hydrolysed ATP molecule . This is also indicated by the translocating base pairs (cyan and red) of the Holliday junction crossover. Cluster switches, in which RuvB subunits undergo 'register shifts', cause the repositioning (walking) of the DNA substrate in the central pore and regeneration of state s1. This enables RuvB motors to generate iterative power strokes, and thus provides the mechanistic basis for continuous branch migration.

its nucleotide-binding pocket, which in turn causes the entire N terminus to fold away from the pocket (Fig. 4d,e). This prepares subunit E for the release of ADP in the next translocation step, when the cluster switch has occurred and subunit E has transitioned into the conformational cluster [D] where it is finally subject to nucleotide exchange. This is further reflected by a constantly increasing $d1$ and triangle angles in cluster E (Fig. 3a and Extended Data Fig. 6c), which, on a molecular level, weakens the hydrophobic interaction between N-terminal *cis*-Leu20 and its *cis*-binding partners Thr193, Ile196, Phe197 and Asn221. As a result, the destabilization of *cis*-Leu20 impairs the ability of *cis*-Arg21 to coordinate the ADP α-phosphate (Extended Data Fig. 10f and Supplementary Video 6). In addition, the retrograde signalling affects subunit D at the bottom of the staircase, which reaches the maximum opening of its binding pocket in state 4, demonstrating that although ADP release is achieved already in s2, nucleotide exchange evolves over four states (s2→→s5) (Fig. 4d,e and Extended Data Fig. 10g). The acquisition of a new ATP molecule (s4→s5) is then accompanied by a concerted motion of subunits E and F together with the large domain of subunit

D (hereafter called 'converter': F–E–D$^L$) (Fig. 4d,e and Extended Data Fig. 10h). As a part of this motion, the coordination of the newly obtained ATP molecule is restored by the N terminus in subunit D (Extended Data Fig. 10i,j). Consequently, the gate-opening (cluster [E]) and gate-closing (cluster [D]) motions of the RuvB N terminus serve as additional proof for the directionality of the nucleotide cycle. Finally, the retrograde signalling causes subunit D (large domain) to become part of the rigid area in the RuvB motor, which marks the completion of the nucleotide cycle (Fig. 4c–e and Supplementary Video 6).

In summary, our findings establish that the conformations of all RuvB subunits are context-dependent within the hexamer and the converter (F–E–D$^L$) functions as a RuvB motor-operating multi-domain module, which undergoes highly coordinated motions during the nucleotide cycle. The critical position of subunit E in the centre of this module enables the binding of RuvA$^{D3}$ to pass information through inter-subunit signalling to stimulate ATP hydrolysis in distant subunit A and nucleotide exchange in adjacent subunit D (Fig. 4f and Supplementary Video 7).

## Lever mechanism

To gain insight into the linkage of conformational changes observed in the converter of the RuvB motor and DNA translocation, we examined the five structures of the nucleotide cycle (s1 to s5) by aligning all states to the centre of the converter (subunit E). The analysis revealed that the sequential movement follows a trajectory that translates into a lifting motion of the RuvB motor, in which the individual areas of the hexamer lift proportionally to their distance from subunit E (Figs. 3b and 5a). This causes the DNA-binding interface together with its bound DNA to be lifted by around 7.0 Å away from the RuvA–HJ core. Thus, our data provide evidence that RuvB motors act as molecular levers, which convert the energy obtained throughout the nucleotide cycle into a pulling force to physically move the DNA by approximately 7.0 Å—that is, two nucleotides—and thereby achieve branch migration during DNA recombination (Fig. 5b–d and Supplementary Videos 8 and 9).

Notably, the subsequent cluster switch only repositions the RuvB hexamer (walking along the DNA substrate) after the nucleotide cycle, but does not exert a direct mechanical force onto the DNA and thus does not actively contribute to strand exchange (branch migration) in the RuvA–HJ core[48] (Fig. 5d and Supplementary Video 10). The largest conformational changes in the converter of the RuvB motor are initiated with the recruitment of the second RuvA[D3] (s1→s2), accompanied by the nucleotide exchange reaction of ADP ejection (s1→s2) and ATP uptake (s4→s5), indicating that these two events contribute the most to DNA translocation (Extended Data Fig. 6e,f). Consistently, motions that are associated with nucleotide exchange have recently also been proposed as a force-generating step in the AAA+ ATPase motor of the 26S proteasome[49,50]. On the basis of our findings, we posit that RuvA functions as a fulcrum, which enables RuvB motors to facilitate branch migration by producing a power stroke that pulls the DNA through the RuvA core (Fig. 5c). In summary, the RuvB AAA+ ATPase motor undergoes two consecutive processes (nucleotide cycle and cluster switch) that account for both the maintenance of the unaltered structure of the DNA and the need for its rotation during branch migration.

## Time-resolved cryo-EM

In the course of the structural analysis of the t2 dataset we found two additional subsets of particles that exhibit a nucleotide occupancy, which does not line up with the sequential nucleotide cycle described above. The first subpopulation contains particles that lack the centrally localized RuvA oligomer ($s0^{-A}$) (Extended Data Fig. 2e). These clearly show that the four RuvB subunits A to D are occupied by ATPγS and subunits E and F are occupied by ADP (Extended Data Fig. 7a,b). Notably, specific densities are visible at low density thresholds, indicating the partial presence of ATPγS and $Mg^{2+}$, thus determining that an asymmetrically formed RuvB hexamer can carry up to five ATP molecules (Extended Data Fig. 7a,b). The particles of the other subpopulation (s0, RuvA bound) were found to have the same nucleotide configuration as the RuvA-deficient particles ($s0^{-A}$) (Extended Data Figs. 2c,e and 7a,b). Because ATP hydrolysis (s2→s3) precedes the acquisition of a new ATP molecule (s4→s5) in the nucleotide cycle, the simultaneous presence of ATP in subunits A and D suggests that state s0 is not part of the hydrolytic cycle. Moreover, we also noticed that the converter in state s0 assumes a hybrid conformation, which is different from any of the conformations seen in the nucleotide cycle (s1 to s5) (Fig. 5e and Extended Data Fig. 6b). Therefore, we hypothesized that such a state resembles a RuvB hexamer that has not entered the nucleotide cycle yet and therefore must first undergo ATP hydrolysis or exchange to adopt the position- and conformation-dependent sequence of the nucleotide arrangement as displayed throughout the states s1 to s5. We refer to such a state as the 'initiation state' (s0).

To test this hypothesis, we performed cryo-EM on RuvAB–HJ particles under the same conditions but vitrified the sample shortly after in vitro reconstitution (at 30 min (t1 dataset) instead of 5 h (t2)) (Extended Data Fig. 1h). Only two states ($s0_{t1}$ and $s1_{t1}$) could be recovered at high resolution (3.3 Å) from this dataset (Extended Data Figs. 2d and 3i–n and Extended Data Table 1). In both t1 states, only a single RuvA[D3] binds subunit D in the RuvB hexamer (Extended Data Fig. 2d,e), implying that states s2 to s5 observed after a 5 h incubation (t2) are indeed actively generated by a progressing nucleotide cycle. In addition, the finding confirms that the RuvAB–HJ complexes (t2) were vitrified in the process of active branch migration. At the structural level, state $s0_{t1}$ is similar to s0 (t2) (Fig. 5e and Extended Data Fig. 10k), yet it contains a fifth ATP molecule in subunit F. This finding corroborates the notion that state s0 (t2) can eventually be generated from $s0_{t1}$ through ATP hydrolysis in subunit F (non-processive). Given that ATP levels typically exceed those of ADP in bacterial cells[51], it appears likely that in vivo RuvB motors first assemble initiation states by preferentially loading ATP stochastically at RuvB subunits (s0 with four or five ATPs), to then enter the processive sequential nucleotide cycle (s0→s1→→s5) to promote branch migration.

## An integrated model for branch migration

Our results lead us to propose a model for initiation and processive branch migration that postulates that DNA translocations occur through a lever mechanism executed and controlled by the RuvA-tethered RuvB hexamer combined with DNA rotation[34] (Fig. 5f).

Non-processive initiation phase (stochastic): (1) RuvA tetramers bind to the Holliday junction and their flexible RuvA[D3] recruit RuvB subunits to assemble as hexamers arranged in a spiral staircase around the newly formed DNA and in opposite orientations on each side of the RuvA-bound Holliday junction (tripartite RuvAB–HJ complex). (2) The RuvB hexamers are stochastically loaded with nucleotides (ATP or ADP) and initial out-of-register ATP hydrolysis and/or nucleotide exchange take place to adopt a sequential nucleotide arrangement such as represented by state s1 (A–B–C–D–E–F: ATP-ATP-ATP-ADP-ADP-ADP).

Processive translocation phase (sequential): (1) The hexameric RuvB motor works as a unit and undergoes a forward and retrograde signalling wave mediated by the converter and fuelled by the nucleotide cycle: at first ADP is ejected at the bottom of the staircase in subunit D, causing ATP hydrolysis in subunit A at the top of the staircase, followed by ATP uptake in subunit D. (2) Because RuvB is anchored to domain III of RuvA during the nucleotide cycle, rotation of RuvB is accompanied by a pulling of the DNA out of the RuvA core, advancing branch migration by two nucleotides (the power stroke). (3) Following the nucleotide cycle, the RuvB motor is repositioned (cluster switch), whereby RuvB subunits will adopt the conformation of their adjacent neighbours. (4) After the cluster switch and completion of the rotation, RuvA[D3] must dissociate owing to physical constraints of the tether and is free to rebind the next advancing RuvB subunits. The motor is now reset by keeping the conformational clusters [E] and [D] confined within reach of RuvA. To go through a full rotation of 360°, the process is repeated six times. Each subunit will go through at least five position-specific conformations and the branch migration complex consumes in total 12 ATP molecules (6 ATP molecules per RuvB motor) and advances the recombined DNA by 12 nucleotides.

## Discussion

This work reveals the critical role of substrate-disengaged RuvB subunits, whose highly coordinated motions control the nucleotide cycle in the RuvB hexamer. These subunits are part of a converter through which the binding of RuvA[D3] to subunit E can stimulate long-range inter-subunit signalling and which leads to ATP hydrolysis and nucleotide exchange. Substrate-disengaged subunits are a unifying feature across most ring-forming AAA+ motors[20,37,39], suggesting that variations of the converter probably also operate other AAA+ ATPases. To be able to repeatedly exert their critical function on a rotating RuvB motor,

RuvA[D3] domains need to constantly release from the RuvB hexamer and bind to newly generated binding interfaces that are produced by the nucleotide cycle. Although the driving force behind this rotation remains to be identified, it seems plausible that the energy for this motion is derived from the nucleotide cycle. As the DNA substrate already refolds into a double helix within the confinement of the double-tetrameric RuvA core, we propose that the RuvB motor rotation is powered by the rewinding of the translocating DNA. In this view of the RuvAB machinery, the double RuvA tetramer serves an important function in stabilizing the Holliday junction, ensuring that the two DNA substrates can rewind into a double helix and providing a rationale for the rotation of RuvB motors.

With five distinctive transition-state intermediates (s1 to s5), our data establish structurally that in RuvB motors, the nucleotide cycle progresses around the ring, providing proof of concept for a conserved core mechanistic principle in hexameric AAA+ ATPase translocases[37]. In the context of the RuvAB complex, the sequential nucleotide cycle of the rotating RuvB motor causes the converter to be maintained in the same area with respect to the central RuvA–HJ complex. As a result, a single RuvA tetramer is probably sufficient to control the nucleotide cycle of both RuvB motors. However, in other hexameric AAA+ ATPase motors, sequential ATP hydrolysis events should consequently cause the corresponding substrate-disengaged subunits to progress around the ring. To operate the nucleotide cycle in these motors, putative converter interactors must therefore be able to reach every subunit of the AAA+ ATPase motor. This may provide a rationale for the embedding of ring-shaped AAA+ ATPase motors within multimeric scaffolds, such as in the proteasome or ClpA/X-P[50,52,53]. Alternatively, the regulatory function of RuvA may instead be carried out directly by the substrate.

Further, we show that the nucleotide cycle is a spatiotemporal continuum of conformational changes through which RuvB AAA+ ATPase motors convert the chemical energy retained in ATP to a lever action. The RuvA[D3]-bound subunits in the converter are at the heart of this process, as their physical connection to the RuvA core complex generates the fulcrum that is needed to turn the RuvB motor into a molecular lever. Notably, while the DNA is levered, it remains associated with its binding interface; our data thus enable us to decompose the lever action (sequential steps during the nucleotide cycle) from the cluster switch (following the nucleotide cycle). This reveals that the nucleotide cycle serves to promote DNA pulling, while also priming the RuvB hexamer for a cluster switch. This priming event, which is not part of the nucleotide cycle itself, is critical for enabling the propagation of the nucleotide cycle around the ring and, consequently, for continuous DNA translocation (Fig. 5f and Supplementary Video 11). Notably, hexameric AAA+ ATPases specific for nucleic acid as well as protein translocation share a conserved asymmetric spiral organization around their cognate substrates and are furthermore believed to share a similar translocation rate per hydrolysed ATP molecule[20,38,54]. Similarly, the pulling of DNA, RNA and protein substrates is thought to be powered by a common sequential nucleotide cycle[21,39,44,49,50]. On the basis of their shared geometrical and mechanistic properties, our findings suggest that the majority of ring-shaped AAA+ ATPase translocases may function as molecular levers that efficiently convert a concerted wave of conformational changes associated with their nucleotide cycles into a defined lift-height of their central pores, as a common basic mechanism to facilitate substrate translocation.

Finally, our findings reveal that RuvB motors are most variable in the converter, which changes from a hybrid conformation in the initiation states (s0 and s0[tl]) to the spatiotemporal continuum observed in the nucleotide cycle (s1–s5). As a functional DNA damage response is essential for intracellular bacterial pathogens to cope with the oxidative environment inside our cells, state-specific targeting of the converter may provide a promising avenue for the inhibition of RuvB motors—and thus homologous recombination—by small molecule interference.

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

## Methods

### Protein engineering, expression and purification

RuvA from *S. typhimurium* was fused to a C-terminal tetra-histidine tag and cloned into pET-52b(+) expression vector (Novagen), using the NcoI and SacI restriction sites. Recombinant protein expression was performed in *E. coli* strain BL21(DE3). Bacterial cells were grown at 37 °C in LB medium supplemented with 100 µg ml$^{-1}$ ampicillin to an absorbance at 600 nm of about 0.6. Expression of RuvA was induced by the addition of 1 mM isopropyl β-D-1-thiogalactopyranoside (IPTG) and cultures were further incubated at 37 °C for 3 h. Cells were then pelleted at 4,250$g$ for 10 min at 4 °C, washed in 20 mM NaCl, 1 mM EDTA, 20 mM Tris-HCl pH 8 buffer (buffer 1), resuspended in 100 mM NaCl, 5% glycerol, 100 mM Tris-HCl pH 8 buffer (buffer 2) and stored at −80 °C. For protein purification, the cell suspension was thawed, supplemented with a complete protease inhibitor cocktail (Sigma Aldrich), lysed by sonication and the resulting cell lysate was cleared by centrifugation (Beckman JA-25.50, 17,500 rpm, 1 h, 4 °C). The supernatant was applied onto a 5 ml HisTrap column (GE Healthcare) equilibrated with buffer 2 and immobilized proteins were recovered by gradient elution using buffer 2 supplemented with 500 mM imidazole. Peak fractions were pooled, dialysed against buffer 2 and loaded onto a Superdex 200 10/300GL size exclusion column (GE Healthcare) equilibrated in 100 mM NaCl, 1 mM EDTA, 0.5 mM DTT, 5% glycerol, 100 mM Tris-HCl pH 8 buffer (buffer 3). The peak fraction containing RuvA was collected, and aliquots were frozen in liquid nitrogen and stored at −80 °C. N-terminally truncated RuvB (16-333) from *S. thermophilus* was C-terminally fused to a tobacco etch virus (TEV) protease cleavage site, followed by a linker and a HA tag, and cloned into the pProEX HTB expression vector (Thermo Fisher Scientific), using the NcoI and HindIII restriction sites. Protein expression and purification were performed as described for RuvA from *S. typhimurium*. The TEV cleavage was performed during the dialysis step. The purity of recombinant RuvA and RuvB proteins was assessed by SDS–PAGE, followed by staining with Coomassie R-250 and was estimated to be higher than 95% (Extended Data Fig. 1a,b, Supplementary information Table 3).

### DNA substrates

Holliday junctions with mobile (HJ-X26)[55] and immobile (HJ-Y2Ap, modified from Y2A[17]) cores were prepared by annealing synthetic oligonucleotides (Sigma Aldrich) provided in Supplementary information Table 3, following a previously published protocol[56]. In brief, the oligonucleotides were purified by native 6% PAGE (TAE buffer) and mixed in appropriate ratios in annealing buffer (buffer 4) (25 mM NaCl, 10 mM Tris-HCl pH 8). The annealing reaction was performed in a 0.2 ml tube and covered with a thin layer of mineral oil to prevent water evaporation. The mixture was heated to 95 °C for 10 min, and the temperature was subsequently decrease in 10 °C temperature steps every 10 min. To obtain homogenous four-way Holliday junction preparations, the annealing reaction was supplemented with a DNA sample buffer (New England Biolabs) and separated by native 6% PAGE (TAE buffer). Bands corresponding to four-way Holliday junctions were cut out from the gel and eluted by incubation in 5 mM Tris-HCl pH 8. For DNA-binding assays (electro mobility shift assay (EMSA)), one oligonucleotide strand was labelled with radioactive $^{32}$P (3,000 Ci mmol$^{-1}$) at the 5′ end prior to annealing. For the branch migration activity assays, one oligonucleotide strand was fluorescently labelled with ATTO 647N.

### RuvAB–HJ in vitro reconstitution

RuvAB–HJ particles were reconstituted as described[17], with minor modifications. Purified Holliday junction and RuvA were mixed and supplemented with 5 mM MgCl$_2$. The mixture was incubated at 37 °C for 30 min and applied to size exclusion chromatography on a Superdex 200 10/300GL column equilibrated with 100 mM NaCl and 5 mM MgCl$_2$, 5 mM Tris-HCl pH 8 buffer (buffer 5). The peak fraction containing RuvA–HJ complexes was mixed with purified RuvB in the presence of 10 mM MgCl$_2$ and an equimolar ratio of ATPγS and ADP (1 mM). To form RuvAB–HJ complexes, the mixture was incubated at 37 °C for 10 min and then cooled to 4 °C. Prior to vitrification, all samples were analysed for RuvAB–HJ complex formation by negative-stain electron microscopy.

### Branch migration activity assay

Branch migration activity was measured as described[57]. Briefly, the branch migration reaction (20 µl) contained 20 nM of purified and fluorescently labelled synthetic HJ-X26 and varying amounts of purified RuvA and RuvB proteins in buffer 6 (15 mM MgCl$_2$, 1 mM DTT, 50 µg ml$^{-1}$ BSA, 2 mM ATP, Tris-HCl pH 8). Following an incubation at 37 °C for the indicated time, RuvA and RuvB proteins were digested by proteinase K treatment (2 mg ml$^{-1}$) and 0.5% SDS at 37 °C for 10 min. Glycerol was added to the reaction (30% final concentration) and branch migration was assayed by electrophoresis (135 V for 35 min, TAE buffer) in a 6% polyacrylamide gel. Bands corresponding to Holliday junction and Holliday junction derivatives were visualized by ChemoStar Touch ECL and fluorescence images (INTAS Science Imaging).

### Electro mobility shift gel assay

Varying amounts of purified RuvA protein were incubated with 5′-$^{32}$P-labelled synthetic Holliday junction (HJ-Y2Ap) for 30 min at 37 °C in 5 mM EDTA, 1 mM DTT, 100 µg ml$^{-1}$ BSA, 30 mM Tris-HCl 8 buffer (buffer 7). DNA sample buffer (New England Biolabs) was added to the reaction and the complex formation was assayed by electrophoresis in a 6% polyacrylamide gel (1× TAE). Electrophoresis was carried out at 4 °C at 150 V for 1.5 h in a 0.5× TAE buffer. Gels were dried, and DNA bands were visualized by autoradiography.

### Grid preparation for cryo-EM

Amorphous carbon (1–1.5 nm) was deposited (Leica ACE60 carbon coater) on freshly cut mica sheets and baked for 0.5 h at 120 °C. Quantifoil grids were cleaned by dipping into chloroform for 60 s and dried for 30 min. Continuous carbon grids were made by floating always freshly prepared amorphous carbon on a water surface onto cleaned and strongly glow discharged (3 min at 25 mA) Quantifoil grids. Grids were dried for 1 h followed by 30 min of baking at 120 °C and stored under controlled vacuum for maximum 2 weeks.

### Negative-staining electron microscopy

Before sample application, grids were positively glow discharged for 30 s at 25 mA using a GloQube Plus Glow Discharge System (Electron Microscopy Sciences). Four microlitres of freshly prepared RuvAB–HJ complexes were applied to carbon-coated copper grids and incubated for 30 s. The sample was blotted off, and then stained with 4 µl of the staining solution (2% uranyl acetate) for 30 s. Excess stain was blotted off and the grids were air-dried for at least 2 min. Grids were imaged using a Thermo Fisher Scientific Talos L120C TEM with a 4K Ceta CEMOS camera.

### Cryo-EM sample preparation and data collection

Freshly in vitro reconstituted RuvAB–HJ complexes were incubated on ice for 30 min (dataset t1) or approximately 5 h (dataset t2) prior to vitrification. *N*-Dodecyl-β-maltoside (DDM) was added to a concentration of ~0.005% prior to application of the protein sample to the grid. Four microlitres of the final RuvAB–HJ sample was applied twice onto glow discharged (30 s, 25 mA) gold Quantifoil grids (2/2 300 mesh), containing a thin layer (1–1.5 nm) of amorphous carbon (made in-house). In brief, after the first sample application at 4 °C for 1 min in a horizontal position, the liquid was blotted off from the side. The procedure was repeated, and the sample was plunge-frozen into a propane:ethane (63:37) mixture using a Vitrobot Mark V (Thermo Fisher Scientific) set to 100% humidity and 4 °C. Blotting times ranged from 4–7 s. Vitrified samples were imaged on a Thermo Fisher Scientific Titan Krios TEM

operating at 300 kV, equipped with a field emission gun (XFEG) and a Gatan Bioquantum energy filter with a slit of 10 eV and a Gatan K3 electron detector. During data acquisition, the slit was re-centred every 6 h. For the t1 dataset, a total of 10,057 micrographs were recorded in electron-counting mode at ×81,000 nominal magnification (1.1 Å per pixel at the specimen level) consisting of 33 frames over 3 s (total electron exposure of of 53 e⁻ Å⁻², corresponding to 1.6 e⁻ Å⁻² per frame) using Thermo Fisher Scientific EPU data collection software. The defocus range was set between −0.3 and 3 μm. For the t2 dataset, 30,053 micrographs at ×130,000 nominal magnification (1.09 Å per pixel at the specimen level) consisting of 20 or 25 frames, respectively, were recorded with a Gatan K2 Summit direct electron detector operated in electron-counting mode and Gatan energy filter with slit of 10 eV. The accumulated electron exposure was 30.7 e⁻ Å⁻² (corresponding to 1.24 or 1.55 e⁻ Å⁻² per frame) during a 5 s exposure at − 0.3 to 4 μm defocus range (Extended Data Table 1).

### Cryo-EM image processing and atomic model building

Single-particle analyses were performed using Relion (v3.0b and v3.1)[58,59]. Micrograph frames (movies) were motion-corrected using MOTIONCOR2 (implemented in Relion)[60], dose-weighted (using 1.24 or 1.55 e⁻ Å⁻² per frame for t2 and 1.55 e⁻ Å⁻² per frame for t1) and the contrast transfer function (CTF) parameters were estimated with CTFFIND4 (v4.1.14)[61]. Particles were automatically picked from the motion-corrected micrographs either using CrYOLO (v1.4)[62], Gautomatch (v0.56)[63] or Relion Autopick trained with a subset of manually picked particles. In the t1 dataset, approximately four million coordinates were picked. Particle images were extracted with a box size of 80 pixels (bin = 4) and subjected to multiple rounds of 2D classifications. Only particles present in homogeneous classes were kept, amounting to 948,812 particles (after duplicate removal). Focused classifications were performed by re-extracting particles with a box size of 360 pixels, centred around the RuvB rings (1,881,624 particles) and the central RuvA–HJ (948,812 particles) part. Subsequently, three rounds of refinement, per-particle CTF and Bayesian polishing were performed. Additionally, for the RuvA–HJ reconstruction, signals emerging for the RuvB rings were subtracted. For the t2 dataset, approximately 9 million coordinates were used for particle extraction, which were subsequently subjected to 4 times binning and multiple rounds of 2D classifications, leading to a total of 1,786,669 particles. From these, three groups of particles were identified, and three particle subsets were generated: (1) tripartite RuvAB–HJ particles (717,780) containing two RuvB motors, (2) bipartite RuvAB–HJ particles containing one RuvB motor (549,364 particles), and (3) RuvB–HJ complexes lacking RuvA (519,525 particles). For the reconstruction of the tripartite RuvA–RuvB–HJ complex, only particles from group 1 were used. At first, an ab initio model was created in Relion using a smaller subset of particles (n = 50,000). Subsequent classifications and refinements led to a consensus reconstruction yielding a resolution of ~8 Å. Particles from group 2 were used to reconstruct the bipartite RuvAB–HJ structure (~3.9 Å). Particles from the group 1 after subtraction of the signal corresponding to one RuvB motor were used to generate pseudo-bipartite particles. Focused reconstruction procedures were performed as described for the t1 dataset, which resulted in 3D reconstructions of the RuvB motor and the central RuvA–HJ subcomplexes, respectively. The RuvA–HJ subcomplex was reconstructed using particles from the combined particle stack (groups 1 and 2). For the RuvB structures, a total of approximately 2.3 million RuvB motors were extracted (from all three groups), centred, 3D classified, and subsets were independently refined. Subsequently, per-particle CTF, Bayesian polishing, and 3D refinements were performed twice. Applying this procedure resulted in 9 distinctive RuvB motor structures, ranging from 2.9 to 4.1 Å in resolution. Local resolution estimates, gold-standard resolution (Fourier shell correlation = 0.143) and sharpened maps (B-factor

range: 30–80 per focused refinements) and multibody refinements were calculated using Relion 3.1[64].

Model building started by generating homology models for RuvA and RuvB with SWISS-MODEL[65]. For RuvA, Protein Data Bank (PDB) entry 1BVS served as a structural template, and PDB entry 1HQC[66] served as a reference model for RuvB. Models were fitted into electron microscopy maps using the fit-in-map tool in UCSF Chimera (v1.13)[67]. Initial model refinements were performed with Rosetta (v3.12)[68] controlled via StarMap v.1.1.12[69]. Further interactive refinement was carried out in ISOLDE (v1.1.2)[70], a molecular dynamics-guided structure refinement tool within UCSF ChimeraX (v1.2.5)[71]. Finally, the resulting coordinate files were refined with Phenix.real_space_refine (v1.19.1-4122)[72] using reference model restraints, strict rotamer matching and disabled grid search settings. MolProbity server[73], EMringer[74] (via phenix) and Z-score were used to validate model geometries and model-to-map fits (Extended Data Fig. 3e–m, Extended Data Table 1).

### Visualization and analysis

UCSF Chimera (1.13), ChimeraX (v1.1 and v1.2.5) and PyMOL (2.4.1) were used for visualizations and analysis. For the dihedral angle analysis following residues were used: (1) large ATPase: residues 36, 73, 80, 174, 55, 155, 170, 94 and 121; (2) small ATPase: residues 249, 227, 209 and 196; (3) head: residues 282, 284, 265, 306 and 263. For the triangle angle analysis, the centre of mass determined with following residues: (1) large ATPase: residues 20–180; (2) small ATPase: residues 181–256, (3) head: residues 257–325. The variance analysis was performed over the distances of each Cα atom in all models to their corresponding centroids (models were aligned to RuvB subunit C).

### Reporting summary

Further information on research design is available in the Nature Research Reporting Summary linked to this article.

### Data availability

Cryo-EM density maps resolved in this study have been deposited in the Electron Microscopy Data Bank (EMDB) (www.emdataresource.org) under accession codes: EMD-13294, EMD-13295, EMD-13296, EMD-13297, EMD-13298, EMD-13299, EMD-13300, EMD-13301, EMD-13302, EMD-13303, EMD-13304, EMD-13305, EMD-15085 and EMD-15126. The corresponding coordinates have been deposited in the Protein Data Bank (PDB) (https://www.pdb.org) under accession codes: 7PBL, 7PBM, 7PBN, 7PBO, 7PBP, 7PBQ, 7PBR, 7PBS, 7PBT and 7PBU. Uncropped versions of all gels and blots are provided in Supplementary Fig. 2. All other data are available from the corresponding authors upon reasonable request.

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

**Acknowledgements** We thank all current and former members of the Marlovits laboratory for their support in this project; S. C. West for critical suggestions on the initial design of the study and for kindly providing purified RuvA and RuvB proteins (*E. coli*), which served as positive controls in our early experiments; H. Kotisch and T. Heuser for their support during initial cryo-EM sample preparation trials; M. Strauss for helpful suggestions to improve cryo-EM vitrification; F. DiMaio for help with Rosetta (StarMap); T. I. Croll for help with ISOLDE; K. Karius for help with the initial model of the Holliday junction; and H. Sondermann and R. Hennell James for critical input to the manuscript. Cryo-EM optimizations and initial data collection were performed at the VBC electron microscopy facility (Vienna, Austria). Final cryo-EM data collections were performed at the Cryo-EM Facility at CSSB Hamburg (supported by the University of Hamburg, the University Medical Center Hamburg–Eppendorf and DFG grant numbers INST152/772-1, 152/774-1, 152/775-1, 152/776-1 and 152/777-1 FUGG.).

High-performance computing (HPC) was possible through access to the HPC at DESY/Hamburg (Germany) and Vienna Scientific Cluster (Austria). This project was supported by funds available to T.C.M. through the Behörde für Wissenschaft, Forschung und Gleichstellung of the city of Hamburg at the Institute of Structural and Systems Biology at the University Medical Center Hamburg–Eppendorf (UKE), Deutsches Elektronen Synchrotron (DESY), the Institute for Molecular Biotechnology (IMBA) of the Austrian Academy of Sciences, and the Research Institute of Molecular Pathology (IMP). L.C. was supported by the Austrian Science Fund (FWF) (I 2408-B22 granted to T.C.M).

**Author contributions** T.C.M. is the lead corresponding author on this Article. J.W. designed the experiments, generated constructs, expressed and purified proteins, performed biochemical assays, assembled complexes, performed negative-stain electron microscopy screenings, optimized vitrification and collected and processed all cryo-EM data. D.F., N.G.-M. and J.W. built atomic models. J.W., D.F. and T.C.M. interpreted and conceptualized the data, prepared figures and videos and wrote manuscript. W.L. implemented and adjusted scripts for HPC. J.W., D.F., T.C.M., L.C. and O.V. performed data visualization. All authors read, corrected and approved the manuscript. T.C.M. conceived the project, received funding and supervised the project.

**Funding** Open access funding provided by Universitätsklinikum Hamburg-Eppendorf (UKE).

**Competing interests** Authors declare no competing interests.

**Additional information**
**Correspondence and requests for materials** should be addressed to Jiri Wald, Dirk Fahrenkamp or Thomas C. Marlovits.

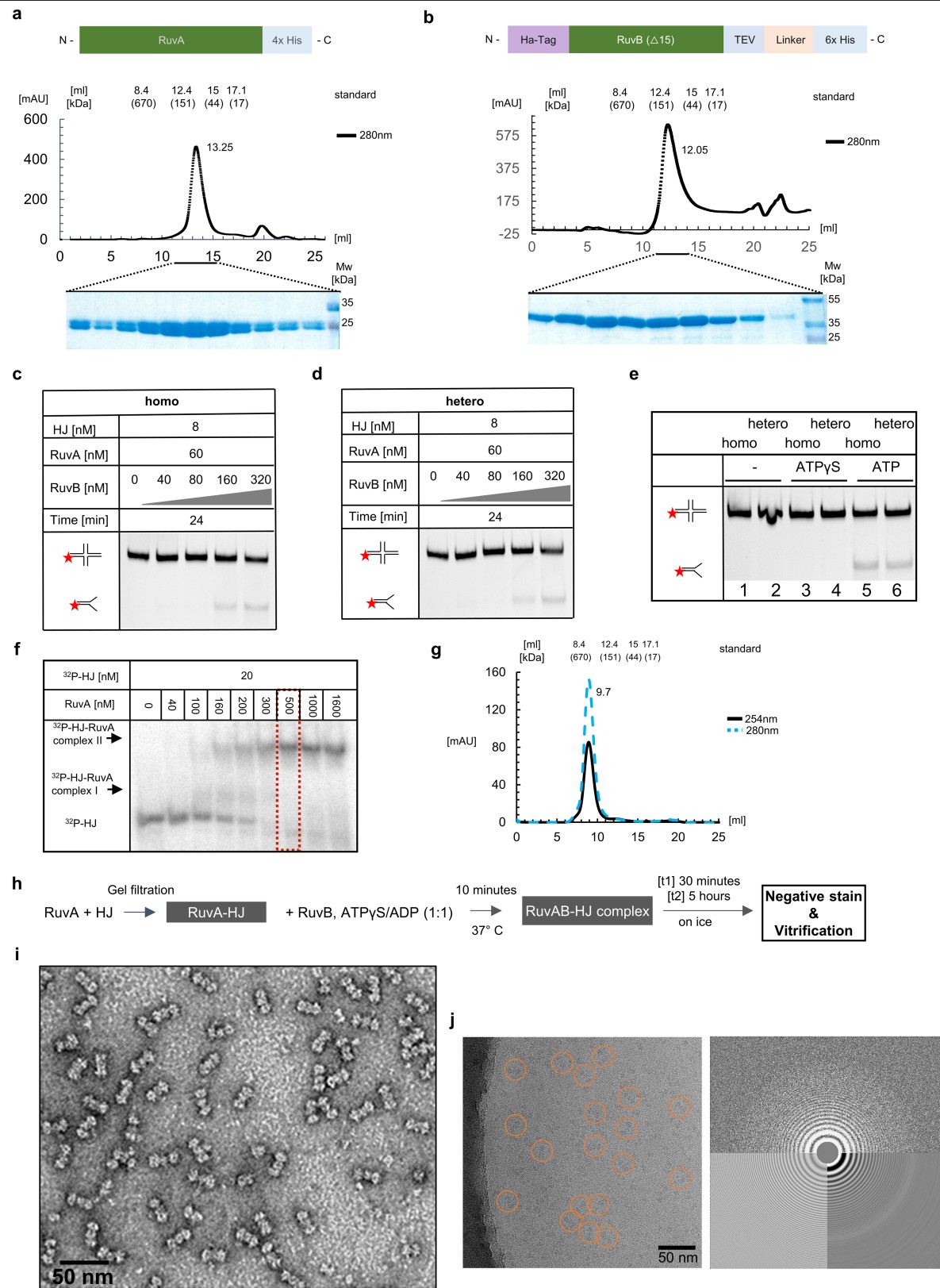

**Extended Data Fig. 1** | See next page for caption.

**Extended Data Fig. 1 | Protein purification and in vitro reconstitution of the RuvAB–HJ complex.** a, b, Domain organization and purification of the RuvA (*S. typhimurium*) (a) and RuvB (*S. thermophilus*) (b) Gel filtration and analysis of individual fractions by SDS-PAGE and Coomassie staining. RuvA elutes at a peak at 13.25 ml corresponding to 92 kDa, likely representing a tetrameric species (4 x 24 = 96 kDa). Similarly, RuvB elutes at 12.05 ml corresponding to a molecular weight species of 153 kDa representing a tetrameric complex in solution (4 x 37 kDa =148 kDa). Molecular weights were estimated based on the retention times of a gel filtration standard. The experiments were repeated at least ten times with similar results. c, d, Comparison of the HJ branch migration activities using recombinant *S. typhimurium* RuvA and either recombinant *S. typhimurium* RuvB (c) or recombinant *S. thermophilus* RuvB (d). Branch migration of both complexes was assessed with increasing RuvB concentrations. Branch migration assay was performed using fluorescently labeled (red star) HJs X26 at 37 °C. The HJ contains a 26-base pair homologous core with heterologous sequences in the shoulders to impair spontaneous branch migration. The experiments were repeated three times with similar results. e, Direct comparison of the HJ branch migration of homo- and hetero-complexes in absence of nucleotides (lanes 1-2), presence of ATPγS (lanes 3-4) or ATP (lanes 5-6). Branch migration assay was performed with 8 nM fluorescently labeled (red star) HJs X26, 60 nM RuvA and 320 nM RuvB incubated at 37 °C for 16 min. The experiment was repeated three times with similar results. f, Electrophoretic mobility shift assay (EMSA) using constant amounts of $^{32}$P-labeled HJ (Y2AP) and increasing amounts of recombinant RuvA to confirm its HJ binding capacity. Saturation of binding to complex II (two tetramers bound to one HJ) was obtained at RuvA-concentrations >500 nM and subsequently used for in vitro reconstitution experiments. At lower RuvA concentrations (<200 nm) complex I is observed, representing one RuvA tetramer bound to one HJ. The experiment was repeated three times with similar results. g, Gel filtration profile of the in vitro reconstituted RuvA–HJ complex, which elutes at a peak at 9.7 ml corresponding to a molecular weight of ~500 kDa, indicating a homogeneous population of double tetramer bound Holliday junction particles. The experiment was repeated at least ten times with similar results. h, Schematic of the in vitro reconstitution strategy applied in this study. i, Electron microscopy analysis of negatively stained RuvAB–HJ branch migration hetero-complex. j, Electron microscopy analysis of vitrified RuvAB–HJ branch migration hetero-complex and corresponding power-spectrum for determination of underfocus and astigmatism using CTFFIND4. The experiment was repeated at least ten times with similar results.

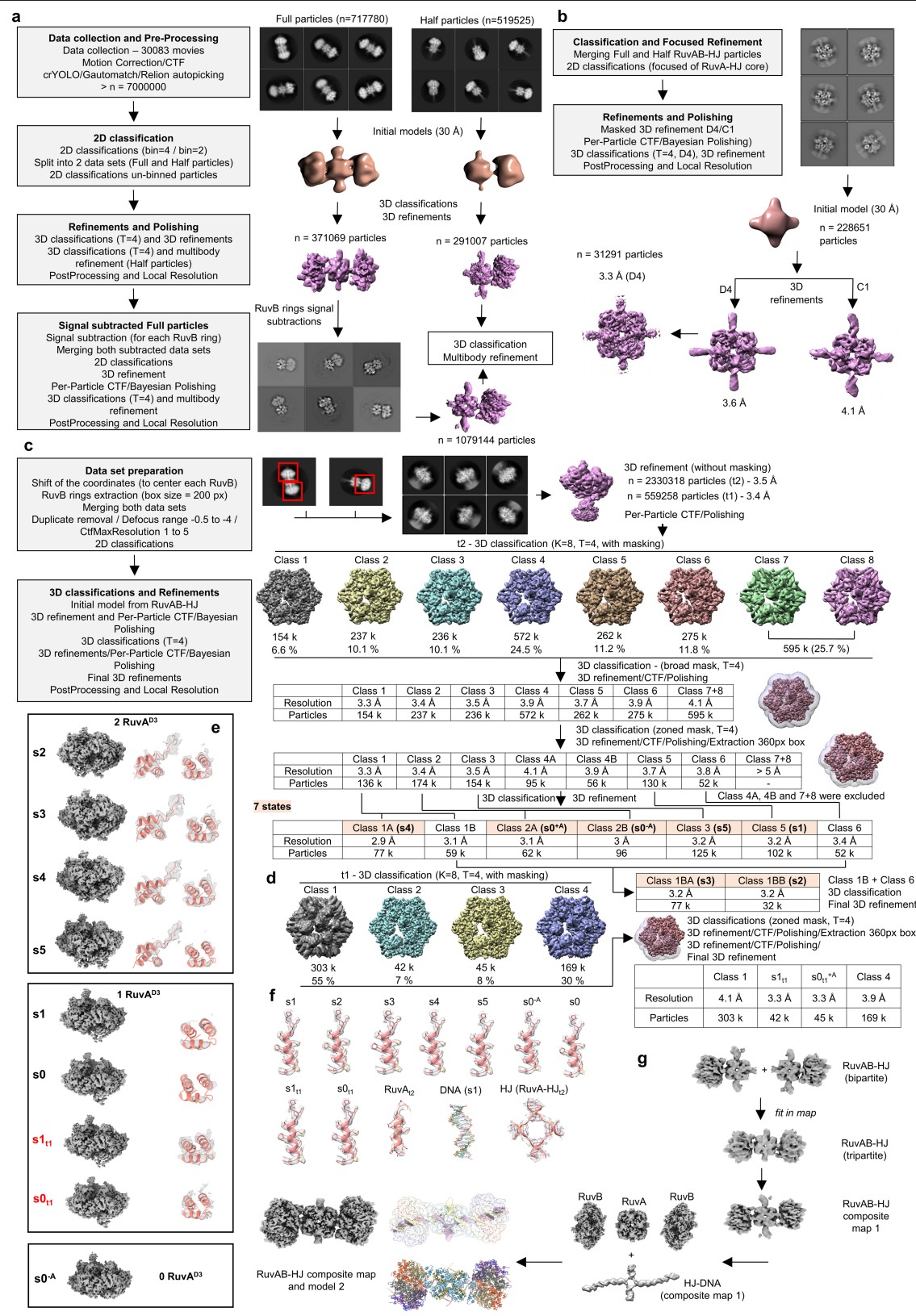

**Extended Data Fig. 2** | See next page for caption.

**Extended Data Fig. 2 | Cryo-EM data collections and single particle processing.** a, Workflow for single particle processing of RuvAB–HJ complexes. b–d, RuvA-HJ core complexes and RuvB-HJ complexes (dataset t2 in c, and dataset t1 in d). e, Final post-processed RuvB maps and densities corresponding to RuvA$^{D3}$, which were extracted for each RuvB map. f, Representative example cryo-EM density and built models from different states of RuvB (residues 64-81), RuvA (65-80), DNA within RuvB and the resolved HJ within RuvA. g, Assembly strategy of the low- and high-resolution composite maps. 3D-reconstructed pseudo bipartite particles (tripartite particles after signal subtraction of a single RuvB motor) (top) were rigid body-fitted into the consensus reconstruction of the tripartite RuvAB–HJ particle (middle) using the *fit in map* tool in ChimeraX. Individual maps were then combined into one composite map 1 (bottom), using the *volume add* tool in ChimeraX. To generate the high-resolution composite map, four components were used: the cryo-EM density corresponding to the HJ in composite map 1, two focus-refined RuvB motors (s2) and the focus-refined RuvA-HJ complex. The four components were fitted into composite map 1 and then combined as described before.

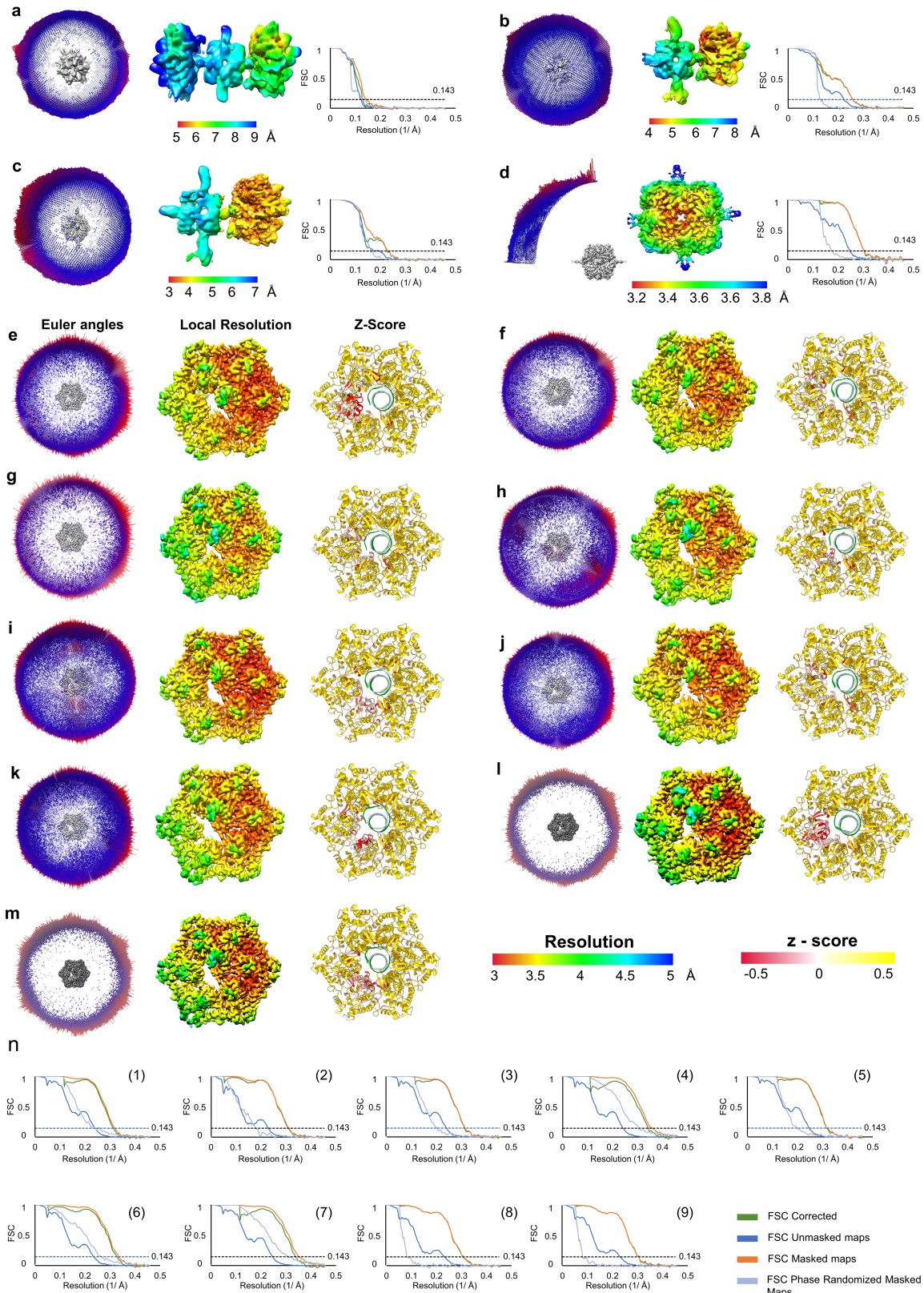

**Extended Data Fig. 3 | Single particle reconstructions.** a–d, Angular distribution plots, local resolution estimations and Fourier Shell Correlation (FSC) plots of the C1 reconstruction of the tripartite RuvAB–HJ complex (a), the pseudo-bipartite RuvAB–HJ complex (tripartite particles after signal subtractions of a single RuvB motor [either "left" or "right" and subsequent merging of the dataset]) (b), the bipartite RuvAB–HJ complex (c) and the RuvA-HJ core complex from the t2 dataset (d). e–m, Angular distribution plots, local resolution estimations and Z-Scores of the RuvB motor states: s1 (e), s2 (f), s3 (g), s4 (h), s5 (i), s0 (j), s0$^{-A}$ (k), s1$_{t1}$ (l) and s0$_{t1}$ (m). n, Fourier Shell Correlation (FSC) plots of the RuvB motor reconstructions: s1 (1), s2 (2), s3 (3), s4 (4), s5 (5), s0 (6), s0$^{-A}$ (7), s1$_{t1}$ (8) and s0$_{t1}$ (9).

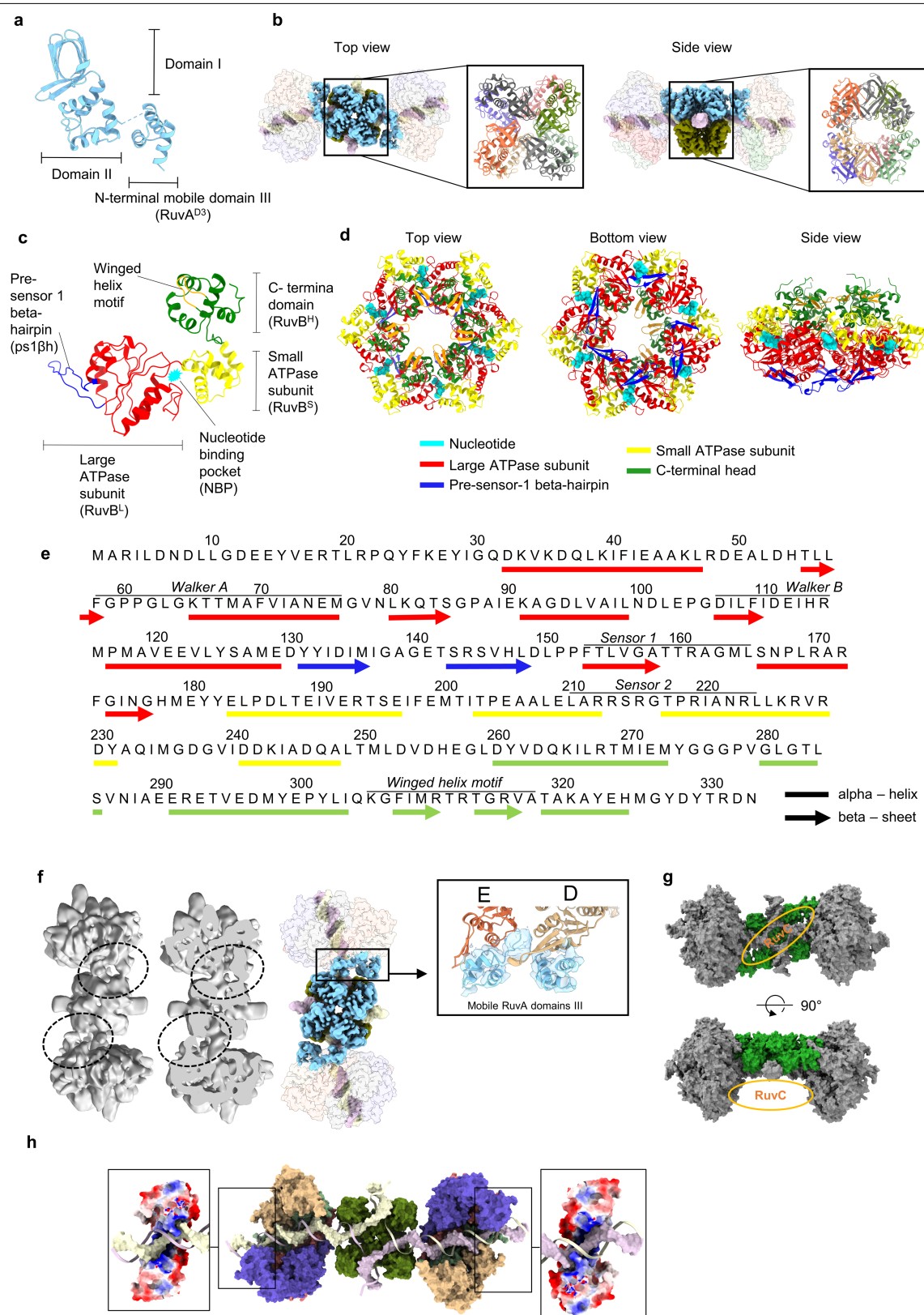

**e**

M A R I L D N D L L G D E E Y V E R T L R P Q Y F K E Y I G Q D K V K D Q L K I F I E A A K L R D E A L D H T L L

Walker A

F G P P G L G K T T M A F V I A N E M G V N L K Q T S G P A I E K A G D L V A I L N D L E P G D I L F I D E I H R

Walker B

M P M A V E E V L Y S A M E D Y Y I D I M I G A G E T S R S V H L D L P P F T L V G A T T R A G M L S N P L R A R

Sensor 1

F G I N G H M E Y Y E L P D L T E I V E R T S E I F E M T I T P E A A L E L A R R S R G T P R I A N R L L K R V R

Sensor 2

D Y A Q I M G D G V I D D K I A D Q A L T M L D V D H E G L D Y V D Q K I L R T M I E M Y G G G P V G L G T L

S V N I A E E R E T V E D M Y E P Y L I Q K G F I M R T R T G R V A T A K A Y E H M G Y D Y T R D N

Winged helix motif

— alpha – helix

→ beta – sheet

**Extended Data Fig. 4** | See next page for caption.

**Extended Data Fig. 4 | Structure of RuvA and RuvB subunits and their oligomeric organisation within the RuvAB–HJ branch migration complex.**
a–c, RuvA consists of three domains. Domains I and II are responsible for binding the HJ and the oligomerisation into tetramers, whereas domain III (N-terminal) extends from the RuvA core (domain I, and II) and binds to the RuvB motor. b, Top and side view of the double-tetrameric organization of the RuvA within the context of the fully assembled RuvAB–HJ complex. Magnifications highlight the individual RuvA subunits constituting the central core by different colours. c, RuvB consists of three domains. A large (RuvB$^L$) and a small (RuvB$^S$) ATPase subdomain, together forming the ATP-binding domain, and a C-terminal "head" domain (RuvB$^H$) binding the DNA substrate via a winged-helix motif. The presensor-1 β-hairpin of RuvB, a distinguishing feature of the PS1 insert superclade, is part of the RuvB$^L$ but shown in blue colour. d, Hexameric assembly of the RuvB motor, using the domain colour code in c. Akin to other hexameric AAA+ ATPase translocases, the nucleotide binding pocket is located between adjacent subunits to enable nucleotide-dependent inter-subunit signalling. e, RuvB (*S. typhimurium*) amino acid sequence with the visualization of secondary structure elements. f, 3D reconstruction of the entire tripartite RuvAB–HJ (low resolution), cut-away view and the composite map. The four RuvA$^{D3}$ domains localize to the same side of the HJ crossover. The magnification highlights the two adjacently located binding interfaces of RuvA$^{D3}$ to subunit D, E of RuvB. g, Putative location of RuvC dimer within the the RuvABC–HJ resolvasome. One RuvA tetramer (in green) binds the HJ crossover and operates both RuvB motors simultaneously. The second RuvA tetramer is replaced by a dimer of the RuvC resolvase. h, Spiral staircase organization of the DNA binding interface in RuvB motors. Charge distribution representation of the RuvB$^H$ domains in the RuvB motor staircase. Together, the RuvB$^H$ domains of RuvB subunits A, B, C, D form a positively charged pit to stably accommodate one strand of the double-stranded DNA substrate. One strand of each maternal DNA substrate (pink/yellow) is processed by one RuvB motor.

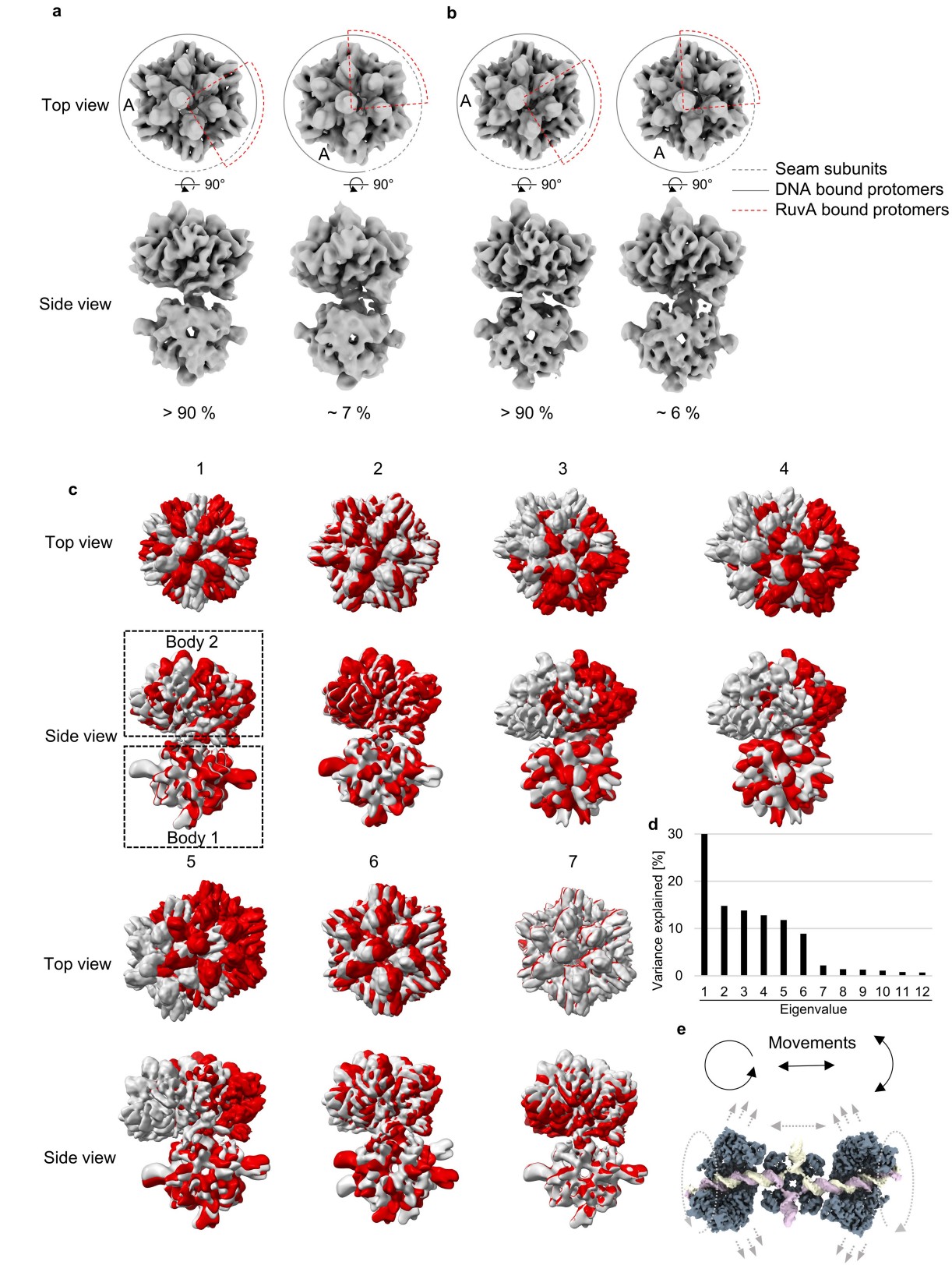

**Extended Data Fig. 5** | See next page for caption.

**Extended Data Fig. 5 | Rotation of the RuvB motor.** a, 3D-classification of the bipartite RuvAB–HJ particles. Repeated classifications revealed that in approx. ~7 % of the particles the RuvB motor is rotated by 60° with respect to the RuvA-HJ core complex. b, 3D-classification of the tripartite (pseudo-bipartite) RuvAB–HJ particles. Repeated classifications revealed that in approx. ~6 % of the particles the RuvB motor is rotated by 60° with respect to the RuvA-HJ core complex. c, Multibody refinement of RuvAB–HJ particles. Maps corresponding to the seven most abundant eigenvectors after principal component analysis are shown in top and side views. Repositioning of the reconstructed body densities along the individual eigenvectors (grey and red colour correspond to the start and end point of the movement). Body 1 and 2 indicate body definitions for multibody refinement in RELION. d, Contribution of each of the twelve eigenvectors to the overall variance (in %). The first 7 eigenvectors cover ~95 % of all movements. The rotational motions of eigenvectors 1 and 2, together, cover ~45 % of the variance. A wobbling motion of the RuvB motor with respect to the RuvA core is represented by eigenvectors 3-6 and amount to ~47.5 % of the variance. The motion increasing the gap size (bouncing) between the two bodies (eigenvector 7) covers ~2.5 % of the observed variance. e, Illustration of the directionalities corresponding to the three predominant trajectories: rotation (45 %), wobbling (47.5 %) and bouncing (2.5 %).

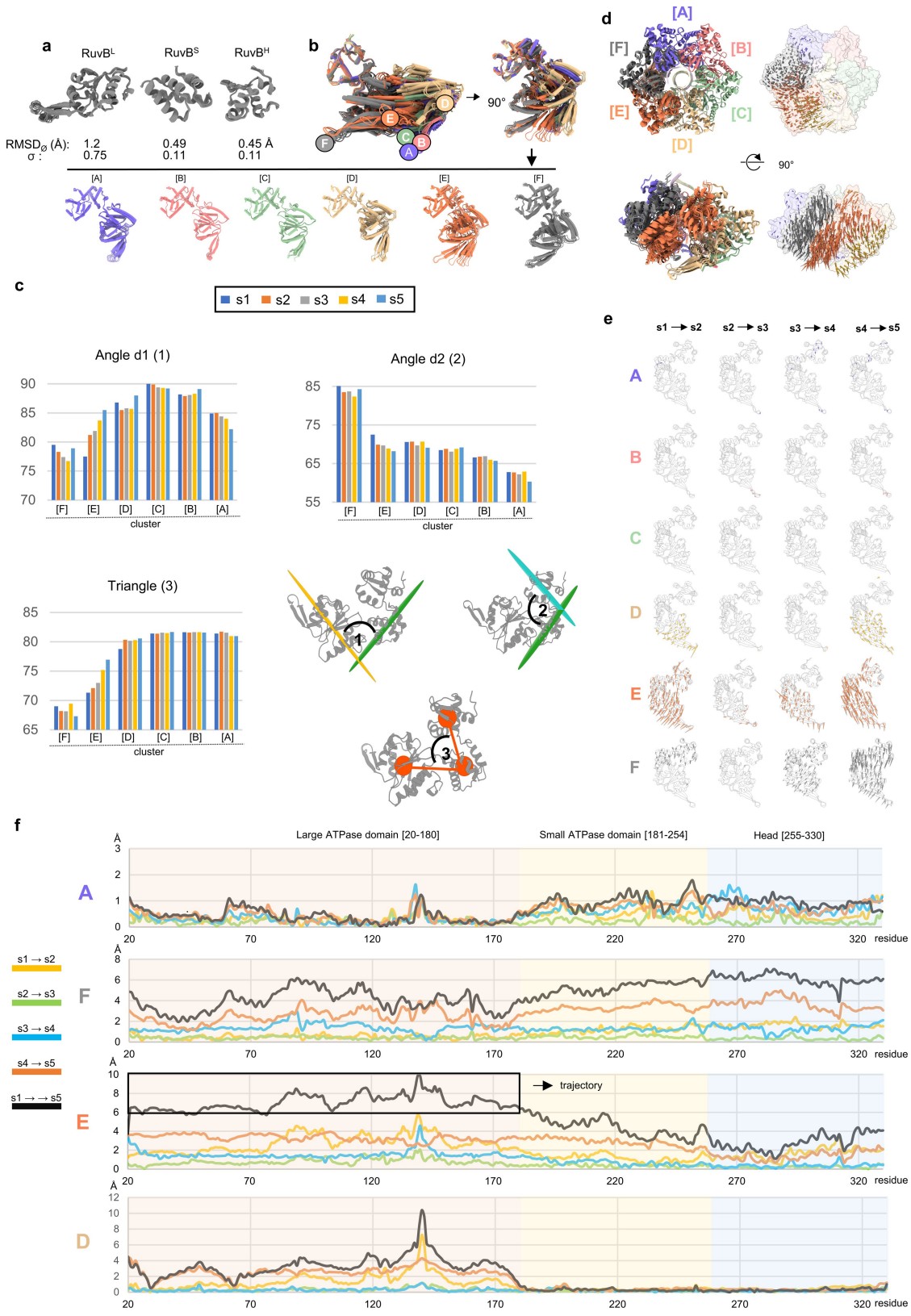

**Extended Data Fig. 6** | See next page for caption.

**Extended Data Fig. 6 | Structural analysis of RuvB domains, subunits and hexamers.** a, Superposition of RuvB$^L$, RuvB$^S$ and RuvB$^H$ domains. Domains belong to the RuvB subunits of the RuvB motor states obtained from the t2 dataset (s1-s5, s0 and s0$^{-A}$). Low RMSD$_\emptyset$ together with σ values demonstrate that RuvB domains move as rigid bodies in the nucleotide cycle of the RuvB motor. Note, the higher RMSD$_\emptyset$ value for RuvB$^L$ is mainly due to the flexibility of the presensor-1 β-hairpin. When excluded from the analysis, the RMSD$_\emptyset$ drops to 0.57 with σ = 0.25. b, Structural comparison of RuvB subunits by aligning on RuvB$^H$ domain. The analysis reveals that RuvB subunits form conformational clusters, which also reflect their position in the RuvB hexamer. Colours indicate conformational clusters [A]-[F]. Notably, subunit E of the initiation states s0 and s0$^{-A}$ groups into cluster [F] and not in [E], highlighting the hybrid conformation of the converter in these states. c, Angle measurements in RuvB subunits from the nucleotide cycle s1-s5 (t2 dataset). Two dihedral angles (d1 between RuvB$^L$ and RuvB$^S$ and d2 between RuvB$^S$ and RuvB$^H$) and one triangle angle (between the centre of masses of RuvB$^L$:RuvB$^S$:RuvB$^H$) are plotted. When reading from the left to right (cluster [F] to [A]), the plotted angular changes correspond to the conformational changes of a RuvB subunit, progressing through the nucleotide cycle of the RuvB motor. Bar colours correspond to the nucleotide cycle states. d, Motion analysis of the RuvB hexamers highlighting the movements of the converter (*F:E:D$^L$*) during the nucleotide cycle. Top and side views of atomic models of RuvB s1-s5 (hexamers aligned to invariable RuvB subunit C). Arrows indicate the magnitude (distance in Å) and directionality of the motion between matching C$_\alpha$-atom pairs [residues 19-330]. For improved visibility a cut-off of 1 Å was chosen and arrows are shown only for C$_\alpha$-atom pairs corresponding to every second residue. To aid visualization the length of each arrow is 2.5 times the length of the measured distance. e, Deconvolution of the motion analysis in (d) of all individual subunits. Arrows indicate the magnitude (distance in Å) and directionality of the motion between matching C$_\alpha$-atom pairs. For improved visibility a cut-off of 1 Å was chosen and arrows are shown only for C$_\alpha$-atom pairs corresponding to every second residue. To aid visualization the length of each arrow is 2.5 times the length of the measured distance. f, Plotted distances [residues 20-330] (in Å) of matching C$_\alpha$-atom pairs in the RuvB subunits A, F, E and D, respectively, measured based on the superposition shown in (d). The analysis reveals that RuvB subunits E and F and RuvB$^L$ (subunit D) as part of the converter (*F:E:D$^L$*) are highly flexible (Fig. 2e, Extended Data Fig. 6e). The box indicates the trajectory observed for RuvB subunit E within the large ATPase domain. In subunit A, the overall motions are smaller (<3 Å) and largely restricted to RuvB$^S$ and RuvB$^H$ domains (Fig. 2e, Extended Data Fig. 6e).

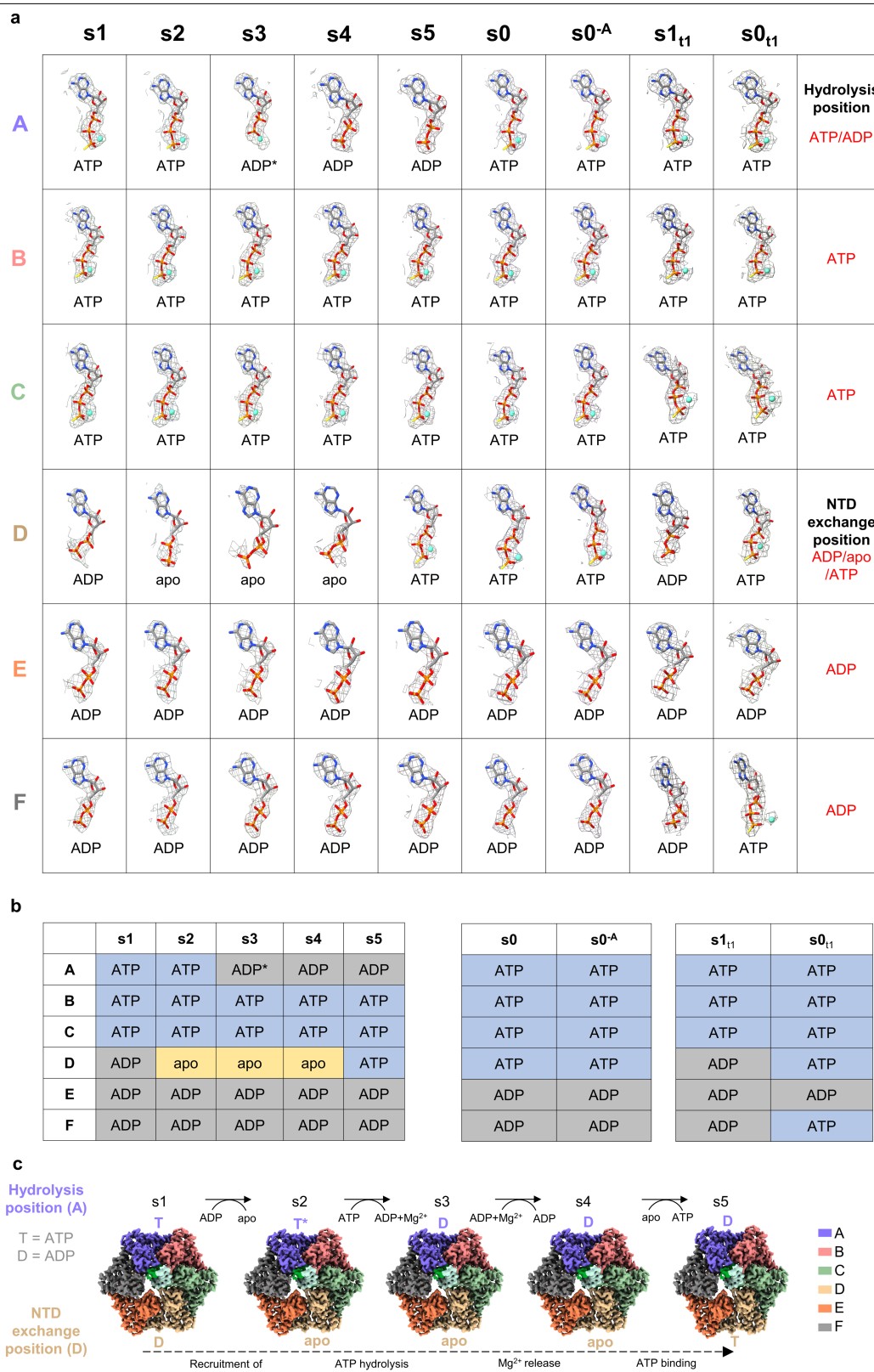

**Extended Data Fig. 7 | Coexisting nucleotides in different RuvB motor states.** a, cryo-EM densities within the nucleotide binding pocket and modeled nucleotides in all states obtained from the t2 (s1, s2, s3, s4, s5, s0, s0⁻ᴬ) and t1 (s1ₜ₁ and s0ₜ₁) data set and built nucleotide models. Note that labels shown as 'ATP' refer to ATPγS, which has been used for structure determination. 'apo' labels refer to discontinuous densities within the nucleotide binding pockets. 'ADP*' notation refers to [ADP + Mg²⁺]. All cryo-EM densities are shown at the same isosurface (within each dataset) threshold. b, Table listing the nucleotides in the RuvB nucleotide binding pockets according to the RuvB subunit/cluster and the RuvB motor state. The table exemplifies that the nucleotide cycle (s1-s5) starts and completes with 3 ATP and 3 ADP bound ligands (state s1: ATP bound in subunits A, B, C; state 5: ATP bound in subunits B, C, D). c, Linear representation of the nucleotide cycle (s1-s5), visualizing its progression through the RuvB hexamer. To simplify the process, the rotation of the RuvB motor has been neglected.

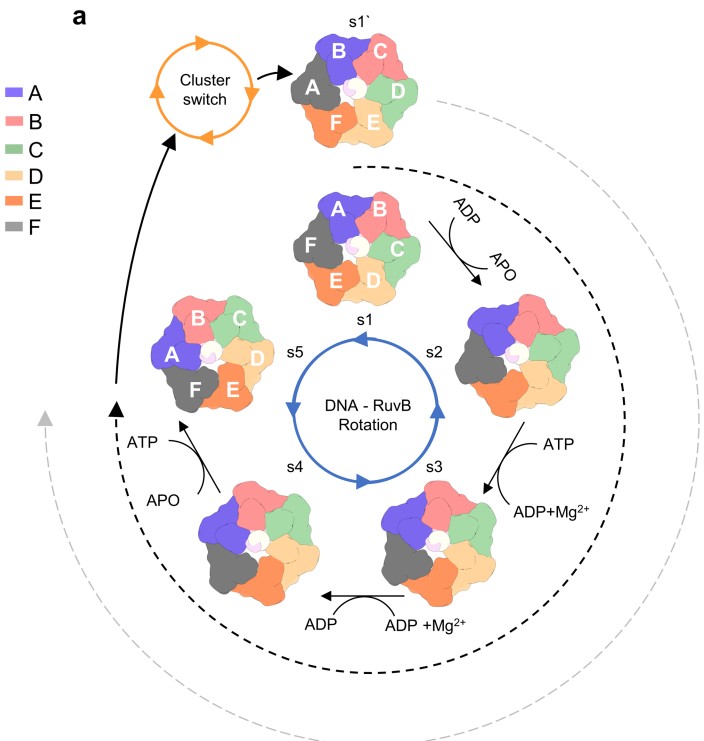

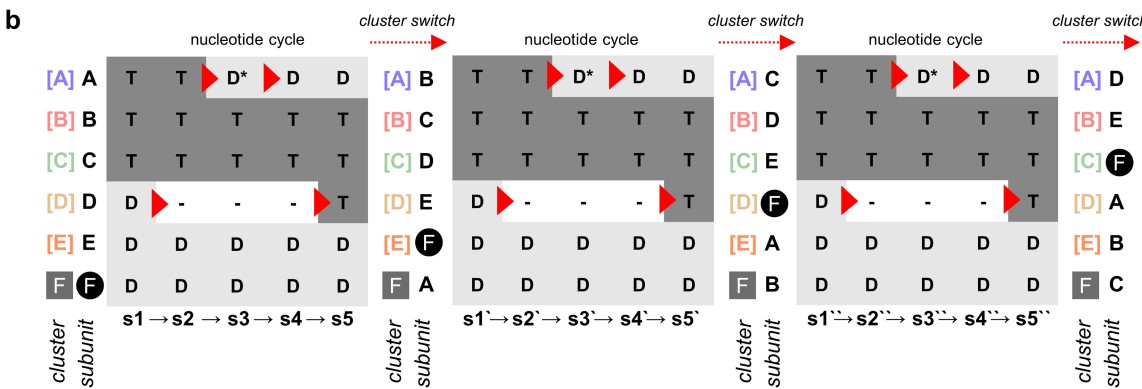

**Extended Data Fig. 8 | Nucleotide cycle, conformational clusters and subunit positions.** a, Location of conformational clusters and subunits of RuvB during one nucleotide cycle. The six subunits (A-F) within the RuvB hexamer adopt similar, yet different conformations (conformational clusters) throughout the nucleotide cycle s1-s5 (colours indicate a specific conformational cluster within the hexamer). The hexamer rotates such that after one nucleotide cycle the position of subunits has changed (approx. 60˚; for example subunit B in s5 is now located at position of subunit A in s1). In order to prepare for the next round of the nucleotide cycle, the hexameric motor is reset (s5>s1'), by obtaining the conformation represented by s1 but shifted by one subunit (cluster switch; for example subunit B in s5 will change its conformation to become the conformation of subunit A in s1). This repositioning process keeps the conformation of subunits confined and within reach for RuvA^D3, necessary to tether the hexamer to the RuvA core, and can be described as 'walking of the RuvB motor along the DNA'. Of note, repeated cluster switches could be the mechanistic basis for the previously described helicase activity of isolated RuvB motors (Video 10). b, Nucleotide identity and membership of conformational cluster of specific subunits during three consecutive nucleotide cycles and cluster switches. For example, subunit F stays within its conformational cluster (i.e. [F]) during the first nucleotide cycle, and changes its conformation to become member of cluster [E] in the second, and cluster [D] in the third nucleotide cycle.

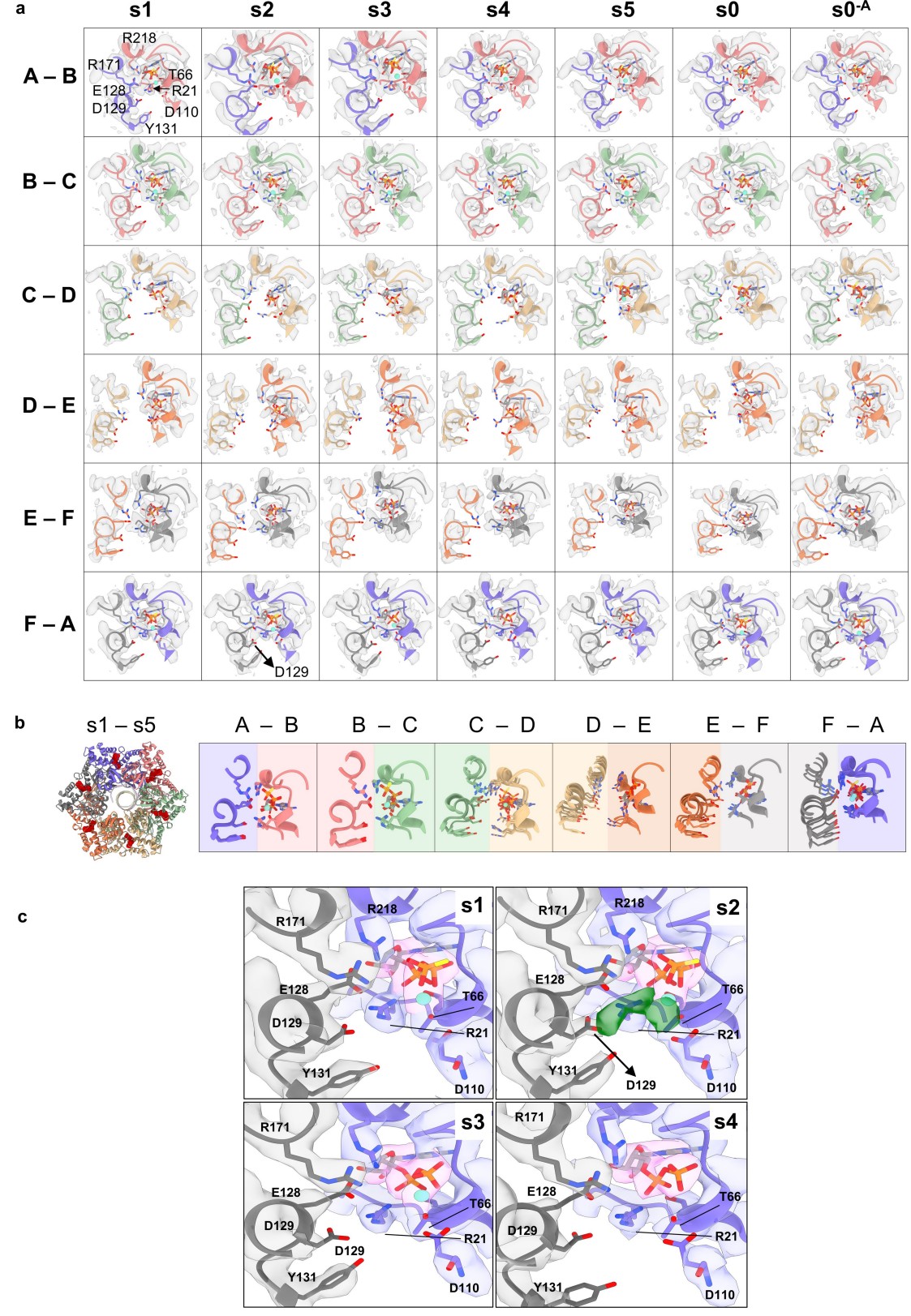

**Extended Data Fig. 9** | See next page for caption.

**Extended Data Fig. 9 | Opening and closing motions of the RuvB nucleotide binding pocket at the inter-subunit interface during the nucleotide cycle.** a, cryo-EM density (grey) and corresponding atomic models (cartoons) of inter-subunit interfaces of the nucleotide binding pockets in RuvB. The cryo-EM densities have been contoured at the same threshold level (0.026). Residues contributing to ATP-binding and ATP hydrolysis are shown in stick representation. b, Superpositions of the RuvB nucleotide binding pocket inter-subunit interfaces according to their clusters [A]-[F], using the respective nucleotide-bound *cis*-subunit from state s1 as an alignment reference to illustrate the movement of both subunits contributing to the interface across the five nucleotide cycle states (s1-s5). The interfaces between clusters [A] and [B] and clusters [B] and [C] are almost invariant. The interface between cluster [F] and [A] highlights the motion which triggers ATP hydrolysis in cluster [A]. c, Magnified nucleotide binding pocket in RuvB subunit A together with the *cis*-residues from subunit F. Magnification highlights unmodeled cryo-EM density (green density) in state s2, which likely corresponds to ordered water molecules initiating the nucleophilic attack on the ATP γ-phosphate in the course of the ATP hydrolysis reaction. Note, that residue D129 is stabilized (and fully covered by the EM density) in state s2, but not in others.

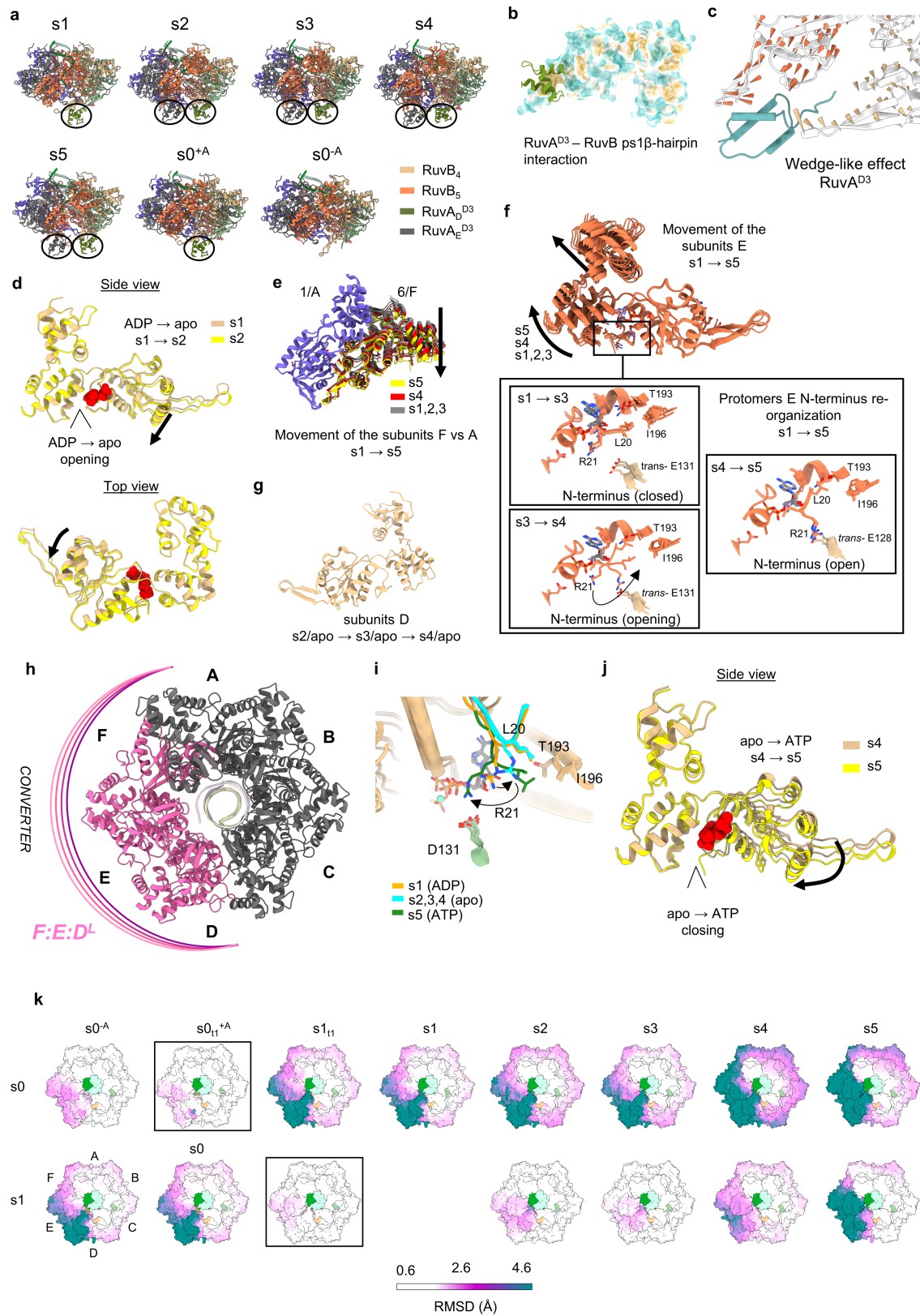

**Extended Data Fig. 10** | See next page for caption.

**Extended Data Fig. 10 | Conformational changes in the converter.** a, Side-by-side comparison of the atomic models corresponding to the RuvB motors states in the nucleotide cycle (s1-s5) and the initiation states (s0$^{-A}$, s0) obtained from dataset t2. b, A hydrophobic interaction of RuvA$^{D3}$ is established with α-helix α3 and the presensor-1 β hairpin of RuvB subunit D or E. Surface representation of RuvB with hydrophilic amino acids shown in turquoise and hydrophobic residues shown in sepia. A cartoon model of only one RuvA$^{D3}$ is shown. c, Motion analysis of RuvB subunits D and E focused on the RuvA$^{D3}$ binding interface highlighting the wedge-like effect. Arrows indicate the magnitude and directionality of the motion between matching C$_α$-atom pairs. d, Domain rearrangements associated with nucleotide exchange in cluster [D] in the transition from state s1 to s2 (ADP→apo). Nucleotides are shown in surface representation and highlighted in red. To visualise the motions, RuvB subunits were superimposed on the head domain of subunit D. e, Unidirectional motion of subunits F with respect to RuvB subunits A during the nucleotide cycle. The largest motion occurs in the transition from state s4 (yellow) to s5 (red), when the ATP hydrolysis reaction is completed and the Mg$^{2+}$ has dissociated from the ADP in the nucleotide binding pocket of RuvB subunit A. f, Opening motion of the RuvB subunit E N-terminus during the progression of the nucleotide cycle. Note that the opening motion is mainly visible in cluster [E]. g, Superposition analysis of the nucleotide exchange facilitating RuvB$_D$ subunits from states s2, s3 and s4. A low average RMSD$_∅$ of 0.3 Å reveals that subunits RuvB D remain almost invariable during the three APO states.

h, Position of the converter in the RuvB motor. The converter consists of RuvB subunits E and F together with the large ATPase domain of subunit D (all shown in pink). The converter connects the ATP-hydrolysing nucleotide binding pocket of RuvB subunit A with the nucleotide-exchanging nucleotide binding pocket of RuvB subunit D. Lines indicate the downwards-directed motion of the converter during the nucleotide cycle. i, Closing motion of the RuvB subunit D N-terminus during the progression of the nucleotide cycle. Note that the closing motion is associated with the acquisition of a new ATP molecule and can therefore only be observed in subunits D. Since the opening and closing of the RuvB N-terminus take place over three RuvB conformational clusters ([F], [E] and [D]), these motions occur over three translocation steps/nucleotide cycles. j, Domain rearrangements associated with nucleotide exchange in cluster [D] in the transition from state s4 to s5 (apo→ATP). Nucleotides are shown in surface representation and highlighted in red. To visualize the motions, RuvB subunits were superimposed on the head domain. k, Structural plasticity between all RuvB motor states obtained in this study (t1 and t2 dataset). States were aligned to the DNA. Colours indicate the RMSD (in Å). In the top panel, initiation state s0 served as a reference. The two most similar states are the states s0$_{t1}$ (boxed) and the RuvA$^{D3}$-free state s0$^{-A}$, obtained from RuvB-HJ particles. In the lower panel, nucleotide cycle state s1 served as a reference. The most similar state is s1$_{t1}$ (boxed). In both cases, the comparison with states s2 to s5 highlights that those motions of RuvB subunits are largest restricted to the converter.

Extended Data Table 1 | Cryo-EM sample vitrification, data collection, single particle analysis processing summary and model building summary

Cryo-EM data collection, refinement and validation statistics

| | RuvB s1$_{t2}$ (EMD-13294) (PDB 7PBL) | RuvB s2$_{t2}$ (EMD-13295) (PDB 7PBM) | RuvB s3$_{t2}$ (EMD-13296) (PDB 7PBN) | RuvB s4$_{t2}$ (EMD-13297) (PDB 7PBO) | RuvB s5$_{t2}$ (EMD-13298) (PDB 7PBP) | RuvB s0$_{t2}$*A (EMD-13299) (PDB 7PBQ) | RuvB s0$_{t2}$-A (EMD-13300) (PDB 7PBR) | RuvB s1$_{t1}$ (EMD-13302) (PDB 7PBT) | RuvB s0$_{t1}$*A (EMD-13301) (PDB 7PBS) | RuvA$_{t2}$ (EMD-13303) (PDB 7PBU) | RuvAB-HJ tripartite (EMD-15126) | RuvAB-HJ (Half subtracted) (EMD-13305) | RuvAB-HJ (Half original) (EMD-13304) |
|---|---|---|---|---|---|---|---|---|---|---|---|---|---|
| **Data collection and processing** | | | | | | | | | | | | | |
| Magnification | 130000 | 130000 | 130000 | 130000 | 130000 | 130000 | 130000 | 81000 | 81000 | 130000 | 130000 | 130000 | 130000 |
| Voltage (kV) | 300 | 300 | 300 | 300 | 300 | 300 | 300 | 300 | 300 | 300 | 300 | 300 | 300 |
| Electron exposure (e–/Å$^2$) | 30.72 | 30.72 | 30.72 | 30.72 | 30.72 | 30.72 | 30.72 | 53.3 | 53.3 | 30.72 | 30.72 | 30.72 | 30.72 |
| Defocus range (µm) | -0.5 to -3 | -0.5 to -3 | -0.5 to -3 | -0.5 to -3 | -0.5 to -3 | -0.5 to -3 | -0.5 to -3 | -0.5 to -3 | -0.5 to -3 | -0.5 to -3 | -0.5 to -3 | -0.5 to -3 | -0.5 to -3 |
| Pixel size (Å) | 1.09 | 1.09 | 1.09 | 1.09 | 1.09 | 1.09 | 1.09 | 1.1 | 1.1 | 1.09 | 1.09 | 1.09 | 1.09 |
| Symmetry imposed | C1 | C1 | C1 | C1 | C1 | C1 | C1 | C1 | C1 | D4 | C1 | C1 | C1 |
| Initial particle images (no.) | 2330318 | 2330318 | 2330318 | 2330318 | 2330318 | 2330318 | 2330318 | 1881624 | 1881624 | 228651 | 948 812 | 1370000 | 519525 |
| Final particle images (no.) | 102619 | 32612 | 77587 | 77356 | 125425 | 62542 | 96370 | 41209 | 45834 | 31291 | 371069 | 529554 | 291007 |
| Map resolution (Å) | 3.2 | 3.2 | 3.2 | 2.9 | 3.2 | 3.1 | 3 | 3.3 | 3.3 | 3.3 | 8 | 4.4 | 3.9 |
| FSC threshold | 0.143 | 0.143 | 0.143 | 0.143 | 0.143 | 0.143 | 0.143 | 0.143 | 0.143 | 0.143 | 0.143 | 0.143 | 0.143 |
| Map resolution range (Å) | 3 – 4 | 3 – 4 | 3 – 4 | 3 – 4 | 3 – 4 | 3 – 4 | 3 – 4 | 3 – 4 | 3 – 4 | 3 – 4 | 5 – 9 | 4 – 7 | 4 – 7 |
| **Refinement** | | | | | | | | | | | | | |
| Initial model used (PDB code) | 1HQC | RuvB s1$_{t2}$ | RuvB s1$_{t2}$ | RuvB s1$_{t2}$ | RuvB s1$_{t2}$ | RuvB s1$_{t2}$ | RuvB s1$_{t2}$ | RuvB s1$_{t2}$ | RuvB s1$_{t2}$ | 1BVS | - | - | - |
| Model resolution (Å) | | | | | | | | | | | - | - | - |
| FSC threshold | | | | | | | | | | | | | |
| Model resolution range (Å) | | | | | | | | | | | - | - | - |
| Map sharpening B factor (Å$^2$) | -45 | -41 | -50 | -33 | -46 | -35 | -30 | -43 | -47 | -79 | -10 | -20 | -20 |
| **Model composition** | | | | | | | | | | | | | |
| Non-hydrogen atoms | 15883 | 16264 | 16260 | 16259 | 16264 | 15888 | 15531 | 15965 | 15893 | 8342 | | | |
| Protein residues | 1918 | 1968 | 1968 | 1968 | 1968 | 1918 | 1870 | 1918 | 1918 | 1063 | - | - | - |
| Ligands | 30 | 30 | 30 | 30 | 30 | 30 | 30 | 30 | 30 | - | | | |
| B factors (Å$^2$) | | | | | | | | | | | | | |
| Protein | 79.68 | 61.73 | 64.67 | 63.03 | 73.46 | 68.80 | 67.48 | 82.51 | 86.97 | 83.68 | - | - | - |
| Ligand | 67.16 | 57.22 | 57.05 | 57.64 | 58.52 | 62.27 | 60.82 | 72.96 | 74.71 | | | | |
| **R.m.s. deviations** | | | | | | | | | | | | | |
| Bond lengths (Å) | 0.004 | 0.004 | 0.004 | 0.004 | 0.004 | 0.004 | 0.004 | 0.004 | 0.005 | 0.005 | - | - | - |
| Bond angles (°) | 0.695 | 0.715 | 0.726 | 0.739 | 0.726 | 0.717 | 0.717 | 0.705 | 0.712 | 0.781 | | | |
| **Validation** | | | | | | | | | | | | | |
| MolProbity score | 0.88 | 0.83 | 0.83 | 0.83 | 0.89 | 0.88 | 0.88 | 0.81 | 0.91 | 0.66 | | | |
| Clashscore | 1.43 | 1.15 | 1.18 | 1.24 | 1.49 | 1.43 | 1.43 | 1.05 | 1.59 | 0.47 | - | - | - |
| Poor rotamers (%) | 0.06 | 0.06 | 0.06 | 0.12 | 0.12 | 0.19 | 0.13 | 0.19 | 0.25 | 0 | | | |
| **Ramachandran plot** | | | | | | | | | | | | | |
| Favored (%) | 98.84 | 98.92 | 98.97 | 98.97 | 99.03 | 98.79 | 99.19 | 99 | 99.26 | 99.04 | | | |
| Allowed (%) | 1.16 | 1.08 | 1.03 | 1.03 | 0.97 | 1.21 | 0.81 | 1 | 0.74 | 0.96 | - | - | - |
| Disallowed (%) | - | - | - | - | - | - | - | - | - | - | | | |

|---|---|

# Reporting Summary

## Statistics

For all statistical analyses, confirm that the following items are present in the figure legend, table legend, main text, or Methods section.

| n/a | Confirmed | |
|---|---|---|
| ☐ | ☒ | The exact sample size (*n*) for each experimental group/condition, given as a discrete number and unit of measurement |
| ☒ | ☐ | A statement on whether measurements were taken from distinct samples or whether the same sample was measured repeatedly |
| ☒ | ☐ | The statistical test(s) used AND whether they are one- or two-sided *Only common tests should be described solely by name; describe more complex techniques in the Methods section.* |
| ☒ | ☐ | A description of all covariates tested |
| ☒ | ☐ | A description of any assumptions or corrections, such as tests of normality and adjustment for multiple comparisons |
| ☒ | ☐ | A full description of the statistical parameters including central tendency (e.g. means) or other basic estimates (e.g. regression coefficient) AND variation (e.g. standard deviation) or associated estimates of uncertainty (e.g. confidence intervals) |
| ☒ | ☐ | For null hypothesis testing, the test statistic (e.g. *F*, *t*, *r*) with confidence intervals, effect sizes, degrees of freedom and *P* value noted *Give P values as exact values whenever suitable.* |
| ☒ | ☐ | For Bayesian analysis, information on the choice of priors and Markov chain Monte Carlo settings |
| ☒ | ☐ | For hierarchical and complex designs, identification of the appropriate level for tests and full reporting of outcomes |
| ☒ | ☐ | Estimates of effect sizes (e.g. Cohen's *d*, Pearson's *r*), indicating how they were calculated |

*Our web collection on statistics for biologists contains articles on many of the points above.*

## Software and code

Policy information about availability of computer code

| Data collection | CryoEM: Thermo Fisher EPU v1.09 and v2.4, Negative stain EM: Thermo Fisher TIA v4.15 |
|---|---|
| Data analysis | MotionCor2 v1.2.1 and v1.3, CTFfind v4.1.14, crYOLO v1.4, Gautomatch v0.56, RELION2, 3.1-beta and 3.1, Chimera v 1.11 and v1.13.1 and v1.14, ChimeraX v1.1 and 1.2.5, Pymol v2.4.1, Phyre2 (unversioned), Phenix v1.18.2, EMringer (unversioned), ISOLDE v1.0b5 and v1.1.2, MolProbity (unversioned), Phenix.real_space_refine v1.18-6831, Starmap v1.1.12, Rosetta v3.12 |

For manuscripts utilizing custom algorithms or software that are central to the research but not yet described in published literature, software must be made available to editors and reviewers. We strongly encourage code deposition in a community repository (e.g. GitHub). See the Nature Portfolio guidelines for submitting code & software for further information.

## Data

Policy information about availability of data

All manuscripts must include a data availability statement. This statement should provide the following information, where applicable:

- Accession codes, unique identifiers, or web links for publicly available datasets
- A description of any restrictions on data availability
- For clinical datasets or third party data, please ensure that the statement adheres to our policy

Cryo-EM density maps resolved in this study have been deposited in the Electron Microscopy Data Bank (EMDB) (www.emdataresource.org) under accession codes: EMD-13294, EMD-13295, EMD-13296, EMD-13297, EMD-13298, EMD-13299, EMD-13300, EMD-13301, EMD-13302, EMD-13303, EMD-13304, EMD-13305,

## Human research participants

Policy information about studies involving human research participants and Sex and Gender in Research.

| | |
|---|---|
| Reporting on sex and gender | Not relevant to this study |
| Population characteristics | Not relevant to this study |
| Recruitment | Not relevant to this study |
| Ethics oversight | Not relevant to this study |

Note that full information on the approval of the study protocol must also be provided in the manuscript.

# Field-specific reporting

Please select the one below that is the best fit for your research. If you are not sure, read the appropriate sections before making your selection.

☒ Life sciences   ☐ Behavioural & social sciences   ☐ Ecological, evolutionary & environmental sciences

For a reference copy of the document with all sections, see nature.com/documents/nr-reporting-summary-flat.pdf

# Life sciences study design

All studies must disclose on these points even when the disclosure is negative.

| | |
|---|---|
| Sample size | Sample sizes were chosen as a maximum possible while considering practical limitations for data collection and subsequent data processing. The size of the final particle set was determined by the ability to reach resolutions sufficient in 3D reconstructions. |
| Data exclusions | No data was excluded from the analysis. During cryoEM data clustering, good cryoEM images were chosen for further processing based on their achieved resolution and 3D map quality, a standard method for cryoEM high resolution structural determination. |
| Replication | All biochemical and vitrification experiments have been successfully replicated. |
| Randomization | Particles/images were randomly partitioned for resolution and quality assessment. |
| Blinding | Blinding during data collection and analysis is not a commonly applied procedure in cryoEM. |

# Reporting for specific materials, systems and methods

We require information from authors about some types of materials, experimental systems and methods used in many studies. Here, indicate whether each material, system or method listed is relevant to your study. If you are not sure if a list item applies to your research, read the appropriate section before selecting a response.

## Materials & experimental systems

| n/a | Involved in the study |
|---|---|
| ☒ ☐ | Antibodies |
| ☒ ☐ | Eukaryotic cell lines |
| ☒ ☐ | Palaeontology and archaeology |
| ☒ ☐ | Animals and other organisms |
| ☒ ☐ | Clinical data |
| ☒ ☐ | Dual use research of concern |

## Methods

| n/a | Involved in the study |
|---|---|
| ☒ ☐ | ChIP-seq |
| ☒ ☐ | Flow cytometry |
| ☒ ☐ | MRI-based neuroimaging |

