## [Peer Review File · Nature]

Manuscript Title: Mechanism of AAA+ ATPase-mediated RuvAB-Holliday junction branch

migration Reviewer Comments & Author Rebuttals

Reviewer Reports on the Initial Version:

Referees' comments:

Referee #1:

The present work by Wald et al. describes the complete structure of the bacterial RuvAB Holliday Junction (HJ) branch migration system. Using time-resolved cryoEM, the complex – which consists of as many as 12 RuvB motor subunits (arranged into hexameric rings), two RuvA tetramers, and an HJ DNA – is captured in a series of intermediate catalytic states. The reconstructions are used to develop a model which posits that RuvB uses a rotary ATPase cycle to exert a sequence of 'walking' and 'pulling' movements that lead to DNA translocation. The ATP hydrolysis cycle is suggested to produce the pulling force, while a 'cluster switch' in which each subunit shifts over to take the position of its partner (with respect to RuvA) is implicated in walking. The net result is the pulling of the DNA through RuvA to effect strand exchange and branch migration.

Although in one sense the findings corroborate long-standing models for how the RuvAB system works, it should be clear: the work is an absolute tour de force, not just for understanding the mechanisms of HJ branch migration, but for understanding the operation of ring ATPase motors in general. There is no question that Nature is an appropriate forum for publication.

This said, the data and model, as described in the text, are extremely complicated and difficult to follow (borderline impenetrable in places). Part of the inaccessibility and confusion arises from the haphazard order of figure panel callouts. Another is the very complex way in which the data are variously described with respect to states, clusters, walking, rotating, pulling, etc. When you need a glossary to understand the work (and even this doesn't fully achieve the goal), there is a problem. The density of the organization and technical jargon of the writing detracts from the tremendous impact of structural data and their mechanistic implications.

Given this caveat, the following comments are meant to aid in clarifying the work. They should not be taken as weaknesses that might preclude its eventual acceptance by the journal.

Primary comments:

1) There is a question as to whether the changes in ATPase status in the RuvB ring are fully known with respect to their rotational disposition to RuvA. The authors obtain a series of RuvB ATPase states (s1-s5, as well as s0) from reconstructions that were generated using masking and local refinement procedures that isolate the motor region to improve map quality. By comparison, the observation that the RuvB hexamer can rotate 60° with respect to RuvA appears to be based on lower-resolution (non-focused) reconstructions of the entire system. A key question is therefore whether intermediate RuvB rotational substeps with respect to RuvA are seen for each RuvB state (s1-s5) – as implied by Fig. 3d and by movies 7-10 – or whether these substeps are inferred from the observations that there are 1) multiple RuvB catalytic states and 2) two distinct RuvB/RuvA rotational states. If the former, then it needs to be clarified that rotational substeps indeed correlate with specific ATPase states, and that these are observed at a convincing level of resolution. If the latter, then then the actual translocation mechanism may need to be reconsidered.

2) Regardless of whether RuvB rotational substeps are seen with respect to RuvA, there may be a simpler way to explain the action of RuvB that would seem compatible with the data as presented. The key is to focus on subunits A and D. As subunit A goes through a hydrolysis cycle (ATP to ADP-Mg to ADP), subunit D goes through an exchange cycle (ADP to Apo to ATP). This cycle causes the RuvB ring to rotate by 60° with respect to RuvA. (Rotation may occur because DNA is helical and because RuvA acts as a partial impediment to DNA twisting, necessitating that RuvB initially revolve around DNA as it climbs along the duplex, rather than vice versa; however, rotation need not be a prerequisite for translocation *sensu strictu* – see (3) below.) Hydrolysis and release of Mg⁺⁺ in subunit A frees this protomer to undock from its position at the top of the staircase, providing conformational plasticity that concomitantly allows subunit D to undergo conformational changes associated with ADP ejection and ATP binding. This action also leads subunit E (along with its partner, subunit F) to shift ‘downward’ (toward RuvA), where E picks up a new dinucleobase unit to constitute a motor step. Because subunits D and E are anchored to domain 3 of RuvA, the rotation of RuvB is accompanied by a lifting of the DNA out of RuvA, advancing HJ migration by a step of 2 nt, the step size of the RuvB motor. Once the rotation is complete, domain 3 must dissociate from RuvB because its movement is restricted by a tether that connects this region to the rest of RuvA; this element is then free to bind to the newly advanced protomers (one to E, one to F). In this view, which parallels the action of related AAA⁺ translocases, each round of ATP hydrolysis drives a single step of HJ migration of 2.

(NB: If there are biochemical data on RuvB that support or refute a motor step size of 2 per ATP, these should be acknowledged.)

3) A point to consider, which may or may not accord with the data, is whether RuvA is acting like a stator. In this view, the 60° rotated state might correspond to a stable form when RuvB is actively translocating but RuvA is simply restraining further motor rotation. Rpt1-6 of the proteasome and the F0/F1 ATP synthase are exemplar rotary ATPases that have stators which prevent ring rotation. Regardless, the authors should note that the rotation model they are proposing for RuvB with respect to RuvA is analogous to that advanced for the action of ClpA (or ClpX) when tethered to ClpP.

4) It's very important that the transitions in movies 7-10 be distinguished from the actual observed states, s1-s5. Movies that pause at each reconstruction and then morph between these states would be most helpful for showing which frames correspond to reality and which to a morph interpolation. For example, is the state in which both domain 3 elements of RuvA are pulled back from RuvB observed or inferred?

5) Is it possible to reconcile the observed states of RuvB with single molecule data obtained by the Bustamante

lab for a different dsDNA translocase (the phi29 packaging motor), which has shown that ATPase activity occurs in bursts? These data have suggested that there are hydrolysis-dependent micro- and macrosteps for the motor. The system may be too different than RuvB, but perhaps it is worth considering.

Other comments:

1) Ext. Data Fig. 12 is called out before EDF 3, EDF 10 after 12, EDF 11d before 11a, Fig. 2e before 2d – these are just a few examples of out-of-order callouts. Please order all supplementary figures and panels as they are called out in the text.

2) Line 122: "Akin to other AAA+ translocases"

3) Line 126 and elsewhere: Wouldn't it make more sense to label as RuvBE+F, as this is how the figures depict it, rather than RubB5+6?

4) Line 133: Why does this correlation infer a single (i.e., 2-nt) translocation step? Couldn't this be linked to a larger macrostep, or a stall point before resetting?

5) Lines 193-195: The textual description of the movements seem a bit misleading. The way this sentence is phrased makes it seem as if DNA is leaping from one set of subunits to another set, when in reality there is a release of the DNA from the subunit at the top of the staircase as the bottom-most subunit engages it which changes the set of bound subunits from one cluster to another.

6) Line 242: Do the authors really mean 1 Å? This doesn't seem like a very significant distance, given the resolution of the data.

5) EDF 15: Why do the columns for panels a and b have different labels? Is $s1t1$ the same as $s5^*(t1)$? Also, it would be clearer for this figure and Fig. 3c to have the ADP-Mg states highlighted in the table, as opposed to just ADP.

Referee #2:

In this manuscript, Wald et al. present cryo-EM structures of RuvAB in complex with the Holliday junction (HJ) in several conformational states, allowing them to propose a detailed mechanism for branch migration during DNA recombination. They provide high-resolution information for the assembly of the RuvAB-HJ complex, and most impressively, provide a complete picture of the ATP hydrolysis cycle in the AAA+ motor, RuvB, through extensive cryo-EM processing and structural analysis. This in particular will be of wide interest to the AAA+ ATPase field. They also present an elegant explanation for how RuvA is able to stimulate the ATPase activity of RuvB, by analysing the conformational changes associated with RuvAD3 binding during the ATPase cycle. This manuscript represents an impressive amount of work – the structural analysis is of high quality, and the manuscript is generally well-written.

I do have some specific comments and questions below which should be addressed, none of which are too major:

- In the PDB validation reports for RuvB motors in complex with D3 domains of RuvA, very often it appears that the RuvA chains (usually chains G and H) have up to 100% of residues being flagged as having “poor fit to the EM

map". Do the authors know why this would be? Would it be possible to show some close-up images of the RuvAD3 PDB fitted to the EM density to confirm that the models have been built in correctly? Specifically, this is the case for PDB 7PBQ chain G, 7PBN chains G/H, 7PBO chains G/H, 7PBP chains G/H, 7PBS chain G, 7PBM chains G/H, 7PBT chain G, and 7PBL chain G.

- Line 176, "density that shows remote similarity to nucleotides only at high map thresholds (s2t2-s4t2)": The densities in these 'apo' subunits do not actually look too dissimilar to the ADP density in cluster D s1t2 in these images, although there are indeed discontinuities in the densities. Were all of these NBP panels rendered with equivalent thresholds, or were the thresholds for s2t2-s4t2 lowered to visualise the density here? If these panels are indeed visualised at lower map thresholds (not 'higher') then this should be noted on Ext. Data Fig. 15. Otherwise I would moderate the phrase 'remote similarity'.

- It would greatly help the flow of the paper if the Ext. Data figures appeared in the order they are referred to in the text. Currently, the reader has to jump back and forth through 19 Ext. Data figures, which makes the paper harder to read. In addition, Ext. Data Figs 3, 5, and 6 do not appear to be referred to at all in the text; this should be rectified.

- It should be mentioned somewhere in the introduction (or early in the results) that the RuvA and RuvB proteins used are from *Salmonella typhimurium* and *Streptococcus thermophilus*, respectively. Could the authors comment on why proteins from two different species were used? Intuitively it would make more sense to characterise a complex where both proteins are from the same species.

- It would be useful to report the resolution of the cryo-EM reconstructions (or at least the resolution range for different components in the composite map) early on in the results. Currently, the resolution for the nine RuvB maps is reported at line 113, but the resolution for the RuvA core is not reported at all in the main text or figure legends.

- Scale bars should be included in Fig. 1a,f.

- Is there any rationale between choosing time points of 30 min and 5 hr for the two datasets? If so, it would be useful to explain this.

- The notation system referring to different RuvB states (s0t2+A, s2t2-s5t2, etc.) introduced in lines 115-117 is rather complicated. While I appreciate that finding a way to name the different states is not straightforward, the current notation system may make it difficult for non-experts to understand. If simplifying the naming of different states is not possible, it could be useful to include a statement here clarifying that five of the states (s1-s5) represent five different points in the ATP hydrolysis cycle, and two states (s0-A and s0+A) represent two initiation states, as this is only specified later in the manuscript.

- Ext. Data Fig. 1c legend, "In lower protein concentration (>200nM)": The '>' sign should be reversed.

- Ext. Data Fig. 14 is first referred to on line 159, and contains multiple references to the 'converter module'. However, the concept of the converter module is only introduced much later in the main text at line 299, which makes interpretation of Ext. Data Fig. 14 difficult. It would be useful to introduce this term earlier in the main text for clarity.

- Line 176, “RuvB4 protomers in cluster D contain either ATP γ S (s1t2), ADP (s5t2)”: These should be swapped; ADP is seen in s1 and ATP γ S in s5.
- Line 215: Could the authors please confirm whether the “cis-acting arginine finger”, Arg218, is in fact the conserved sensor 2 arginine seen in most AAA+ ATPases? Most AAA+ ATPases have a conserved sensor 2 arginine in the small AAA+ domain that coordinates the γ -phosphate. If it is, it would be good to refer to it as such for consistency. Similarly, it seems that Arg171 is in fact the canonical arginine finger and should be referred to as such.
- In Fig. 4, it would be useful to more clearly indicate the location of RuvAD3 in the structures, as the colouring makes it hard to distinguish between RuvA and RuvB.
- Lines 270/273: ‘PS-I superclade’ and ‘PS1 superclade’ should be ‘PS1 insert superclade’.
- Line 226, ‘at high map thresholds’: Should this be at low map thresholds?
- In line 351, there seems to be a missing word in “...hybrid conformation, which is from any of the conformations...”.
- Line 353: Should the time for the t2 dataset be 5 or 6 hr?
- Line 373: ‘DNA-disengaged’ could be reworded to ‘substrate-disengaged’, reflecting the fact that many AAA+ ATPases are in fact protein translocases.
- The captions for movies 1-6 should be more descriptive.
- Ext. Data Fig. 12: The legend refers to panels a and b, but only panel a appears in the figure.
- It seems based on the figures and on the model presented in movie 10 that only one RuvA tetramer contributes D3 domains during branch migration. Did the authors ever see cryo-EM density that could correspond to RuvAD3 domains from the second RuvA tetramer? Do they expect that the second RuvA tetramer would ever interact with RuvB during ATP hydrolysis and branch migration, or that it would play a purely structural role in the complex? This could be useful to mention briefly in the discussion.

Referee #3:

Summary:

The work of Wald et al. uses time-lapse cryo-EM to understand how RuvA and RuvB coordinate to drive branch migration. This paper is a tough read with a lot of details and data, and the images/movies are beautiful. It has the potential to make a significant contribution to the field of homologous recombination and DNA repair.

However, there are some critical issues that detract from this study. First, and this was very disappointing, the authors used RuvA from *Salmonella* and RuvB from *Streptococcus*. No explanation was given and the authors stated that initial studies were done with *E. coli* proteins. So why not use the *E. coli* proteins? At its core, protein

homology or lack thereof does not limit the quality of the work performed. However, it is impossible to relate residue numbers to *E. coli* numbering. This makes sense because a large body of data exists for the *E. coli* proteins and when trying to relate the mechanism proposed and residues used, it made no sense. Also and very important, does the interspecies RuvAB complex catalyze branch migration as efficiently as the *E. coli* RuvAB control?

Then, the authors have omitted multiple key references and as a result, they do not do a good job of relating their mechanism to what is known for RuvAB. This is significant. In particular the single molecule study from the Harada group showing that the DNA rotates during branch migration is not included. Further, the control of the ATPase cycle of RuvB by RuvA domain III has already been published, so this is not new. The authors did not do a good job relating their mechanism to what is already known.

Finally, the issue of branch migration being driven by a RuvAB complex with either one or two RuvA tetramers bound is critical. This is vaguely addressed early on, but the authors must do a better job as this is an important issue.

Listing of issues:

1. I had great difficulty trying to relate residues numbers in the text to what is published. When I pored over the M&M, I found out why. The authors use RuvA from *Salmonella typhimurium* and RuvB from *Streptococcus thermophilus*. This boggles the mind why this was done. They compare their data to the large body from the *E. coli* literature. The *E. coli* and *Streptococcus thermophilus* RuvB proteins are 60% identical, while the RuvA proteins are 95% identical. They used the *Streptococcus thermophilus* numbering to explain the data, and this made it difficult to follow when keeping *E. coli* numbering in mind.
2. Use of the word “protomer” is inaccurate and confusing and also frustrating. The definition is the smallest structural unit of an oligomeric protein composed of at least two different protein chains. The work of the Cox lab to understand the ATPase activity of RuvAB determined that RuvB monomers are arranged or working together as a trimer of dimers. Thus when reading I was trying to understand how the trimer of dimers with each dimer being a protomer could function to drive duplex translocation. In the end I concluded that the authors mean a monomer within the hexamer. Thus confusing terminology has to be changed or clearly defined at the outset.
3. The references from the Cox lab are not cited. These are critical to the ATPase mechanism proposed by the authors. See *Biochemistry* 34, 9809-9818 and *Biochemistry* 35, 11228-11238.
4. DNA rotation has been published and is discussed in *Proc. Natl Acad. Sci. U. S. A.* 2006 103, 11544-8. This paper has not been included.
5. Two additional single molecule papers relevant to this study are also absent. These are DOI: 10.1073/pnas.0404369101 and DOI: 0404332101 [pii]10.1073/pnas.0404332101.
6. The role of RuvB R174 as an allosteric effector in the ATPase activity of RuvB is not discussed. This is the work of Hishida et al. (*Proc. Natl Acad. Sci. U. S. A.* 2004 101, 9573–9577). This paper is also not cited. This may correspond to some of the arginine residues discussed in the text, but this was not made clear.
7. "Oligonucleotides to construct Holliday junctions were purified by agarose gel electrophoresis.": Either this is an error in writing, or the DNA molecules are not homogeneous. The arms of each HJ are 56 nt in length. I doubt that

an agarose gel would enable purification of this length of DNA.

8. Line 248: Change “RuvA operates” to “RuvA regulates”.

9. Line 365: A reference for the concentrations of ATP and ADP must be provided.

10. In the Introduction, state that the HJ is a central intermediate in recombination and fork rescue (cite - Nucleic Acids Res. 2009 37, 3475-92 and/or Nucleic Acids Res. 2021 49, 4220-4238).

11. The authors used a mixture of ATP- γ -S and ADP to slow reactions down. They need to show that this mixture is competent for branch migration.

12. For the reconstitution of RuvA-B-ATP- γ -S/ADP complexes, the authors purify RuvA-HJ complexes by gel filtration. Under the conditions used, do they separate the one RuvA tetramer-HJ complex from the two tetramer complex? If so, which complex did they use for reconstitution, and does this influence the images obtained?

Suggestions for improvement:

1. Cite the relevant references and relate the current data to what is already known.

2. Create a cartoon model of the step-wise mechanism, indicating the power stroke, cluster switch and converter module, etc.

3. Fix the residue numbering.

4. Define protomers, or use monomers: either RuvB monomer or RuvA monomer.

Author Rebuttals to Initial Comments:

Point-to-Point reply to reviewer comments:

Referee #1:

Summary:

The present work by Wald et al. describes the complete structure of the bacterial RuvAB Holliday Junction (HJ) branch migration system. Using time-resolved cryoEM, the complex – which consists of as many as 12 RuvB motor subunits (arranged into hexameric rings), two RuvA tetramers, and an HJ DNA – is captured in a series of intermediate catalytic states. The reconstructions are used to develop a model which posits that RuvB uses a rotary ATPase cycle to exert a sequence of 'walking' and 'pulling' movements that lead to DNA translocation. The ATP hydrolysis cycle is suggested to produce the pulling force, while a 'cluster switch' in which each subunit shifts over to take the position of its partner (with respect to RuvA) is implicated in walking. The net result is the pulling of the DNA through RuvA to effect strand exchange and branch migration.

Although in one sense the findings corroborate long-standing models for how the RuvAB system works, it should be clear: the work is an absolute tour de force, not just for understanding the mechanisms of HJ branch migration, but for understanding the operation of ring ATPase motors in general. There is no question that Nature is an appropriate forum for publication.

Reply:

We are very grateful for this assessment of our work and share the enthusiasm with reviewer #1 about understanding the migration of HJ branches and the functioning of ATPase motors widely used in biology.

This said, the data and model, as described in the text, are extremely complicated and difficult to follow (borderline impenetrable in places). Part of the inaccessibility and confusion arises from the haphazard order of figure panel callouts. Another is the very complex way in which the data are variously described with respect to states, clusters, walking, rotating, pulling, etc. When you need a glossary to understand the work (and even this doesn't fully achieve the goal), there is a problem. The density of the organization and technical jargon of the writing detracts from the tremendous impact of structural data and their mechanistic implications.

Reply:

We would like to thank the reviewer for making very helpful suggestions about how to improve the accessibility of our manuscript. Following the reviewer's suggestions (and in part also from reviewer #2 and #3), we (1) ordered the figure panel callouts throughout the manuscript, (2) simplified the manuscript and reduced or removed the amount of "technical jargon", and (3) re-worked the figures. Because of this, we now think that a glossary is not needed anymore.

Given this caveat, the following comments are meant to aid in clarifying the work. They should not be taken as weaknesses that might preclude its eventual acceptance by the journal.

Primary comments:

- 1) There is a question as to whether the changes in ATPase status in the RuvB ring are fully known with respect to their rotational disposition to RuvA. The authors obtain a series of RuvB ATPase states (s1-s5, as well as s0) from reconstructions that were generated using masking and local refinement procedures that isolate the motor region to improve map quality. By comparison, the observation that the RuvB hexamer can rotate 60° with respect to RuvA appears to be based on lower-resolution (non-focused) reconstructions of the entire system. A key question is therefore whether intermediate RuvB rotational substeps with respect to RuvA are seen for each RuvB state (s1-s5) – as implied by Fig. 3d and by movies 7-10 – or whether these substeps are inferred from the observations that there are 1) multiple RuvB catalytic states and 2) two distinct RuvB/RuvA rotational states.

If the former, then it needs to be clarified that rotational substeps indeed correlate with specific ATPase states, and that these are observed at a convincing level of resolution. If the latter, then the actual translocation mechanism may need to be reconsidered.

Reply:

Thank you very much for this comment, which allows us to better clarify the data that we obtained and the deduced working model. As reviewer #1 clearly pointed out, the observation of a ~60 degrees rotation of RuvB in respect to RuvA was obtained from lower-resolution structures, which precludes linkage of a specific catalytic state (s1-s5) with a specific rotational substrate. Our figure and associated movie indeed suggest such a linkage and is based on the notion that without such a linkage the catalytic states would be going through the catalytic states on a rather fixed position in respect to RuvA. In such a scenario, the DNA substrate would likely be stretched (and its structure altered between the RuvA core and the bound RuvB motor) until the pulling force of the RuvB motor overcomes the resistance of the substrate to translocate and subsequently cause RuvB to rotate by 60 degrees in a 'switch-like' mechanism. Alternatively, the sequence of catalytic states (along with the conformational changes of the motor) is associated with different rotational substates along a 60 degree 'rotation trajectory' to complete one full catalytic cycle, lifting the DNA and ultimately translocate DNA at the same time. Despite the fact that we have extensively analysed our very large dataset (~900k particles), we could not link a specific state to a defined rotational angle (within 0-60 degrees) at this point in time, suggesting that such transitions actually occur along a rotational trajectory but at less defined angles. Importantly, in such a model, DNA deformation would not occur and the RuvB motor, which binds to the DNA with four subunits (DNA-engaged subunits) rotates together as a unit with the DNA. (Rotation of the DNA has been observed by single molecule studies previously by the Harada group (also mentioned by reviewer #3)). To clarify that the catalytic sub-steps cannot be currently assigned to a defined rotational angle, we (1) removed figure 3d from the original manuscript, (2) improved legends of the corresponding movie captions (movie 7-10) and (3) adapted the following passages within the discussion in our current manuscript accordingly:

Line number 140:

“Moreover, as the RuvB subunits are positioned $\sim 60^\circ$ apart from each other within the RuvB hexamer, these data further imply that the RuvB motors’ rotation is linked to the events occurring within one translocation step.”

Line number 328:

“In summary, the RuvB AAA+ motor undergoes two consecutive processes (nucleotide cycle and *cluster switch*) (Fig. 5f) that account for both the maintenance of the unaltered structure of the DNA and the need for its rotation during branch migration.”

And also discussed at:

Line number 397:

“To be able to repeatedly exert their critical function on a rotating RuvB motor, RuvA^{D3} domains need to constantly release from the RuvB hexamer and bind to newly-generated binding interfaces that are produced by the nucleotide cycle. Although the driving force behind this rotation remains to be identified, it appears reasonable that the energy for this motion is derived from the nucleotide cycle. As the DNA substrate already refolds into a double helix within the confinement of the double tetrameric RuvA core, we propose that the RuvB motor rotation is powered by the rewinding of the translocating DNA (movie 9). In this view of the RuvAB machinery, the double RuvA tetramer serves an important function in stabilizing the HJ, ensuring that the two DNA substrates can rewind into a double helix and providing a rationale as to why RuvB motors rotate.”

- 2) Regardless of whether RuvB rotational substeps are seen with respect to RuvA, there may be a simpler way to explain the action of RuvB that would seem compatible with the data as presented. The key is to focus on subunits A and D. As subunit A goes through a hydrolysis cycle (ATP to ADP-Mg to ADP), subunit D goes through an exchange cycle (ADP to Apo to ATP).

This cycle causes the RuvB ring to rotate by 60° with respect to RuvA. (Rotation may occur because DNA is helical and because RuvA acts as a partial impediment to DNA twisting, necessitating that RuvB initially revolve around DNA as it climbs along the duplex, rather than vice versa; however, rotation need not be a prerequisite for translocation *sensu strictu* – see (3) below.) Hydrolysis and release of Mg⁺⁺ in subunit A frees this protomer to undock from its position at the top of the staircase, providing conformational plasticity that concomitantly allows subunit D to undergo conformational changes associated with ADP ejection and ATP binding. This action also leads subunit E (along with its partner, subunit F) to shift ‘downward’ (toward RuvA), where E picks up a new dinucleobase unit to constitute a motor step. Because subunits D and E are anchored to domain 3 of RuvA, the rotation of RuvB is accompanied by a lifting of the DNA out of RuvA, advancing HJ migration by a step of 2 nt, the step size of the RuvB motor. Once the rotation is complete, domain 3 must dissociate from RuvB because its movement is restricted by a tether that connects this region to the rest of RuvA; this element is then free to bind to the newly advanced protomers (one to E, one to F). In this view, which parallels the action of related AAA+ translocases, each round of ATP hydrolysis drives a single step of HJ migration of 2.

Reply:

Excellent. We thank reviewer for the concise description of the cycle and have used it as a guideline to assemble a final paragraph in the “Results” section combining and highlighting the individual steps about the initiation and processive branch migration as “Integrative model for initiation and processive branch migration of the RuvAB complex”. It also contains now that RuvA plays a critical role for regulating the nucleotide cycle but also for DNA lifting (as presented in details in the individual paragraphs of the results section). Of note, previous biochemical studies have firmly established that RuvA stimulates the ATPase activity of RuvB, yet the underlying mechanism behind this remained unresolved. In our study, we now uncovered the mechanistic basis for this observation: the binding of the second RuvA^{D3} to subunit E on the RuvB motor exerts a wedge-like effect on the hexamer (s1>s2) causing ATP hydrolysis in subunit A (s2>s3) but also ADP ejection in subunit D (s1>s2). RuvA thus serves as an activator for ATPase activity (on subunit A) and a nucleotide exchange factor (subunit D) at the same time.

- 3) A point to consider, which may or may not accord with the data, is whether RuvA is acting like a stator. In this view, the 60° rotated state might correspond to a stable form when RuvB is actively translocating but RuvA is simply restraining further motor rotation. Rpt1-6 of the proteasome and the F0/F1 ATP synthase are exemplar rotary ATPases that have stators which prevent ring rotation. Regardless, the authors should note that the rotation model they are proposing for RuvB with respect to RuvA is analogous to that advanced for the action of ClpA (or ClpX) when tethered to ClpP.

Reply:

As the reviewer pointed out correctly, the wording ‘stator’ is intimately connected to the association of a mechanical device that restricts a rotary motion. Our structural data reveal that RuvA stimulates the ATP hydrolysis cycle, which in turn powers the rotation of the RuvB motor. Consequently, RuvA does not restrain but trigger the rotation of the RuvB ring, which is why we believe naming RuvA a stator has potential to unnecessarily confuse the readers of our manuscript. However, we do agree that the function of RuvA is otherwise in good agreement with that of a stator as the binding of RuvA to the rotational RuvB ‘unit’ ultimately enables the conversion of the chemical energy contained in ATP into the mechanical translocation of the HJ through the RuvA core (s1-s5).

We are aware of the newest ClpA (ClpX) /ClpP models. Initially, we were very curious to compare RuvB (RuvAB) rotational model with the Clp systems. However, ClpA/X motor rotation in respect to the ClpP protease is still under debate, thus we finally decided not to include a direct comparison. We think, however, that a good place to discuss the mechanism of various AAA+ ATPases with a specific focus covering the linkage of molecular rotation, ATP catalysis, and intracellular signalling at length could be a dedicated focused review or “News and Views” article in the future.

Nevertheless, in the light of the following studies describing (1) the rotation¹ and (2) the proteolytic activity without rotation², we are currently exploring the hypothesis that some AAA+ motors (like ClpA/X) optionally rotate, while others AAA+ motors (like RuvB) must rotate in order to perform their principal task (movement of the substrate). In a simplified view it could be the type or nature of the substrate that would require concomitant rotation of the AAA+ motor instead of rotating or revolving inside the AAA+ motor pore itself.

We added following citations to Clp proteolytic system:

Line number 413:

“To operate the nucleotide cycle in these motors, putative *converter*-interactors must therefore be able to reach every subunit of the AAA+ motor. This may provide a rationale as to why ring-shaped AAA+ motors are often found to be intimately embedded within multimeric scaffolds such as in the proteasome or ClpA/X-P^{61,66,68,69}.”

- 4) It's very important that the transitions in movies 7-10 be distinguished from the actual observed states, s1-s5. Movies that pause at each reconstruction and then morph between these states would be most helpful for showing which frames correspond to reality and which to a morph interpolation. For example, is the state in which both domain 3 elements of RuvA are pulled back from RuvB observed or inferred?

We greatly appreciate this suggestion and modified the movies and the corresponding movie captions to clarify, which parts have been generated by interpolation.

- 5) Is it possible to reconcile the observed states of RuvB with single molecule data obtained by the Bustamante lab for a different dsDNA translocase (the phi29 packaging motor), which has shown that ATPase activity occurs in bursts? These data have suggested that there are hydrolysis-dependent micro- and macrosteps for the motor. The system may be too different than RuvB, but perhaps it is worth considering.

Reply:

This is an interesting and exciting suggestion as it would allow to further deconvolute rotation, states, and enzymatic activity. We are currently entertaining as well as engaging with experts in the field to potentially perform specific experiments in a future collaborative project. Although these experiments are beyond the current manuscript, we would like to thank the reviewer for this suggestion.

Other comments:

- 1) Ext. Data Fig. 12 is called out before EDF 3, EDF 10 after 12, EDF 11d before 11a, Fig. 2e before 2d – these are just a few examples of out-of-order callouts. Please order all supplementary figures and panels as they are called out in the text.

Reply:

We apologise for the confusing figure callouts in the original version of the manuscript. In the revised version this has been fixed.

2) Line 122: "Akin to other AAA+ translocases"

Reply:

Fixed.

Line number 130:

"Akin to other AAA+ translocases⁴⁵⁻⁵⁰, four RuvB subunits (A, B, C, D) together assemble into a spiral staircase."

3) Line 126 and elsewhere: Wouldn't it make more sense to label as RuvBE+F, as this is how the figures depict it, rather than RubB5+6?

Reply:

We changed the nomenclature throughout the entire paper. Now, all subunits are named with capital letters ranging from A-F and clusters [A]-[F].

4) Line 133: Why does this correlation infer a single (i.e., 2-nt) translocation step? Couldn't this be linked to a larger macrostep, or a stall point before resetting?

Reply:

Thank you for raising this point. We agree with reviewer #2, that strictly speaking, we should not directly infer the size of the translocation step only from the staircase organization of the RuvB-DNA interface. Of note, recent literature for the AAA+ hexameric ATPases suggests that the size of such a step stems from correlating a progressing mechanism around the ring together with the staircase organisation of the substrate-engaged subunits.

Hence, in the revised version we removed this part and, instead, conclude the size of the translocation step from the following observations: (1) the average travelled distance by subunit E matches two nucleotides, (2) the organisation of the RuvB-DNA engaged subunits in the staircase (every two nucleotides), and (3) the rigid character of the RuvB subunits building the staircase around the substrate (A, B, C and D). This correlation becomes very striking when one aligns all nucleotide cycle RuvB motor states to the "lifting" ATPase domains of subunit E. When doing so, the DNA located in the center of the motor advances by 2 nucleotides (distance of 7 Å) (Fig. 5 a-d).

Line number 171:

“Notably, the length of the trajectory within cluster [E] corresponds well to the step size of the RuvB staircase of two nucleotides (the distance between nucleotides in DNA is ~ 3.5 Å), suggesting that the five RuvB structures (s1 to s5) could represent consecutive atomic snapshots of an active RuvB motor as it progresses through one translocation step.”

Line number 311:

“The analysis revealed that the sequential movement follows a trajectory that translates into a lifting motion of the RuvB motor, in which the individual areas of the hexamer lift proportionally to their distance from subunit E (Fig. 3b, 5a). This causes the DNA binding interface to be lifted by ~ 7.0 Å

away from the RuvA-HJ core. Hence, our data provide evidence that RuvB motors act as molecular levers, which convert the energy obtained throughout the nucleotide cycle into a pulling force to physically move the DNA by ~ 7.0 Å i.e. two nucleotides (RuvB motor “pulling”) and thereby achieve DNA recombination (Fig. 5b-c, Movie 8-9).”

Line number 378:

“(4) Because RuvB is anchored to domain III of RuvA during the nucleotide cycle, rotation of RuvB is accompanied by a pulling of the DNA out of the RuvA core, advancing branch migration by 2 nt (power stroke).”

- 5) Lines 193-195: The textual description of the movements seem a bit misleading. The way this sentence is phrased makes it seem as if DNA is leaping from one set of subunits to another set, when in reality there is a release of the DNA from the subunit at the top of the staircase as the bottom-most subunit engages it which changes the set of bound subunits from one cluster to another.

Reply:

Thank you very much for pointing that out. We have reworked this part in our revision.

Line number 197:

“Remarkably, the DNA remains bound to all four staircase subunits (A to D) across all five states and, hence, the interaction of the DNA substrate with these subunits is independent of the type of nucleotide bound, including at the ATP hydrolysis (A) and at the exchange position (D) (Extended Data Fig. 10c). Consequently, our data reveal that in order to relocate the DNA substrate inside the central RuvB motor pore, RuvB subunits must be subject to additional conformational changes that follow the nucleotide cycle. We therefore reason that the nucleotide cycle in fact functions first to prime the RuvB subunits over five states to then acquire the conformations of their respective neighbouring clusters. This is also supported by the fact that the nucleotide arrangement in state s5 corresponds to the same configuration as in state s1, but the respective conformations of the six subunits have shifted forward by one to occupy the new successor state ($s5 \rightarrow s1'$: $A(s5) \rightarrow F(s1')$, $B(s5) \rightarrow A(s1')$, etc). It follows that all subsequent processes now take place in the respective adjacent subunit, implying that nucleotide hydrolysis and all other processes operate around the hexameric ring in repeated sequences. When this event occurs, all six RuvB subunits simultaneously transition to the next conformational cluster without any additional changes to the nucleotide arrangement, resetting the conformation of the entire hexamer to state s1. We therefore refer to this process as a *cluster switch* ($s5 \rightarrow s1'$) (Extended Data Figure 11).”

- 6) Line 242: Do the authors really mean 1 Å? This doesn't seem like a very significant distance, given the resolution of the data.

Reply:

As pointed out correctly, the travelled distances are small and can be measured at this resolution. Although we agree that the movements are small, we see them throughout the states. (Extended Data Fig. 9b only show movements larger than 1 Å, which are prominent in the head domain of subunit A). We think that these movements prime the head domain of subunit A to gradually decrease its affinity towards the DNA to then allow the cluster switch (subunit A will adopt the DNA-disengaged conformation of cluster [F] of in subunit A). Such movements are not observed for example in the DNA-bound subunits B, C, which maintain the binding to the DNA even after the cluster switch and are thus not primed. In our revised manuscript we have specifically removed the comment about the 1 Å distance to also be considerate of the fact that additional experimental data would need to be collected to strengthen such a statement. Moreover, we now show only the movements larger than 1 Å in all Figures showing the distance analysis (Fig. 4f, Extended Data Fig. 9 a,b).

- 7) EDF 15: Why do the columns for panels a and b have different labels? Is s1t1 the same as s5*(t1)? Also, it would be clearer for this figure and Fig. 3c to have the ADP-Mg states highlighted in the table, as opposed to just ADP.

Reply:

Fixed. In state "s3" we added an asterisk to highlight the presence of ADP (Fig. 3c, Extended Data Fig. 10a-b).

1. Kim, S., Zuromski, K. L., Bell, T. A., Sauer, R. T. & Baker, T. A. Clpap proteolysis does not require rotation of the clpa unfoldase relative to clpp. *Elife* **9**, 1–12 (2020).
2. Ripstein, Z. A., Vahidi, S., Houry, W. A., Rubinstein, J. L. & Kay, L. E. A processive rotary mechanism couples substrate unfolding and proteolysis in the ClpXP degradation machinery. *Elife* **9**, (2020).

Referee #2:

Summary:

In this manuscript, Wald et al. present cryo-EM structures of RuvAB in complex with the Holliday junction (HJ) in several conformational states, allowing them to propose a detailed mechanism for branch migration during DNA recombination. They provide high-resolution information for the assembly of the RuvAB-HJ complex, and most impressively, provide a complete picture of the ATP hydrolysis cycle in the AAA+ motor, RuvB, through extensive cryo-EM processing and structural analysis. This in particular will be of wide interest to the AAA+ ATPase field. They also present an elegant explanation for how RuvA is able to stimulate the ATPase activity of RuvB, by analysing the conformational changes associated with RuvAD3 binding during the ATPase cycle. This manuscript represents an impressive amount of work – the structural analysis is of high quality, and the manuscript is generally well-written.

Reply:

We would like to thank the reviewer for critically evaluating our manuscript and very helpful suggestions. We highly appreciate the overall positive assessment of presented mechanistic models, cryoEM structures and the final manuscript. We also appreciate the recognition about the amount of work and effort invested into this study. We are excited to hear from reviewer #2 that the content of our work will be of wide interest to the AAA+ ATPase field.

I do have some specific comments and questions below which should be addressed, none of which are too major:

- 1) In the PDB validation reports for RuvB motors in complex with D3 domains of RuvA, very often it appears that the RuvA chains (usually chains G and H) have up to 100% of residues being flagged as having “poor fit to the EM map”. Do the authors know why this would be? Would it be possible to show some close-up images of the RuvAD3 PDB fitted to the EM density to confirm that the models have been built in correctly? Specifically, this is the case for PDB 7PBQ chain G, 7PBN chains G/H, 7PBO chains G/H, 7PBP chains G/H, 7PBS chain G, 7PBM chains G/H, 7PBT chain G, and 7PBL chain G.

Reply:

We also noticed the comment obtained from the validation report. However, this behaviour stems from the fact of anisotropic resolutions within cryoEM structures and thresholding with a defined number used for validation. In particular, resolution often drops towards the edge of a map resulting in less defined areas at specific resolution filtering and sharpening.

In our cases, RuvA^{D3} sits at the periphery of the motor and the corresponding densities are less well resolved compared to other parts of the motor. During model building, several maps of the same 3D reconstruction

are generated by filtering to varying resolutions (within the range obtained by calculating the local resolution of the maps) and sharpening (B-factors). All these maps are then used to manually inspect and build atomic models. We are convinced that our models have been built as accurately as possible and we now provide close-up views of our RuvA^{D3} models in a dedicated Extended Data Fig. 2e.

- 2) Line 176, “density that shows remote similarity to nucleotides only at high map thresholds (s2t2-s4t2)”: The densities in these ‘apo’ subunits do not actually look too dissimilar to the ADP density in cluster D s1t2 in these images, although there are indeed discontinuities in the densities. Were all of these NBP panels rendered with equivalent thresholds, or were the thresholds for s2t2-s4t2 lowered to visualise the density here? If these panels are indeed visualised at lower map thresholds (not ‘higher’) then this should be noted on Ext. Data Fig. 15. Otherwise I would moderate the phrase ‘remote similarity’.

Reply:

All panels were rendered with the same thresholds within a dataset (t1 and t2 datasets) and are thus directly comparable especially between the states s1 to s5 (of t2). To reemphasize this point, we now mention this fact also in the panel figure legends (Extended Data Fig. 12a).

Line number 183:

“The fragmented and interrupted densities are indicative of low nucleotide occupancy, suggesting that these sites have an apo-like configuration.”

- 3) It would greatly help the flow of the paper if the Ext. Data figures appeared in the order they are referred to in the text. Currently, the reader has to jump back and forth through 19 Ext. Data figures, which makes the paper harder to read. In addition, Ext. Data Figs 3, 5, and 6 do not appear to be referred to at all in the text; this should be rectified.

Reply:

This is an excellent suggestion, and we apologise for the inconvenience. We changed the figure panel callouts according to their appearance in the text.

- 4) It should be mentioned somewhere in the introduction (or early in the results) that the RuvA and RuvB proteins used are from *Salmonella typhimurium* and *Streptococcus thermophilus*, respectively. Could the authors comment on why proteins from two different species were used? Intuitively it would make more sense to characterise a complex where both proteins are from the same species.

Reply:

We agree and have implemented the following sentences in our revised version. We addressed this point already at the beginning of this letter. For convenience, we put the entire comment here again:

>>>

The revised manuscript now contains additional biochemical data elaborating the ATP-dependent branch migration of the Holliday junction of the *in-vitro* assembled homo- and hetero- complexes. We now show (Fig. 1b, Extended Data Fig. 1c-e) that similar biochemical activities are observed between the homo- and hetero-complexes suggesting that individual components from different species can be exchanged while keeping the system active. This observation is also in line with previous studies^{1,2} highlighting that the underlying mechanism of branch migration employed by RuvAB is highly conserved. We now also mention that we focused our studies on heterocomplexes, because during the course of sample optimization only hetero-complexes could be vitrified suitable for structural studies.

The following sentence has been introduced at line number 79:

“To visualize this process, we reconstituted RuvAB-HJ complexes *in vitro* from individually purified components originating from *Salmonella typhimurium* and *Streptococcus thermophilus*, respectively, and tested their functionality in a branch migration assay (Fig. 1b). Both, homo- (RuvA and RuvB from *S. typhimurium*), and hetero- (RuvA from *S. typhimurium* and RuvB from *S. thermophilus*) complexes processed the HJ similarly upon addition of ATP, suggesting a highly conserved underlying mechanism due to interchangeability of individual components (Fig. 1b, Extended Data Fig. 1a-e).”

<<<

- 5) It would be useful to report the resolution of the cryo-EM reconstructions (or at least the resolution range for different components in the composite map) early on in the results. Currently, the resolution for the nine RuvB maps is reported at line 113, but the resolution for the RuvA core is not reported at all in the main text or figure legends.

Reply:

We agree here as well and have added the resolution of the RuvA complex in the manuscript text as follows:

Line number 91:

“The cryoEM structure of the RuvAB-HJ complex revealed highly flexible and linearly arranged bipartite (3.9 Å) and tripartite (8 Å) assemblies, with eight RuvA molecules symmetrically arranged in two tetramers (3.3 Å) and the four-way HJ flanked by, and flexibly connected to, one or two RuvB hexamers (2.9 - 4.1 Å), respectively (Fig. 1c-e, Extended Data Fig. 2, 3a-d, 4a-b, Extended Data Table 1-2).”

6) Scale bars should be included in Fig. 1a,f.

Reply:

Thank you for this suggestion. In the revised version of the manuscript, we included scale bars in Figure 1d, f.

7) Is there any rationale between choosing time points of 30 min and 5 hr for the two datasets? If so, it would be useful to explain this.

Reply:

The rationale for choosing the time points $t_1=30$ minutes and $t_2=5$ hours was the result of a systematic sample optimisation process with the conditions to obtain and analyse fully assembled complexes as early (after the start of the in vitro reconstitutions in the presence of nucleotides) and as late as possible but before any disintegration of complexes. We found that within the first 30 minutes the number of fully assembled particles gradually increased to reach a high number of particles suitable for cryoEM.

Furthermore, we noticed that after 5 hours particles started to disassemble impacting greatly on the sample quality for vitrification. Consequently, we choose 30 minutes (t_1 dataset) and 5 hours (t_2 dataset) for cryoEM data collection for single particle analysis. In the revised version of the manuscript, we explain this directly in the main text.

Line number 85:

“To capture the catalytic steps of this rapid process, we first slowed down the reaction by replacing ATP with an equimolar mixture of the slowly hydrolysable ATP γ S³⁹ and ADP and incubated the reaction on ice either for 30 minutes (dataset “t1”) or for 5 hours (dataset “t2”) to mimic an initiation and an equilibration phase of the RuvAB-HJ complex (Extended Data Fig. 1f-i). Subsequent vitrification of samples led to aggregates and low numbers of individual particles for homo- complexes, whereas the distribution of hetero-complexes over the grid was largely monodisperse and suitable for single particle analysis.”

8) The notation system referring to different RuvB states (s0t2+A, s2t2-s5t2, etc.) introduced in lines 115-117 is rather complicated. While I appreciate that finding a way to name the different states is not straightforward, the current notation system may make it difficult for non-experts to understand. If simplifying the naming of different states is not possible, it could be useful to include a statement here clarifying that five of the states (s1-s5) represent five different points in the ATP hydrolysis cycle, and two states (s0-A and s0+A) represent two initiation states, as this is only specified later in the manuscript.

Reply:

We agree that the large number of subunits, states, presence/absence of RuvA. etc turns into a large number of defined conformations or configurations and as such requires a notation system that is simple as well as precise at the same time. We now have simplified the system and call each of the RuvB protein subunits within the hexamer with a capital letter ranging from A-F (corresponding to the chain-numbering in the deposited structures of the PDB files), the ensemble of conformations of the subunit cluster [A] to cluster [F], and the specific states s1-s5, as well as s0. For the latter we indicate whether RuvA is bound or not (-/+ A). The subscript t2 had been eliminated and is only shown for the t1 dataset to highlight their origin.

9) Ext. Data Fig. 1c legend, “In lower protein concentration (>200nM)”: The ‘>’ sign should be reversed.

Reply:

Thank you for spotting the mistake. We have corrected this.

10) Ext. Data Fig. 14 is first referred to on line 159, and contains multiple references to the ‘converter module’. However, the concept of the converter module is only introduced much later in the main text at line 299, which makes interpretation of Ext. Data Fig. 14 difficult. It would be useful to introduce this term earlier in the main text for clarity.

Reply:

Thank you for the very careful reading and spotting this inconsistency. In the revised version of the manuscript, we refer to Extended Data Fig 13 in the paragraph “*Regulation of the nucleotide cycle in RuvB hexamers by forward and retrograde inter-subunit signalling*”. More specifically, we refer to Extended Data Fig. 13h and Fig. 4f-g which show the *converter* on the end of the paragraph which describes it.

Line number 293:

“The acquisition of a new ATP molecule (s4→s5) is then accompanied by a concerted motion of subunits E and F together with the large domain of subunit D (hereafter called *converter*: F:E:D^L) (Fig. 4f-g, Extended Data Fig. 13h). As a part of this motion, the coordination of the newly obtained ATP molecule is restored by the N-terminus in subunit D (Extended Data Fig. 13h-i). Consequently, the gate-opening (cluster [E]) and gate-closing (cluster [D]) motions of the RuvB N-terminus serve as additional proof for the directionality of the nucleotide cycle. Finally, the retrograde signalling causes subunit D (large domain) to become part of the rigid area in the RuvB motor, which marks the completion of the nucleotide cycle (Fig. 4e, Movie 6).

In summary, our findings establish that the conformations of all RuvB subunits are context-dependent within the hexamer and the *converter* (F:E:D^L) functions as a RuvB motor-operating multi-domain module, which undergoes highly coordinated motions during the nucleotide cycle. The critical position of subunit E in the centre of this module enables the binding of RuvA^{D3} to pass information through inter-subunit signalling to stimulate ATP hydrolysis in distant subunit A and nucleotide exchange in adjacent subunit D (Fig. 4f-g).”

- 11) Line 176, “RuvB4 protomers in cluster D contain either ATPγS (s1t2), ADP (s5t2)”: These should be swapped; ADP is seen in s1 and ATPγS in s5.

Reply:

Fixed.

Line number 182:

“At the opposing lower side of the hexamer, cluster [D] contain either ADP (s1), fragmented and interrupted densities (s2 to s4), or ATPγS (s5).”

- 12) Line 215: Could the authors please confirm whether the “cis-acting arginine finger”, Arg218, is in fact the conserved sensor 2 arginine seen in most AAA+ ATPases? Most AAA+ ATPases have a conserved sensor 2 arginine in the small AAA+ domain that coordinates the γ-phosphate. If it is, it would be good to refer to it as such for consistency. Similarly, it seems that Arg171 is in fact the canonical arginine finger and should be referred to as such.

Reply:

This is a great suggestion. Indeed, both arginine residues represent previously well characterised residues across many ATPases. For consistency, we now refer to both as such with corresponding literature.

Line number 219:

“Additional contacts are provided by two conserved *cis*-acting arginine residues: Arg21 and sensor-2 arginine Arg218⁵³: Arg21 is located at the N-terminus and binds the ATP α -phosphate, whereas sensor-2 arginine Arg218 is in the small ATPase domain and mediates nucleotide-sensing (Fig. 3f). In agreement with previous studies, ATP γ S-Mg²⁺ *trans*-sensing is achieved by two elements: a conserved signature motif (127-Glu-Asp-130), located on alpha helix α 4, and *trans*-acting Arg171 on alpha helix α 5 (Fig. 3f)^{51,54}. As such, Arg171 represents the canonical arginine finger that is conserved in most AAA+ ATPases and directly coordinates the γ -phosphate⁵⁵.”

13) In Fig. 4, it would be useful to more clearly indicate the location of RuvAD3 in the structures, as the colouring makes it hard to distinguish between RuvA and RuvB.

Reply:

Thank you for this suggestion. In the revised version we changed Figure 4 accordingly. Overall, we made an effort to improve the figure clarity throughout the entire manuscript.

14) Lines 270/273: ‘PS-I superclade’ and ‘PS1 superclade’ should be ‘PS1 insert superclade’.

Reply:

Fixed.

Line number 57:

“Furthermore, these studies demonstrated that the third domain of RuvA (RuvA^{D3}) binds to the presensor-1 β -hairpin of RuvB, a distinguishing feature of the PS1 insert superclade^{29,30}, regulates branch migration and increases ATPase activity of the RuvB motor^{31,32}.”

Line number 260:

“Both RuvA^{D3} bind to a previously described hydrophobic composite interface in their respective RuvB subunits, which is composed of RuvB^L alpha helix α 3 and the presensor-1 β -hairpin (Extended Data Fig. 13b)¹⁵, which in other hexameric AAA+ motors of the PS1 insert superclade coordinates with their substrates either directly or indirectly^{45,59–63}.”

15) Line 226, ‘at high map thresholds’: Should this be at low map thresholds?

Reply:

Thank you for spotting the mistake. We have corrected this sentence.

Line number 337:

“Notably, specific densities are visible at low density thresholds, indicating partial presence of ATPγS and Mg²⁺, thus determining that an asymmetrically formed RuvB hexamer can carry up to 5 ATP molecules (Extended Data Fig. 10a).”

16) In line 351, there seems to be a missing word in “...hybrid conformation, which is from any of the conformations...”.

Reply:

Thank you for spotting this mistake. We fixed this in the revised version of the manuscript.

Line number 344:

“Moreover, we also noticed that the *converter* in s0 assumes a hybrid conformation, which is different from any of the conformations seen in the nucleotide cycle (s1 to s5) (Fig. 5g).”

17) Line 353: Should the time for the t2 dataset be 5 or 6 hr?

Reply:

Typing error has been fixed in the revised manuscript.

Line number 350:

“To test this hypothesis, we performed cryoEM on RuvAB-HJ particles under the same conditions but vitrified the sample shortly after *in vitro* reconstitution (30min (t1 dataset) instead of 5 hours (t2)) (Extended Data Fig. 1j).”

18) Line 373: ‘DNA-disengaged’ could be reworded to ‘substrate-disengaged’, reflecting the fact that many AAA+ ATPases are in fact protein translocases.

Reply:

This is an excellent suggestion. We changed this in the revised manuscript.

Line number 395:

“Interestingly, substrate-disengaged subunits are a unifying feature across most ring-forming AAA+ motors^{29,48,50}, suggesting that variations of the *converter* likely also operate other AAA+ ATPases.”

19) The captions for movies 1-6 should be more descriptive.

Reply:

Thank you for this suggestion. Movie captions are now more descriptive.

20) Ext. Data Fig. 12: The legend refers to panels a and b, but only panel a appears in the figure.

Reply:

Thank you for spotting the mistake. We reworked Extended Data Fig. completely and also fixed this.

21) It seems based on the figures and on the model presented in movie 10 that only one RuvA tetramer contributes D3 domains during branch migration. Did the authors ever see cryo-EM density that could correspond to RuvAD3 domains from the second RuvA tetramer? Do they expect that the second RuvA tetramer would ever interact with RuvB during ATP hydrolysis and branch migration, or that it would play a purely structural role in the complex? This could be useful to mention briefly in the discussion.

Reply:

This point has been raised before. For convenience, we put our reply here as well:

>>>

Our structural investigation clearly showed that **two RuvA tetramer** complexes were reconstituted onto Holliday junctions. Of note, for the *in vitro* reconstitution experiments, we used individually purified components: **RuvA** was purified to homogeneity and characterised as a **monomer as assessed by gel filtration** (Extended Data Fig 1a).

Interestingly, the interpretation of the RuvAB-HJ complex structure revealed that a single RuvA tetramer constitutes the operational unit for branch migration during which the flexible RuvA^{D3} domains interact with the two opposing hexameric RuvB motors (two RuvA^{D3} domains operate one RuvB motor). This represents a new concept, describing a configuration that is compatible with the observed rotation (and the physical constraints of an extended linker between the RuvA core and its D3 domain), and is also in agreement with the subsequent HJ resolution step during which one RuvA tetramer bound at on one side of the HJ is exchanged for the RuvC nuclease at the same binding location (“crossover junction endodeoxyribonuclease”) to generate the so called RuvABC-HJ resolvosome. We think that the arrangement of two RuvA tetramers in the branch migration complex likely provides further biochemical stability for the complex by the second RuvA tetramer and thus may allow for efficient branch migration (as also proposed previously).

This is now also discussed *at line number 91*:

“The cryoEM structure of the RuvAB-HJ complex revealed highly flexible and linearly arranged bipartite (3.9 Å) and tripartite (8 Å) assemblies, with eight RuvA molecules symmetrically arranged in two tetramers (3.3 Å) and the four-way HJ flanked by, and flexibly connected to, one or two RuvB hexamers (2.9 - 4.1 Å), respectively (Fig. 1c-e, Extended Data Fig. 2, 3a-d, 4a-b, Extended Data Table 1-2). This architecture is consistent with previously proposed models of the RuvAB machinery^{22,23,25,31,32,40}. In both particle types, DNA enters and exits the RuvA core as a double helix, with one or two hexameric RuvB motors engaging the minor groove of the rejoined DNA (Fig. 1f). The RuvA core is physically connected to both RuvB motors through domain III of RuvA (RuvAD3) (Fig. 1c). On either side, two RuvAD3 are bound to adjacently positioned RuvB subunits, indicating that these domains could cooperate to control the two RuvB AAA+ motors (Fig. 1c, e). Notably, all four RuvB-coordinating RuvAD3 localize to the same side of the HJ crossover (Extended Data Fig. 5a), implying that a single RuvA tetramer might be sufficient to operate both RuvB motors simultaneously. These findings are also in agreement with the proposed architecture of the RuvABC resolvosome, in which the HJ is believed to be sandwiched by one RuvA tetramer and a dimer of the resolvase RuvC (Extended Data Fig. 5b)^{41,42}.”

<<<

Referee #3:

Summary:

The work of Wald et al. uses time-lapse cryo-EM to understand how RuvA and RuvB coordinate to drive branch migration. This paper is a tough read with a lot of details and data, and the images/movies are beautiful. It has the potential to make a significant contribution to the field of homologous recombination and DNA repair.

Reply:

We thank reviewer #3 for critically evaluating our manuscript and suggesting improvements in order to improve the manuscript. We do share the opinion of reviewer #3 that our manuscript contains lots of details. We also sympathize that the wealth of data can turn into a ‘tough read’. We thus streamlined the text in our revised manuscript including rewriting parts of paragraphs, figure callouts, re-worked the notification system, and added now an integrated model paragraph at the end of the ‘results section’ to provide the reader with a concise working model.

We are excited and thrilled to hear from reviewer #3 that the content of our work has the potential to make a significant contribution to the field of homologous recombination and DNA repair. We replied to all points raised and hope that our work is acceptable for publication.

However, there are some critical issues that detract from this study. First, and this was very disappointing, the authors used RuvA from *Salmonella* and RuvB from *Streptococcus*. No explanation was given and the authors stated that initial studies were done with *E. coli* proteins. So why not use the *E. coli* proteins? At its core, protein homology or lack thereof does not limit the quality of the work performed. However, it is impossible to relate residue numbers to *E. coli* numbering. This makes sense because a large body of data exists for the *E. coli* proteins and when trying to relate the mechanism proposed and residues used, it made

no sense.

Reply:

We agree and apologise that we were not explicitly clear about the relation to the *E. coli* system. One of our long-standing interests in our laboratory is to understand the underlying molecular mechanisms of pathogenicity and/or systems that eventually can be targeted to prevent bacterial infections by *Salmonella* spp. We thus started off our work with the RuvAB system from *Salmonella* and included other species such as

E. coli and *Streptococcus* throughout the study. We also learned that for structural studies, investigating RuvB from *Streptococcus* was the best choice. The procedure to converge onto a single system of a specific organism from a wide range of different organisms tractable for structural studies is not unusual and often almost generic to the beginning of many structural studies as observed by publications, personal communications or even platform-oriented approaches, such as the structural genomics consortia. It is interesting to note that to date structures for RuvB have only been reported from *Thermus thermophilus*, *Thermatoga maritima*, *Pseudomonas aeruginosa*, and *Campylobacter jejuni* sub. To ease comparison from our study with published structures as well as the *E.coli* system, which has been widely used especially for biochemical studies in the past, we have added the following information to the revised manuscript and have added a table comparing critical residues in *E.coli* and *S. thermophilus* (Extended Data Table 3).

Line number 137:

“To aid comparability with the *E. coli* RuvAB system, the corresponding residues are listed in Extended Data Table 3.”

Finally, we also provide RuvB protein sequence alignment between *S. thermophilus*, *E.coli* and *S. typhimurium* (Extended Data Fig Table 6).

Also and very important, does the interspecies RuvAB complex catalyze branch migration as efficiently as the *E. coli* RuvAB control?

Reply:

The revised manuscript now contains additional biochemical data elaborating the ATP dependent Holliday junction branch migration activity of the *in vitro* assembled homo- and hetero- complexes. We now show in (Fig. 1b, Extended Data Fig. 1c-e) that similar biochemical activities are observed between the homo- and heterocomplexes suggesting that individual components from different species can be exchanged while keeping the system active. This observation is also in line with previous studies^{1,2} highlighting that the underlying mechanism of branch migration employed by RuvAB is highly conserved. We now also mention that we focused our studies on hetero-complexes, because during the course of sample optimization only hetero-complexes could be vitrified suitable for structural studies.

The following sentence has been introduced: *at line number 79*:

“To visualize this process, we reconstituted RuvAB-HJ complexes *in vitro* from individually purified components originating from *Salmonella typhimurium* and *Streptococcus thermophilus*, respectively, and tested their functionality in a branch migration assay (Fig. 1b). Both, homo (RuvA and RuvB from *S. typhimurium*, and hetero (RuvA from *S. typhimurium* and RuvB from *S. thermophilus*) complexes processed the HJ similarly upon addition of ATP, suggesting a highly conserved underlying mechanism due to interchangeability of individual components (Fig. 1b, Extended Data Fig. 1a-e).”

Then, the authors have omitted multiple key references and as a result, they do not do a good job of relating their mechanism to what is known for RuvAB. This is significant. In particular the single molecule study from the Harada group showing that the DNA rotates during branch migration is not included. Further, the control of the ATPase cycle of RuvB by RuvA domain III has already been published, so this is not new. The authors did not do a good job relating their mechanism to what is already known.

Reply:

We apologise if we have missed to cite key references and are grateful that we now have the opportunity to cite previous work within our revised manuscript.

References critical to the ATPase mechanism (15, 16):

(15)

Marrione, P. E. & Cox, M. M. RuvB protein-mediated ATP hydrolysis: functional asymmetry in the RuvB hexamer. *Biochemistry* **34**, 9809–9818 (1995).

(16)

Marrione, P. E. & Cox, M. M. Allosteric Effects of RuvA Protein, ATP, and DNA on RuvB Protein-Mediated ATP Hydrolysis†. *Biochemistry* **35**, 11228–11238 (1996).

Line number 51:

“In prokaryotes, the two proteins RuvA and RuvB play critical roles in the processing of the HJ by promoting the ATP-dependent unidirectional strand exchange reaction known as active branch migration^{4–16}.”

References for the control of the ATPase activity of RuvB by the RuvA domain III (11,26,27,44,58):

(11)

Shiba, T., Iwasaki, H., Nakata, a & Shinagawa, H. SOS-inducible DNA repair proteins, RuvA and RuvB, of *Escherichia coli*: functional interactions between RuvA and RuvB for ATP hydrolysis and renaturation of the cruciform structure in supercoiled DNA. *Proc Natl Acad Sci U S A* **88**, 8445–8449 (1991).

(26)

Ohnishi, T., Hishida, T., Harada, Y., Iwasaki, H. & Shinagawa, H. Structure-function analysis of the three domains of RuvB DNA motor protein. *Journal of Biological Chemistry* **280**, 30504–30510 (2005).

(27)

Nishino, T. *et al.* Modulation of RuvB function by the mobile domain III of the holliday junction recognition protein RuvA. *Journal of Molecular Biology* **298**, 407–416 (2000)

(44)

Yamada, K. *et al.* Crystal structure of the Holliday junction migration motor protein RuvB from *Thermus thermophilus* HB8. *Proceedings of the National Academy of Sciences* **98**, 1442–1447 (2001).

(58)

Mitchell, A. H. & West, S. C. Role of RuvA in branch migration reactions catalyzed by the RuvA and RuvB proteins of *Escherichia coli*. *Journal of Biological Chemistry* **271**, 19497–19502 (1996).

Line number 53 (for References 26, 27)

“Previous biochemical and structural evidence suggests that branch migration is facilitated by a tripartite complex: RuvA tetramers assemble around the HJ crossover to provide structural guidance for DNA separation and rewinding and are flanked by two hexameric RuvB AAA+ (ATPase Associated with various cellular Activities) helicases that together fuel the translocation of the newly-emerged recombined DNA^{17–28}.”

Line number 128 (for References 26, 27,44)

“Consistent with previous structural and interaction studies, RuvB oligomerization is driven by the large (RuvB^L) and small (RuvB^S) ATPase domains of adjacent subunits (Extended Data Fig. 4c-e)^{26,27,44}.”

Line number 254 (for References 11, 58)

“The fact that we observed specific binding of RuvA^{D3} to the RuvB hexamer opposite the catalytic centre in subunit A through all states (s1 to s5) at the bottom of the staircase does not explain an increase in ATPase activity in the presence of RuvA^{11,58}.”

References for the rotation of DNA by single molecule studies (43):

(43)

Han, Y.-W. *et al.* Direct observation of DNA rotation during branch migration of Holliday junction DNA by *Escherichia coli* RuvA-RuvB protein complex. *Proceedings of the National Academy of Sciences* **103**, 11544–11548 (2006).

Line number 111:

“Taken together, we reasoned that the reconstituted RuvAB complex is enzymatically active and, therefore has been imaged in distinct conformational states. Moreover, our data reveal that the previously described continuous rotation of the DNA substrates⁴³ is accompanied by a concomitant rotation of the RuvB AAA+ motors themselves.”

Line number 365:

“Our results lead us to propose a model for initiation and processive branch migration which postulates that DNA translocations occurs through a lever-like mechanism executed and controlled by the RuvA-tethered RuvB hexamer combined with previously observed DNA rotation⁴³.”

References for the single molecule studies elaborating the kinetic of RuvAB branch migration (37, 38):

(37)

Amit, R., Gileadi, O. & Stavans, J. Direct observation of RuvAB-catalyzed branch migration of single Holliday junctions. *Proc Natl Acad Sci U S A* **101**, 11605–11610 (2004).

(38)

Dawid, A., Croquette, V., Grigoriev, M. & Heslot, F. Single-molecule study of RuvAB-mediated Holliday-junction migration. **103**, (2004).

Line number 78:

“Branch migration of Holliday junctions driven by the RuvAB machinery is a fast and highly dynamic process that is essential during DNA recombination (Fig. 1a)^{37,38}.”

Finally, the issue of branch migration being driven by a RuvAB complex with either one or two RuvA tetramers bound is critical. This is vaguely addressed early on, but the authors must do a better job as this is an important issue.

Reply:

This point has been raised before. For convenience, we provide the full answer here as well:

>>>

Our structural investigation clearly showed that **two RuvA tetramer** complexes were reconstituted onto Holliday junctions. Of note, for the *in vitro* reconstitution experiments, we used individually purified components: **RuvA** was purified to homogeneity and characterised as a **monomer as assessed by gel filtration** (Extended Data Fig 1a).

Interestingly, the interpretation of the RuvAB-HJ complex structure revealed that a single RuvA tetramer constitutes the operational unit for branch migration during which the flexible RuvA^{D3} domains interact with the two opposing hexameric RuvB motors (two RuvA^{D3} domains operate one RuvB motor). This represents a new concept, describing a configuration that is compatible with the observed rotation (and the physical constraints of an extended linker between the RuvA core and its D3 domain), and is also in agreement with the subsequent HJ resolution step during which one RuvA tetramer bound at on one side of the HJ is exchanged for the RuvC nuclease at the same binding location (“crossover junction endodeoxyribonuclease”) to generate the so called RuvABC-HJ resolvosome. We think that the

arrangement of two RuvA tetramers in the branch migration complex likely provides further biochemical stability for the complex by the second RuvA tetramer and thus may allow for efficient branch migration (as also proposed previously).

This is now also discussed *at line number 91*:

“The cryoEM structure of the RuvAB-HJ complex revealed highly flexible and linearly arranged bipartite (3.9 Å) and tripartite (8 Å) assemblies, with eight RuvA molecules symmetrically arranged in two tetramers (3.3 Å) and the four-way HJ flanked by, and flexibly connected to, one or two RuvB hexamers (2.9 - 4.1 Å), respectively (Fig. 1c-e, Extended Data Fig. 2, 3a-d, 4a-b, Extended Data Table 1-2). This architecture is consistent with previously proposed models of the RuvAB machinery^{22,23,25,31,32,40}. In both particle types, DNA enters and exits the RuvA core as a double helix, with one or two hexameric RuvB motors engaging the minor groove of the rejoined DNA (Fig. 1f). The RuvA core is physically connected to both RuvB motors through domain III of RuvA (RuvA^{D3}) (Fig. 1c). On either side, two RuvA^{D3} are bound to adjacently positioned RuvB subunits, indicating that these domains could cooperate to control the two RuvB AAA+ motors (Fig. 1c, e). Notably, all four RuvB-coordinating RuvA^{D3} localize to the same side of the HJ crossover (Extended Data Fig. 5a), implying that a single RuvA tetramer might be sufficient to operate both RuvB motors simultaneously. These findings are also in agreement with the proposed architecture of the RuvABC resolvosome, in which the HJ is believed to be sandwiched by one RuvA tetramer and a dimer of the resolvase RuvC (Extended Data Fig. 5b)^{41,42}.”

<<<

Listing of issues:

- 1) I had great difficulty trying to relate residues numbers in the text to what is published. When I pored over the M&M, I found out why. The authors use RuvA from *Salmonella typhimurium* and RuvB from *Streptococcus thermophilus*. This boggles the mind why this was done. They compare their data to the large body from the *E. coli* literature. The *E. coli* and *Streptococcus thermophilus* RuvB proteins are 60% identical, while the RuvA proteins are 95% identical. They used the *Streptococcus thermophilus* numbering to explain the data, and this made it difficult to follow when keeping *E. coli* numbering in mind.

Reply:

We apologise for the inconvenience. We now provide an additional table (Extended Data Table 3) to allow easy comparison to *E.coli* RuvB AAA+ and provide the reference to the table *at line number 137*:

“To aid comparability with the *E. coli* RuvAB system, the corresponding residues are listed in Extended Data Table 3”.

Finally, we also provide RuvB protein sequence alignment between *S.thermophilus*, *E.coli* and *S.typhimurium* (Extended Data Table 6).

- 2) Use of the word “protomer” is inaccurate and confusing and also frustrating. The definition is the smallest structural unit of an oligomeric protein composed of at least two different protein chains. The work of the Cox lab to understand the ATPase activity of RuvAB determined that RuvB monomers are arranged or working together as a trimer of dimers. Thus when reading I was trying to understand how the trimer of dimers with each dimer being a protomer could function to drive duplex translocation. In the end I concluded that the authors mean a monomer within the hexamer. Thus confusing terminology has to be changed or clearly defined at the outset.

Reply:

Thank you very much for pointing that out. We agree that using the word “protomer” could be inaccurate or confusing. Thus, to avoid any misunderstanding, we have changed the word “protomer” to “subunit” to make clear that this represents a single protein chain within a complex structure. This corresponds also to our deposition of the structure files to the PDB, where we even use the same letters for individual chains as now shown in all our figures. We do hope that reviewer #3 agrees on this.

- 3) The references from the Cox lab are not cited. These are critical to the ATPase mechanism proposed by the authors. See *Biochemistry* 34, 9809-9818 and *Biochemistry* 35, 11228-11238.
- 4) DNA rotation has been published and is discussed in *Proc. Natl Acad. Sci. U. S. A.* 2006 103, 11544-8. This paper has not been included.
- 5) Two additional single molecule papers relevant to this study are also absent. These are DOI: 10.1073/pnas.0404369101 and DOI: 0404332101 [pii]10.1073/pnas.0404332101.

Reply to point 3-5:

All citations (points 3-5) mentioned above are now included and cited in our revised manuscript. We would like to thank reviewer #3 for his/her recommendations.

- 6) The role of RuvB R174 as an allosteric effector in the ATPase activity of RuvB is not discussed. This is the work of Hishida et al. (*Proc. Natl Acad. Sci. U. S. A.* 2004 101, 9573–9577). This paper is also not cited. This may correspond to some of the arginine residues discussed in the text, but this was not made clear.

Reply:

We would like to thank reviewer #3 for pointing this out. The role of R171 [R174 in *E.coli*] was previously well characterized in the work of *Hishida et al.* (55) In our revised version of the manuscript, we discuss the importance of the canonical conserved arginine finger and the mutagenesis study presented by *Hishida et al.* at line number 222:

“In agreement with previous studies, ATPγS-Mg²⁺ *trans*-sensing is achieved by two elements: a conserved signature motif (127-Glu-Asp-130), located on alpha helix α4, and *trans*-acting Arg171 on

alpha helix $\alpha 5$ (Fig. 3f)^{51,54}. As such, Arg171 represents the canonical arginine finger that is conserved in most AAA+ ATPases and directly coordinates the γ -phosphate⁵⁵.”

- 7) "Oligonucleotides to construct Holliday junctions were purified by agarose gel electrophoresis.": Either this is an error in writing, or the DNA molecules are not homogeneous. The arms of each HJ are 56 nt in length. I doubt that an agarose gel would enable purification of this length of DNA.

Reply:

Thank you for spotting this error, which is now fixed in the revised version. Indeed, we prepared the HJ using native polyacrylamide electrophoresis (PAGE).

- 8) Line 248: Change “RuvA operates” to “RuvA regulates”.

We changed the title of the paragraph to: “Regulation of the nucleotide cycle in RuvB hexamers by forward and retrograde inter-subunit signalling” (*Line number 252*)

- 9) Line 365: A reference for the concentrations of ATP and ADP must be provided.

Reply:

Fixed. We have added a reference to the revised manuscript

(67)

Meyrat, A. & von Ballmoos, C. ATP synthesis at physiological nucleotide concentrations. *Scientific Reports* **9**, (2019).

Line number 359:

“Given that ATP levels typically exceed those of ADP in bacterial cells⁶⁷, it appears likely that *in vivo* RuvB motors first assemble initiation states by preferentially loading ATP stochastically at RuvB subunits (s0 with 4 or 5 ATPs), to then enter the processive sequential nucleotide cycle (s0→s1→→s5) to promote branch migration (Fig. 6).”

- 10) In the Introduction, state that the HJ is a central intermediate in recombination and fork rescue (cite - *Nucleic Acids Res.* 2009 37, 3475-92 and/or *Nucleic Acids Res.* 2021 49, 4220-4238).

Reply:

Thank you for this suggestion, which is now incorporated in the introduction of our revised manuscript.

Line number 49:

“The central and universal element in genetic recombination as well as in double strand break repair and in the process of replication fork rescue is a four-way DNA heteroduplex called Holliday junction (HJ)¹⁻³.”

11) The authors used a mixture of ATP- γ -S and ADP to slow reactions down. They need to show that this mixture is competent for branch migration.

Reply:

In Fig. 1b and Extended Data Fig. 1c-e we show that *in vitro* reconstituted particles perform active branch migration in the presence of ATP. Of note, ATP γ S used for the structural analysis in this work, is a (very) slowly hydrolysable ATP analogue (it often is called ‘non-hydrolysable’ as the rate of hydrolysis is 1/200th that of ATP^{1,2} and the products are often not detected within the observed reaction time). To reach the same endpoint in our branch migration assay as compared with using ATP, an incubation time of at least ~50 hours of the branch migration reaction is needed. Since we found that RuvAB-HJ particles start to disassemble beyond 5 hours after *in vitro* reconstitution, the data obtained from such an extensive incubation time would be difficult to interpret. Instead, we performed structural analysis after 5 hours of incubation and were able to resolve the entire catalytic cycle starting from bound ATP in state s1, necessary nucleophilic attack in state s2 and the subsequent conversion into ADP (cleavage of the gamma bond and Mg²⁺ release). This analysis serves as structural proof for activity of the catalytic reaction. To test, whether a short incubation time has not reached the catalytic cycle yet, we determined structures from a sample that has only been incubated for only 30 minutes after the *in vitro* reconstitution. This experiment confirmed our hypothesis. In the light of our discovery, our experiments might serve for other scientists to re-evaluate their experimental design and data to structurally investigate oligomeric AAA+ proteins. (Of note, studies using slowly hydrolysable ATP γ S or non-hydrolysable AMP-PNP^{3,4} revealed different structures, suggesting that in order to mimic active catalytic reactions, ATP γ S should be used.)

12) For the reconstitution of RuvA-B-ATP- γ -S/ADP complexes, the authors purify RuvA-HJ complexes by gel filtration. Under the conditions used, do they separate the one RuvA tetramer-HJ complex from the two tetramer complex? If so, which complex did they use for reconstitution, and does this influence the images obtained

Reply:

The concentration of RuvA in the first step of reconstitution was based on the electromobility shift gel assay (Extended Data Fig. 1f), where at the protein concentration ~500 [nM] only complex II (two RuvA tetramers bound to HJ and RuvB) is formed. For further clarification, we include the gel filtration profile of the RuvA-HJ complex (Extended Data Fig. 1g), which further suggests preferential formation of complex II at the given RuvA concentration. Thus, for the *in-vitro* reconstitution and following visualisation and 3D reconstruction RuvA complex II was used. Moreover, in our cryoEM analysis complex I (single RuvA tetramer bound to HJ and RuvB) could not be observed.

Suggestions for improvement:

1) Cite the relevant references and relate the current data to what is already known.

Reply:

We added suggested reference to the revised manuscript.

- 2) Create a cartoon model of the step-wise mechanism, indicating the power stroke, cluster switch and converter module, etc.

Reply:

We introduced following paragraph summarizing the model of RuvAB branch migration and provide a cartoon models for the transition between states, cluster switch and power stroke (Fig. 5 d, f).

Line number 364:

“Integrative model for initiation and processive branch migration of the RuvAB complex”

- 3) Fix the residue numbering.

Reply:

We introduced a table comparing critical residues from both species.

- 4) Define protomers or use monomers: either RuvB monomer or RuvA monomer.

Reply:

We renamed “protomers” to “subunit” throughout the revised manuscript.

1. Yasuoka, K., Kawakita, M. & Kaziro, Y. Interaction of Adenosine-5'-O-(3-Thiotriphosphate) with Ca²⁺, Mg²⁺- Adenosine Triphosphatase of Sarcoplasmic Reticulum. *The Journal of Biochemistry* **91**, 1629–1637 (1982).
2. Product Information/Data sheet:
www.sigmaaldrich.com/deepweb/assets/sigmaaldrich/product/documents/712/411/a1388pis.pdf .
3. Gates, S. N. *et al.* Ratchet-like polypeptide translocation mechanism of the AAA+ disaggregase Hsp104. *Science (1979)* **357**, 273–279 (2017).
4. Jean, N. L., Rutherford, T. J. & Löwe, J. FtsK in motion reveals its mechanism for double-stranded DNA translocation. *Proc Natl Acad Sci U S A* **117**, 14202–14208 (2020).

Reviewer Reports on the First Revision:

Referees' comments:

Referee #1:

The authors have addressed all issues raised on the prior round of review. Publication is recommended.

Referee #2:

Overall, the paper is significantly more readable and streamlined, and I commend the authors for doing a fantastic job with the revisions. The simplified notation system is now much clearer, and the detailed discussion of the model for branch migration at line 364 in particular is a great addition to the paper. The inclusion of experiments showing branch migration activity of the hetero-complex also strengthens the manuscript. I'd like to thank the authors for their responses to points raised previously, all of my queries have been adequately addressed and I'm happy to recommend publication of this exciting manuscript. I only have a few minor points as detailed below:

- In the legend for Fig. 1b, please specify what the label 'P' represents (presumably product).
- Thank you for now including the resolution numbers in the main text. However, rather than just saying '3.9 Å' in brackets (line 92), please be clear that this refers to the resolution of the maps by including a phrase like "solved to a resolution of 3.9 Å" or something similar, at least for the first map. It's clear to people in the cryo-EM field that these numbers refer to the resolutions of the map, but for the broad readership of Nature it's best to avoid possible ambiguity here.
- Please specify in the legend for Ext. Data Fig. 1i) whether the negative stain EM micrograph corresponds to the homo- or hetero- complex.
- The RuvA monomer shown in Ext. Data Fig. 4a) is coloured blue, but the figure legend refers to domains coloured green and grey – please ensure that these match up.
- Where full stops are used to break up numbers larger than 1,000 they should be replaced with commas in Ext. Data Tables 1 and 2, and Ext. Data Fig. 2 for clarity.
- Ext. Data Table 4 should be updated with the PDB and EMDB IDs filled in.
- The legend for Ext. Data Fig. 9 refers to a panel d), but this does not appear in the figure itself.
- In the legend for Ext. Data Fig. 10, it could be useful to specify that the "ADP*" notation refers to ADP + Mg²⁺.

- In the text at line 336, Ext. Data Fig. 2d-e are referred to for the t2 dataset states s0-A and s0, but it

appears that these actually appear in Ext. Data Fig. 2c.

Referee #3:

The authors have done an excellent job addressing my concerns. As a result I have no complaints, and I suggest the paper be accepted.

Author Rebuttals to First Revision:

We would like to thank the editorial board and the three referees for carefully reading our revised manuscript and providing their expert opinion and constructive feedback during the reviewing process. We are delighted that all three referees recommend publication.

Point-to-Point reply to reviewer comments:

Referee #1:

The authors have addressed all issues raised on the prior round of review. Publication is recommended.

We would like to thank referee #1 for his/her expert opinion and the recommendation to publish our manuscript.

Referee #2:

Overall, the paper is significantly more readable and streamlined, and I commend the authors for doing a fantastic job with the revisions. The simplified notation system is now much clearer, and the detailed discussion of the model for branch migration at line 364 in particular is a great addition to the paper. The inclusion of experiments showing branch migration activity of the hetero-complex also strengthens the manuscript. I'd like to thank the authors for their responses to points raised previously, all of my queries have been adequately addressed and I'm happy to recommend publication of this exciting manuscript.

We would like to thank referee #2 for his/her expert opinion and the recommendation to publish our manuscript.

I only have a few minor points as detailed below:

- In the legend for Fig. 1b, please specify what the label 'P' represents (presumably product).

Thank you very much for pointing this out. Indeed, 'P' represents 'products' and has now been included in the figure legends.

- Thank you for now including the resolution numbers in the main text. However, rather than just saying '3.9 Å' in brackets (line 92), please be clear that this refers to the resolution of the maps by including a phrase like "solved to a resolution of 3.9 Å" or something similar, at least for the first map. It's clear to people in the cryo-EM field that these numbers refer to the resolutions of the map, but for the broad readership of Nature it's best to avoid possible ambiguity here.

We agree that technical details should be explained for the broader readership at the first instance and have incorporated this suggestion into our main text.

- Please specify in the legend for Ext. Data Fig. 1i) whether the negative stain EM micrograph corresponds to the homo- or hetero- complex.

Fixed.

- The RuvA monomer shown in Ext. Data Fig. 4a) is coloured blue, but the figure legend refers to domains coloured green and grey – please ensure that these match up.

Fixed.

- Where full stops are used to break up numbers larger than 1,000 they should be replaced with commas in Ext. Data Tables 1 and 2, and Ext. Data Fig. 2 for clarity.

Full stops and commas have been removed in order to avoid any potential confusion.

- Ext. Data Table 4 should be updated with the PDB and EMDB IDs filled in.

Fixed.

- The legend for Ext. Data Fig. 9 refers to a panel d), but this does not appear in the figure itself.

Fixed.

- In the legend for Ext. Data Fig. 10, it could be useful to specify that the “ADP*” notation refers to ADP + Mg²⁺.

Fixed (now as part of Ext. Data Fig 7)

- In the text at line 336, Ext. Data Fig. 2d-e are referred to for the t2 dataset states s0-A and s0, but it appears that these actually appear in Ext. Data Fig. 2c.

Fixed.

Referee #3:

The authors have done an excellent job addressing my concerns. As a result I have no complaints, and I suggest the paper be accepted.

We would like to thank referee #3 for his/her expert opinion and the recommendation to publish our manuscript.

Once again, we would like to thank all three reviewers for their excitement of our findings, their critical evaluation and constructive comments of our data. Together this allowed us to improve our manuscript and make it accessible for a broad readership.